# In-ear integrated sensor array for the continuous monitoring of brain activity and of lactate in sweat

Yuchen Xu [1,4], Ernesto De la Paz[2,4], Akshay Paul[1], Kuldeep Mahato[2], Juliane R. Sempionatto [2], Nicholas Tostado[2], Min Lee [1], Gopabandhu Hota[3], Muyang Lin[2], Abhinav Uppal [1], William Chen[1], Srishty Dua[2], Lu Yin[2], Brian L. Wuerstle[3], Stephen Deiss[1], Patrick Mercier [3]✉, Sheng Xu [1,2]✉, Joseph Wang [2]✉ & Gert Cauwenberghs [1]✉

Owing to the proximity of the ear canal to the central nervous system, in-ear electrophysiological systems can be used to unobtrusively monitor brain states. Here, by taking advantage of the ear's exocrine sweat glands, we describe an in-ear integrated array of electrochemical and electrophysiological sensors placed on a flexible substrate surrounding a user-generic earphone for the simultaneous monitoring of lactate concentration and brain states via electroencephalography, electrooculography and electrodermal activity. In volunteers performing an acute bout of exercise, the device detected elevated lactate levels in sweat concurrently with the modulation of brain activity across all electroencephalography frequency bands. Simultaneous and continuous unobtrusive in-ear monitoring of metabolic biomarkers and brain electrophysiology may allow for the discovery of dynamic and synergetic interactions between brain and body biomarkers in real-world settings for long-term health monitoring or for the detection or monitoring of neurodegenerative diseases.

Wearable sensing technologies have greatly expanded health monitoring and human–machine-interface applications, bridging the gaps between traditional clinical instrumentation and the urgent demand for remote and daily health care. Recent advances in high-performance, stretchable and conformal skin interfaces[1,2] are yielding opportunities for unobtrusive physiological[3,4] and metabolic[5,6] monitoring in a highly wearable setting. Several of these sensing modalities can be integrated to offer greater functionality where biosignals of interest are conveniently co-located[7,8].

Among the many sensing modalities, electrophysiological brain-state monitoring and health-related metabolite monitoring are two dimensions that may have substantial implications for early disease detection, health monitoring, body-performance improvement and virtual/augmented-reality applications. Brain-state monitoring, widely employed in brain–computer interfaces, has been used for neuromodulation and rehabilitation, whereas the monitoring of vital metabolites provides real-time analytics on dynamically changing health conditions. Previous medical studies have suggested that some of the day-to-day cognitive state variations such as stress[9] and emotions[10], as well as neurodegenerative diseases, such as epilepsy[11] and Alzheimer's disease[12], can trigger characteristic patterns in both electrophysiological brain-state monitoring and produce abnormal

[1]Shu Chien - Gene Lay Department of Bioengineering, University of California San Diego, La Jolla, CA, USA. [2]Department of Nanoengineering, University of California San Diego, La Jolla, CA, USA. [3]Department of Electrical and Computer Engineering, University of California San Diego, La Jolla, CA, USA. [4]These authors contributed equally: Yuchen Xu, Ernesto De la Paz. ✉e-mail: pmercier@ucsd.edu; shengxu@ucsd.edu; josephwang@ucsd.edu; gcauwenberghs@ucsd.edu

metabolic profiles in an individual. For instance, lactate monitoring has been found to complement electroencephalogram (EEG) recording for the differentiation of generalized epileptic seizures from psychogenic non-epileptic and syncopal events in monitoring epileptic seizures[13,14]. Clinical evidence has further corroborated the conducive role of the monitoring and control of metabolite levels (such as those of lactate) in improving brain functions by enhancing neuroplasticity and angiogenesis[15,16]. These results highlight the need for integrated monitoring of brain states and health-related metabolites.

EEG collected on the scalp with gel-based electrodes allows great spatial coverage and high signal-to-noise (SNR) ratio in active brain-state monitoring, but at the expense of restricting the user's mobility and comfort. Dry-contact EEG electrodes provide much improved user comfort and reduced setup time, but at a loss in SNR primarily when used over hairy sites on the scalp[17]. Whereas skin metabolic health monitoring has been demonstrated using skin-penetrating tools (small filaments, or blood pricking/sampling), non-invasive technologies (such as epidermal patches) or optical procedures (via near-infrared or ultraviolet–visible light or Raman spectroscopy)[18], their availability in the market as reliable commercial technologies is limited in the form of skin-penetrating filaments or blood collection approaches that require small or higher volumes of sample to perform metabolite analysis[19]. In addition, despite the extensive evidence supporting their isolated analysis studies, the integration of these two sensing modalities within a single wearable technology still presents a major challenge[20]. This is in part due to interference present in the measured signals attributed to crosstalk between the sensors of dissimilar sensing modalities[21,22] and the disparity in optimal combined electrophysiological and electrochemical sensing (co-sensing) locations on the body, typically requiring extensive form factors spanning a wide area across the body, such as in an EEG headset or headband[23].

In-ear electrophysiological sensing systems[24,25] provide elegant solutions to unobtrusive brain-state monitoring inside the ear canal. The ear is located close to the central nervous system, major vasculature and auditory cortex while being mechanically stable because of the ear's anchoring structure by nature. In addition to access to physiological parameters, such as EEG, pulse rate and oxygen saturation[26], it has multiple exocrine sweat glands for the analysis of vital metabolites[27]. Due to the extremely limited space in the ear and large anatomical variations across ears, developing a user-generic ear sensor covering a broad range of biophysical modalities of interest to general health monitoring is still an extremely challenging goal[28]. While in-ear sensing of multiple physiological parameters has been demonstrated[24,25,29,30], integrating both brain-state and metabolite monitoring in a single unobtrusive system has remained elusive due to the relatively large form factors of conventional electrochemical sensors.

This work presents an unobtrusive and fully in-ear integrated array of multimodal electrophysiological and electrochemical sensors for simultaneous monitoring of the brain state and dynamic metabolic sweat concentration. Such integrated in-ear electrophysiological and electrochemical system was realized through strategic material selection, layout design and fabrication engineering that not only fitted the irregular ear anatomy between different individuals but also provided simultaneous and real-time operation of both sensing modalities (Fig. 1a,b). Via tight integration with widely used in-ear earphones, the integrated sensors track daily activities along two principal sets of features in biosignal space characterizing general brain–body health state. The first set of features implements a generalized form of a wearable brain–computer interface (BCI) in the ear, tracking brain state-related electrophysiological signals, such as EEG and electrodermal activity (EDA). The second set of features implements electrochemical analysis of metabolites in the ear. For this study, lactate in sweat was selected as the analyte of choice. The combined brain-state and metabolite sensing offers a unique sensing modality for wearable monitoring addressing the relationship between EEG and lactate and

their relevance in a variety of health-related contexts, including traumatic brain injuries[31] and stress variation during the day[10,32]. The implementation of both modalities into a miniaturized in-ear non-invasive system with minimized sensing crosstalk could thus facilitate the process of using multiple instruments for assessing these features during neurological monitoring and potentially allow self-monitoring in patients.

For optimal integration, the layout of the sensors (Fig. 1c) was determined from functional mapping inside the ear canal, where electrophysiological electrodes were oriented towards the temporal lobe with lower secretion of sweat, while the electrochemical electrodes were oriented towards the location with higher secretion of sweat. Such functional mapping is conducive to obtaining a higher signal-to-noise ratio given the proximity to the signal sources while minimizing potential crosstalk. To determine this location, a multiparticipant series of experiments to map the areas with higher sweat secretion was performed using a participant-custom earpiece (Extended Data Fig. 1 and Supplementary Fig. 1) while performing stationary cycling at a fixed exercise level for 30 min. On the basis of the in-ear sweat mapping results, the electrochemical electrodes were oriented towards the tragus, where the highest sweat volume was found (Fig. 1i). To integrate with most earphone silicone tips, the sensors have a flat bottom with an adhesive layer (Fig. 1d). The sensor fabrication employed a fast and low-cost printing-bonding-assembly process (Fig. 1e and Extended Data Fig. 2), resulting in a single integrated device with high mechanical resiliency and efficient space utilization. Structurewise, the sensors consist of a 150-µm-thick terephthalate polyurethane (TPU) substrate with chemical resistance and stretchability. Serpentine interconnection traces and the lactate sensing electrochemical reference electrode (RE) were made with modified stretchable silver (Ag) ink. An insulation layer extends on top of the interconnections using stretchable styrene-ethylene-butylene-styrene block copolymer (SEBS). At the patterned openings of the SEBS layer, the electrophysiological electrodes and the electrochemical reference electrode were printed on top of the interconnection traces with modified stretchable Ag ink. The electrochemical working electrode (WE) and counter electrode (CE) were fabricated using stretchable Prussian-blue (PB) ink and further modified (Extended Data Fig. 3a,b). To account for the anatomical variation in ear shapes across participants and the geometrical mismatch between the round-arc earphone tip and the ear canal, the electrophysiological stretchable Ag electrodes adopt a three-dimensional (3D) structure (Extended Data Fig. 2q–v), while the three planar electrochemical electrodes are covered by a piece of polyvinyl alcohol (PVA) hydrogel (Extended Data Fig. 3c–e) to enhance sweat collection.

Sequential data acquisition from the in-ear sensors was achieved using a customized wearable system to allow real-time sampling and wireless data transmission. The electrophysiological sensing is based on potentiometric measurement by connecting the in-ear electrophysiological electrodes to a portable data acquisition (DAQ) system in the single-ended mode (Supplementary Fig. 3). Five stretchable Ag electrodes were made under the same conditions and used for electrophysiological sensing, including three in-ear electrophysiological electrodes, one electrophysiological reference electrode (REF) at the concha cymba and one driven right leg (DRL) electrode at the concha cavum (Fig. 1h–j). During the measurement, each electrophysiological electrode was measured with respect to the same REF, which went to the inverting input of the fully differential programmable gain amplifier at each channel of the DAQ. The DRL was used to suppress common-mode interference from power lines and other sources. Simultaneously, the electrochemical biosensing implements chronoamperometry (CA) for detecting the hydrogen peroxide product of the enzymatic reaction between the lactate oxidase (LOx)-modified working electrode and the lactate present in the secreted sweat. The in-ear sensors were characterized in a single and parallel sensing modality, both in vitro and on body,

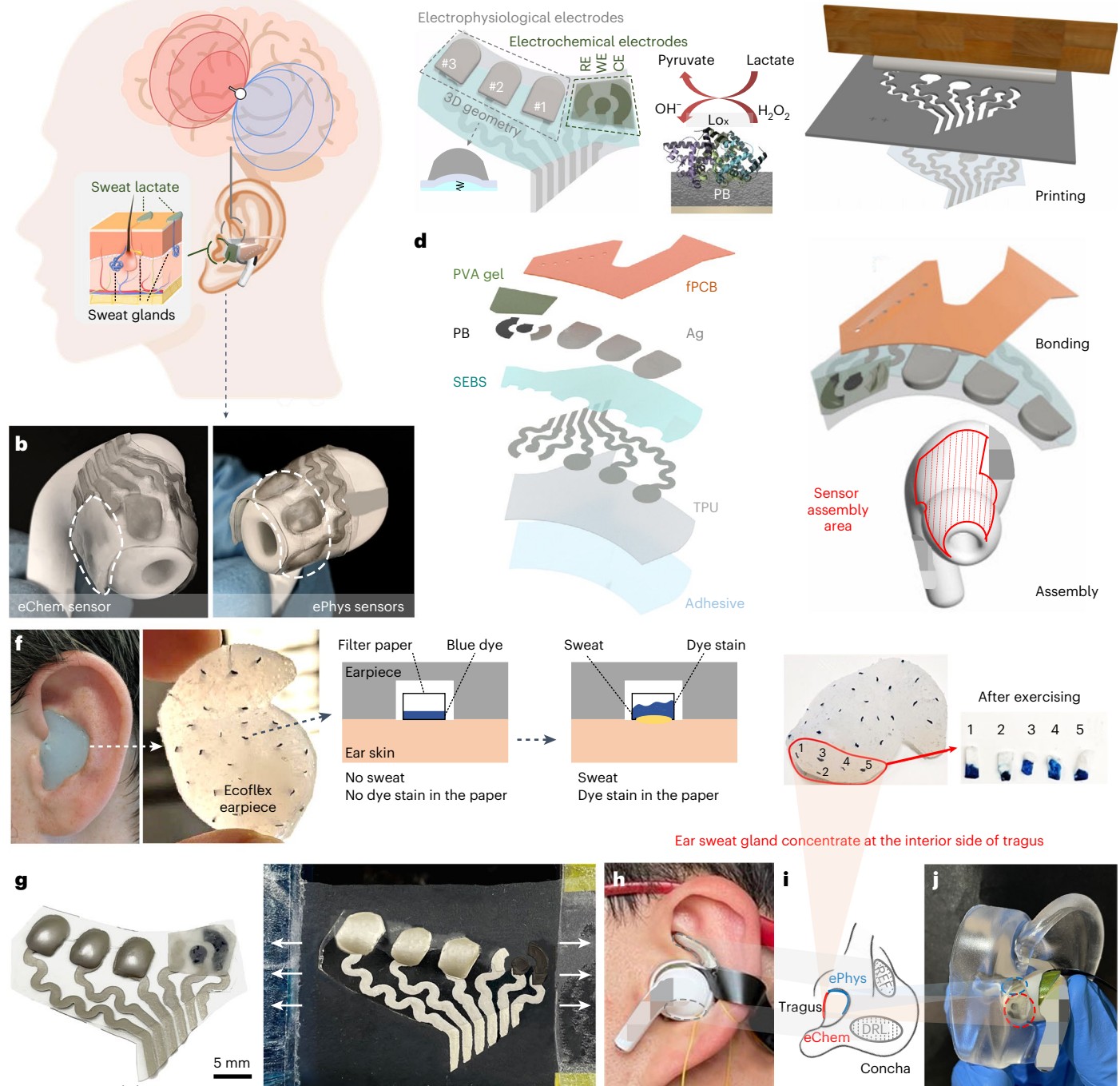

**Fig. 1 | Design of the in-ear integrated sensors. a**, Design schematic of the in-ear integrated sensors. The grey and green dashed outlines denote the electrophysiological and electrochemical sensor sites registering brain activity and sweat secretion, respectively. **b**, Assembled in-ear integrated electrophysiological (ePhys) and electrochemical (eChem) sensing electrodes. The white dashed outlines denote the locations of the electrophysiological and electrochemical electrodes. **c**, Layout of the in-ear integrated sensors (left) showing 3D cushioning of the electrophysiological electrodes and indicating the mechanism of sweat lactate electrochemical sensing. LOx on the working electrode surface (right) catalyses oxidation of lactate acid into pyruvate and hydrogen peroxide ($H_2O_2$). In turn, PB transforms $H_2O_2$ into hydroxyl ions ($OH^-$), generating a sensing current proportional to lactate concentration. **d**, Layer-by-layer structure. From bottom to top, the sensors were made of adhesive, TPU, SEBS, PB, stretchable Ag, PVA hydrogel and flexible printed circuit board (fPCB). **e**, Fabrication process of the sensors showing three main procedural steps: printing of sensors, bonding with electronics and assembly to earphones (details can be found in Extended Data Fig. 2). **f**, In-ear sweat mapping for electrochemical sensing using an Ecoflex earpiece with distributed filter papers, the distribution of which was used as sweat secretion indicators. **g**, Integrated sensors before assembly and after 20% latitudinal stretching. **h**, Assembled integrated sensors in the ear. The grey dashed outlines indicate the locations of electrophysiological REF and DRL electrodes. **i**, Geometry of the ePhys, REF, DRL and eChem electrodes. **j**, Skin contact locations of the ePhys and eChem sensors as revealed by insertion in an ear phantom.

with several human participants. Compared with a scalp EEG headset or a commercial blood lactate metre, the sensors have a considerably smaller form factor, perform similarly, are less obtrusive and are more comfortable to wear. Further details on the sensor design, fabrication and assembly are provided in Supplementary Note 1. Characterization of the robustness of the sensors is described in Supplementary Note 2.

## Results and Discussion

The electrophysiological sensing performance of the in-ear integrated sensors was characterized across multiple signal conditions, and controlled validation was conducted with a commercial dry-contact EEG headset (Supplementary Text Note 3 and Extended Data Fig. 7). The 3D electrophysiological electrode design for the in-ear integrated sensors produced a more robust electrode–ear interface, reducing the possibility of contact loss and increasing the effective contact surface (Extended Data Fig. 2q–v). The electrode–ear impedance was characterized by continuous measurement at 10 Hz (EEG alpha band, Fig. 2a,b) and by electrochemical impedance spectroscopy (EIS) over 1 Hz–1 kHz (Fig. 2c,d and Supplementary Note 3). The continuous impedance measurement revealed electrodermal activity in the ear in terms of a gradual and consistent decrease in impedance at the electrode–ear interface due to accumulation of sweat secreted at the contact area (Extended Data Fig. 4a,b,d,e). On average, the results show a 386 kΩ electrode–ear impedance at 50 Hz with a planar electrode size of 12.56 mm$^2$, comparable to the impedance of a state-of-the-art in-ear dry electrode (377 kΩ at 50 Hz for a 60 mm$^2$ electrode area)[25] but at reduced dimensions. Electrode direct current (DC) offset (EDO) for the in-ear integrated sensors was mainly due to the different electrophysiological and REF electrode sizes, the small impurities within the electrode surface and the slight ionic concentration variations under different sweating conditions. Figure 2e,f show normal distribution fit results with a mean and standard deviation of 0.59 mV and 21 mV, respectively, and a maximum recorded EDO (51.30 mV) that is well within the ±187 mV input range of the analogue front end used in this work.

On-body experiments showed several EEG signals measured by the in-ear integrated sensors. Alpha modulation is a spontaneous EEG pattern in the 8–12 Hz frequency range modulated by the participant's state of visual attention or relaxation. Figure 2g shows the synchronous emergence of alpha-band signals with the participant's eyes closed over two 1-min intervals. Figure 2h shows 4 participants' grand average alpha-band power spectral density (PSD) when they opened and closed their eyes for 1 min each. An evident rise in the alpha-band power was observed, characterized by an alpha modulation ratio of $R_{AM} = 2.44 \pm 0.66$ (V$^2$/V$^2$). The reported $R_{AM}$ is comparable to previously reported results from dry in-ear EEG sensors (average $R_{AM} = 1.2$ (ref. 24) and 2.17 (ref. 25)). Another widely studied evoked EEG pattern originating from the auditory cortex is the auditory steady-state response (ASSR), an ongoing oscillatory brain signal resulting from an acoustic stimulus's continuous amplitude modulation[33]. Figure 2i–l shows the grand average ASSR PSDs across 4 participants. ASSR response peaks corresponding to four 1-min ASSR stimuli (25 Hz, 40 Hz, 55 Hz and 70 Hz) were observed in the PSDs. SNRs of the 25 Hz, 40 Hz, 55 Hz and 70 Hz ASSRs were 12.80 ± 1.27 dB, 8.98 ± 2.26 dB, 10.63 ± 4.88 dB and 10.92 ± 1.88 dB, respectively. The reported ASSR SNRs were between previously reported results from dry in-ear sensors (average SNR of 5.94 dB (ref. 25) and approximate SNR of 15 dB (ref. 24)). Supplementary Note 3 also corroborates the effectiveness of ASSR measurement for in-ear integrated sensors compared with the EEG headset due to its proximity to the auditory cortex. Another electrophysiological sensing modality available in the ear is eye movement characterized by electrooculography (EOG), often considered as artefacts in EEG measurements[34] while finding utility for some BCI applications including drowsiness detection[35], eye vergence therapy[36] and motor control[37,38]. Figure 2m,n show that the in-ear integrated sensors, when using a reference electrode in the same ear, are more resilient to EOG

eye movement artefacts for purposes of EEG measurements (further discussed in Supplementary Note 3 and Extended Data Fig. 5). The in-ear electrophysiological sensors are intrinsically multimodal in that they simultaneously acquire biopotentials from various sources of brain and body electrical activity including, besides EEG and EOG, the electromyogram (EMG), along with electrochemical impedance registering EDA. The richness of concurrently present electrophysiological sensing modalities supports a comprehensive account of physiological processes undergoing an array of cognitive and emotional states while also being subject to motion and other artefacts. A detailed analysis of such simultaneous electrophysiological sensing as well as signal processing for motion-artefact rejection can be found in Supplementary Note 3 and Extended Data Fig. 6.

The performance of the in-ear electrochemical sensor was first evaluated in vitro under the established concentration range of sweat lactate[39]. As shown in Fig. 3a and Extended Data Fig. 8a, the successive additions of 5 mM lactate displayed an increase in the well-defined current response of the sensor. Further characterization of the sensor involved the selective response of the sensor in the presence of relevant interferent constituents found in sweat. As illustrated in Fig. 3b, a 2 mM addition of lactate increased the current response from the sensor. Successive additions of lactic acid (LA), acetaminophen (AC), ascorbic acid (AA), glucose (Gluc) and uric acid (UA) resulted in negligible current changes. Such specific response towards lactate reflects the combination of a selective bioreceptor and a low potential step of −0.2 V applied to the PB-based transducer. The relative change in response from the sensor after running 18 repetitive CA scans at a fixed concentration (that is, 10 mM) displayed in Fig. 3c showed minimal changes (<5%), demonstrating the efficient entrapment of the enzymatic layer on the electrochemical transducer. In addition, the sensor operational stability has been validated by a persistent response to a 10 mM step in lactate over a continuous scan for ~1 h (Fig. 3d). The analytical performance of the electrochemical sensors at various temperatures and humidity levels, matching environmental and human physiological conditions in the ear[40,41], has been evaluated for different lactate concentrations. Figure 3e shows the sensitivity in the slope of current vs concentration (μA/mM) at temperatures ranging from 25 °C to 40 °C. Negligible differences in slope were observed in this range of temperatures, probably owing to the retention of the activity of the lactate oxidase and stable sensor matrix. Similarly, the sensor showed consistent slope across humidity levels ranging from 40% to 70% (Fig. 3f), which could be attributed to the wet interfacing PVA gel resisting the impact of the fluctuating environmental humidity level. Further details on the electrochemical sensor stability characterization are presented in Supplementary Note 2 and Supplementary Fig. 7. After in vitro characterization, the response of the electrochemical sensor to lactate in sweat was evaluated before, during and after performing stationary exercise while wearing sensors with and without an enzymatic layer. On each step of the test, blood samples were collected using a commercial blood lactate metre for validation purposes. The sensor modified with the enzymatic layer showed an incremental current Δ$i$ of −0.4 μA compared with the initial values after 10 min of starting stationary exercise. Also, blood lactate levels displayed their maximum level at this stage of the experiment (Fig. 3g,h). Interestingly, the recovery of the signal close to the pre-exercise levels was observed a few minutes after the exercise. Such reversible behaviour of the sensor during the exercise stage could suggest a depleted concentration of sweat lactate which was generated during exercise. This behaviour is dependent on the sweat rate and is unique to every individual[42], which can be seen in another recorded exercise session in different participants (Supplementary Fig. 11). The difference in the CA fluctuation is evident in these figures (Fig. 3g and Supplementary Fig. 11), also corroborating the different degrees of perspiration among individuals. Following the culmination of exercise, the current displayed a drop in the signal and a decay in the blood lactate levels caused by the resting state of the

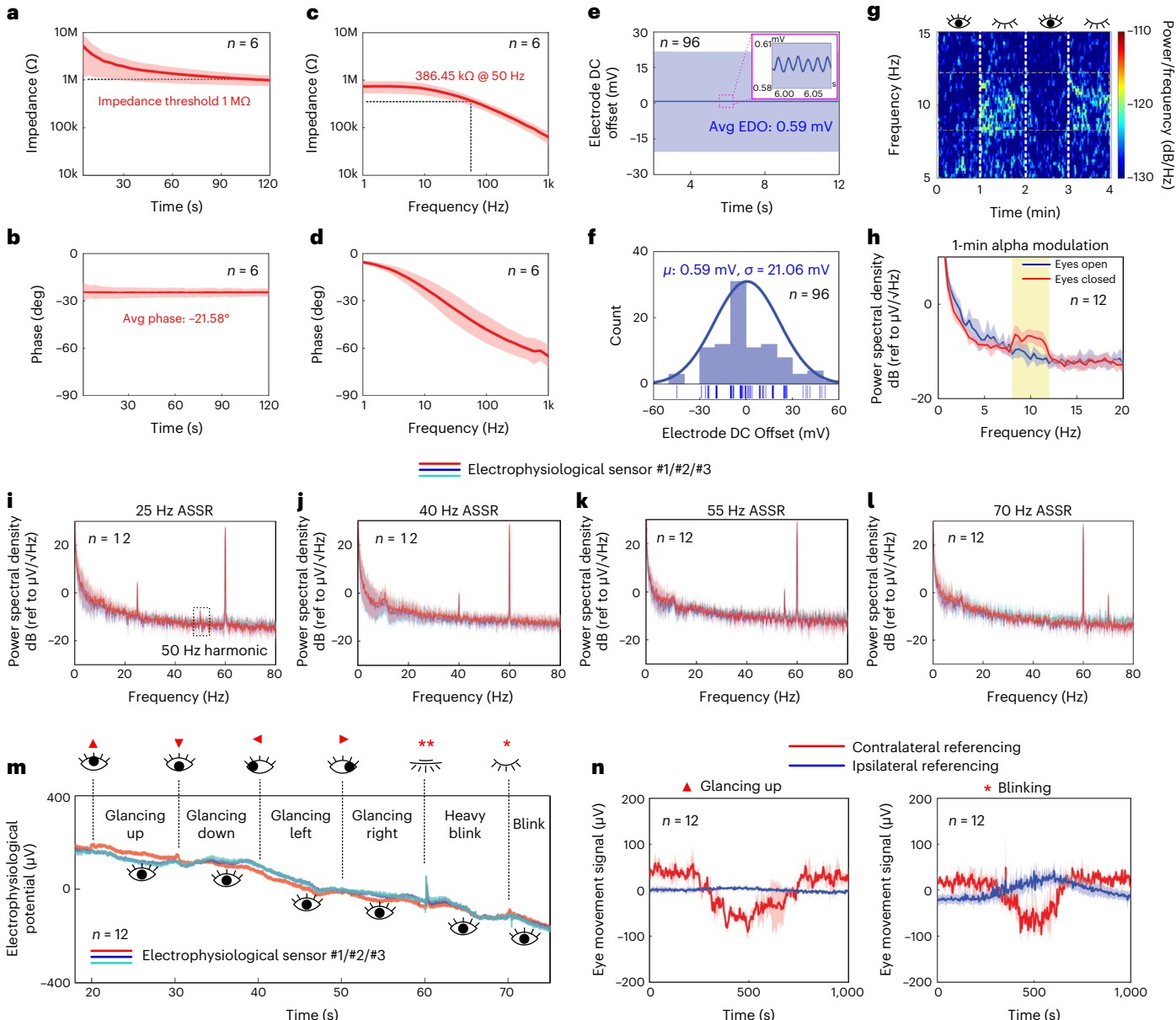

**Fig. 2 | Characterization of the in-ear electrophysiological sensing capability.** **a**–**d**, Electrode–ear impedance characterization with (**a**) impedance magnitude and (**b**) impedance phase over time upon insertion, and (**c**) impedance magnitude and (**d**) phase spectra in steady state after insertion for more than 2 min. **e**,**f**, EDO characterization with (**e**) temporal profile and (**f**) fitted normal distribution of the 96 EDO recordings. **g**,**h**, EEG characterization with (**g**) spectrogram and (**h**) power spectrum density (PSD) for alpha modulation experiments with participants opening and closing their eyes at 1-min intervals. The yellow-shaded interval denotes the 8–12 Hz alpha band in the EEG spectrum. **i**–**l**, Auditory steady state response (ASSR) PSDs for four acoustic stimuli amplitude modulated at (**i**) 25 Hz, (**j**) 40 Hz, (**k**) 55 Hz and (**l**) 70 Hz. **m**,**n**, EOG characterization. **m**, Transient response to various eye movements recorded within one ear with ipsilateral referencing. **n**, Comparison between contralateral and ipsilateral referencing in the time-averaged transient response to two representative eye movements: glancing up and blinking. Further EOG characterization can be found in Extended Data Fig. 5. The total number of measurements taken from each individual electrophysiological sensor per experiment is denoted as index $n$ in each panel. For each electrophysiological sensor, solid lines and shaded bands represent participant-averaged mean ± s.d. of the measured time series and PSD.

individual. In experiments without the enzymatic layer, negligible current changes were observed despite the increasing blood lactate levels during stationary exercise (Fig. 3i,j). Along with the selective response of the enzyme-modified sensor towards lactate, these results support the idea that the current changes obtained in Fig. 3g,h correspond to the electrochemical detection of lactate in sweat.

The in-ear integrated sensors combine potentiometric EEG electrophysiological and chronoamperometric lactate electrochemical sensors as the two primary sensing modalities. The conductive nature and tight space of the ear canal pose challenges in mitigating crosstalk between these two sensing modalities. Co-sensing crosstalk experiments showed minor transient crosstalk interference, which manifested as a brief artefact and a discharging phase to the electrophysiological measurement following the start of the lactate sensing measurement, taking place right after the driving voltage was applied to the electrochemical electrodes (Fig. 3k,l). Under two emulated exercise conditions with different concentrations of sweat, the co-sensing measurements revealed ASSR SNRs comparable to baseline measurement with continuous lactate measurement (Fig. 3m,n and Extended Data Fig. 9d,g,j), indicating that potentiometric and CA measurements

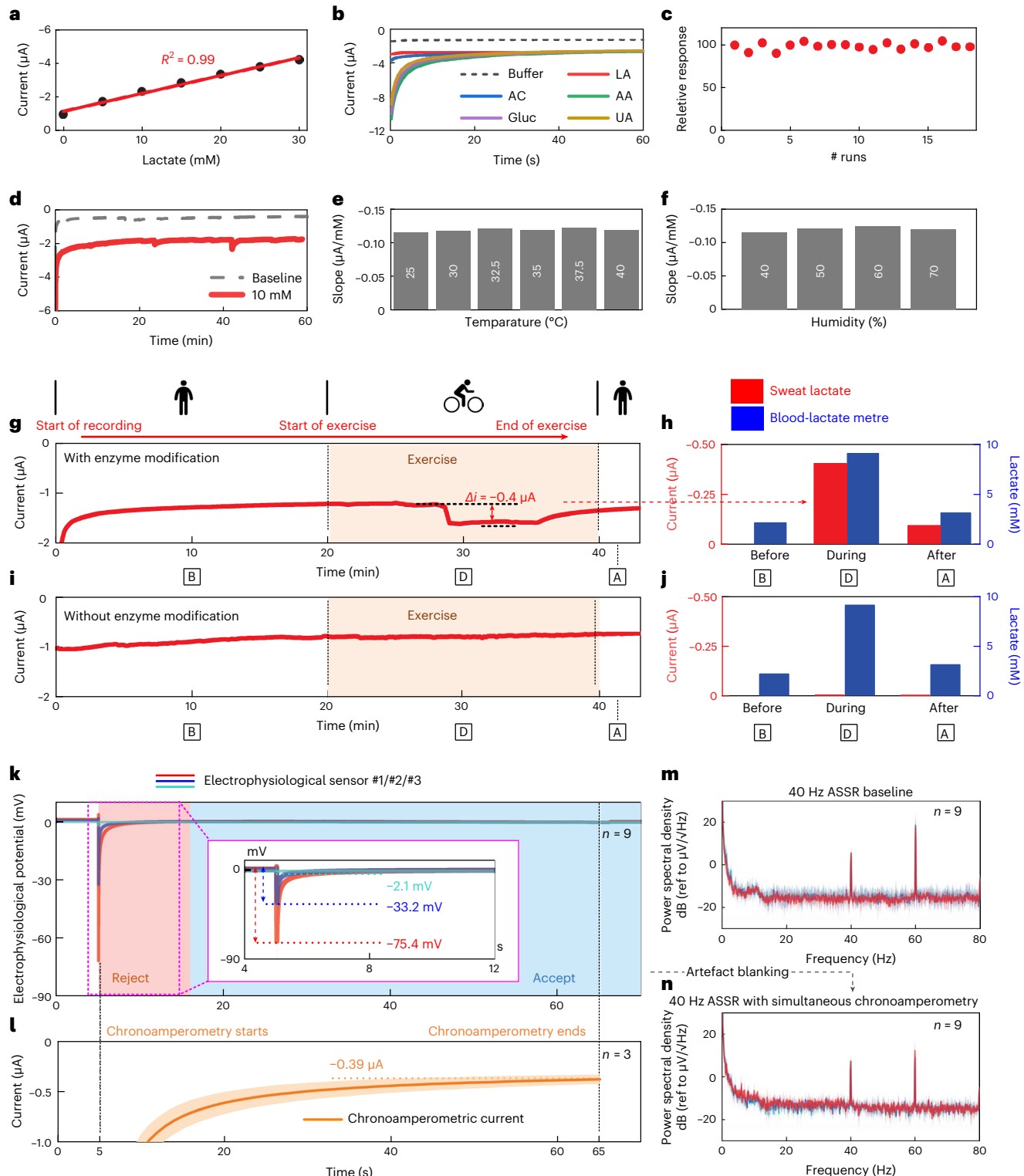

**Fig. 3 | Characterization of the in-ear electrochemical lactate sensing capability and integrated co-sensing capability. a–f**, Lactate sensing characterization. **a**, In vitro lactate sensing calibration with 5 mM lactate additions. **b**, Lactate sensing selectivity test. **c**, Lactate sensing stability results after 18 repetitive runs under a fixed concentration of 2 mM lactate. **d**, Operational stability evaluation by extended scan at 10 mM lactate for ~1 hr. **e**, Sensitivity comparison at various temperatures (25 °C, 30 °C, 32.5 °C, 35 °C, 37.5 °C and 40 °C). **f**, Sensitivity comparison at various relative humidity levels (40%, 50%, 60% and 70%). **g–j**, Representative controlled on-body lactate sensing CA recording with (**g**) or without (**i**) enzyme modification on the electrochemical electrodes, compared with simultaneous blood lactate metre reference recordings (**h,j**) taken at 10 min, 30 min and 40 min, respectively.

**k–n**, Characterization of co-sensing crosstalk on three ears from two participants. **k**, Waveform of three 70-s 40-Hz ASSR measurements with (**l**) concurrent 60-s CA measurements (synchronous with EEG measurement, starting from $t = 5$ s and ending at $t = 65$ s). **m**, 40-Hz ASSR PSD baseline recordings without concurrent CA recording, compared with (**n**) same measurements with concurrent CA recording and artefact blanking (Supplementary Text Note 4). The total number of measurements taken from each individual electrophysiological or electrochemical sensor per experiment is denoted as index $n$ in each panel. For each electrophysiological and electrochemical sensor, solid lines and shaded bands represent the participant-averaged mean ± s.d. of the measured time series and PSD.

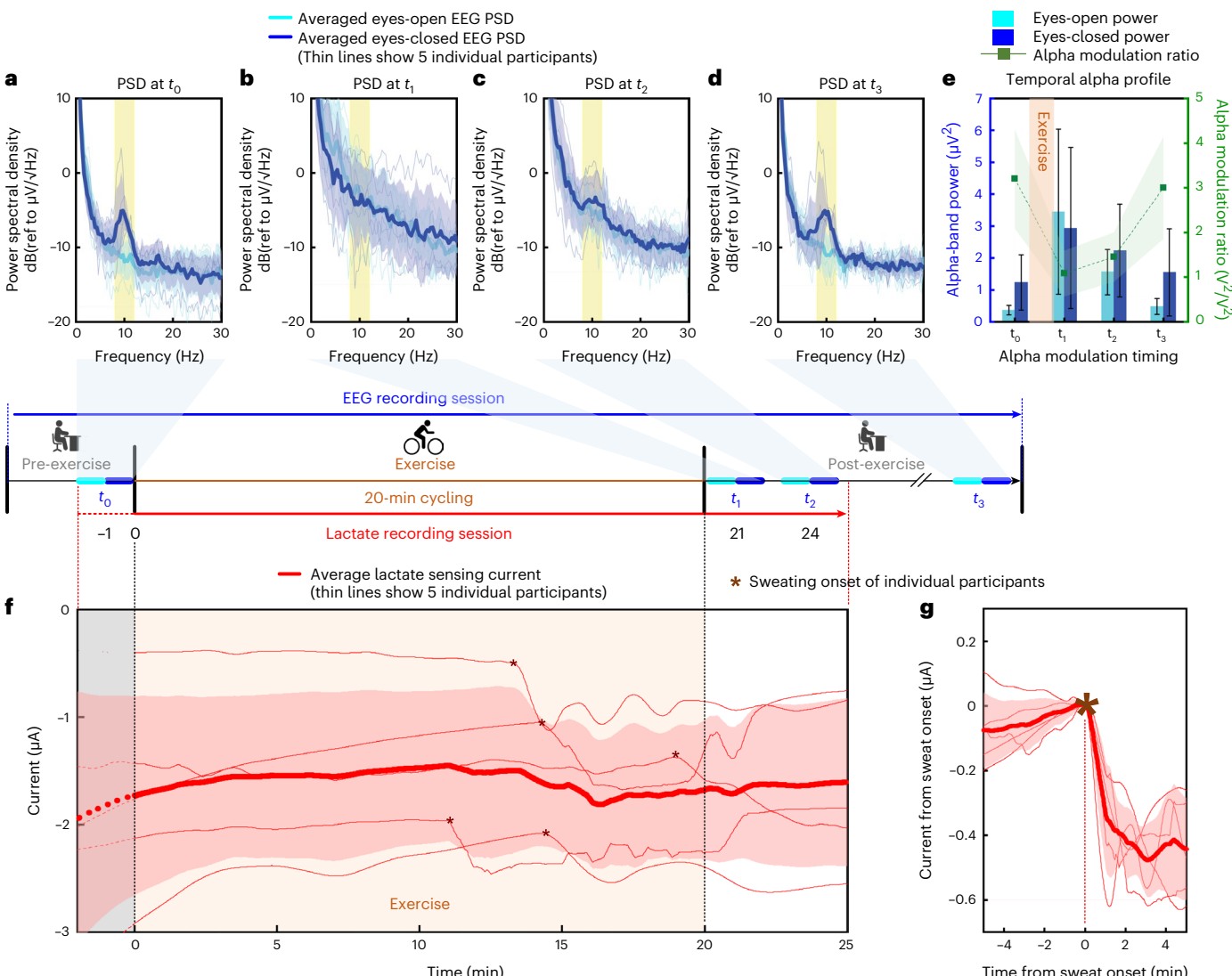

**Fig. 4 | EEG and lactate integrated sensing across five participants before, during and after exercise.** A timeline corresponding to the experimental procedure is shown in the middle. **a**–**d**, Alpha modulation power spectral density at timing (**a**) $t_0$ (pre-exercise), (**b**) $t_1$ (post-exercise-immediate), (**c**) $t_2$ (post-exercise-after 3 min) and (**d**) $t_3$ (post-exercise-relaxed). **e**, Temporal profile of alpha modulation, showing participant-averaged eyes open/closed EEG power in the alpha band (8–12 Hz), and alpha modulation ratios at times $t_0$, $t_1$, $t_2$ and $t_3$. **f**, Recording of the lactate chronoamperometry current throughout the exercise.

Asterisks denote onset of sweating, the timing of which varied across individual participants. Recordings before $t = 0$ min were for stabilization of current and not used in the analysis. **g**, Ten-min lactate CA current centred around dynamically aligned onset of sweating across all participants. Thick solid lines and shaded bands indicate the mean ± s.d. of the 5 participants' measured data (light and dark blue colour for alpha modulation PSD, red colour for lactate sensing current). All individual participant data are overlayed on the average results as thin lines and are further shown in Extended Data Fig. 10.

can be reliably performed concurrently in the ear. The comparable ASSR SNRs also indicated that the design of the in-ear sensors preserved the integrity of the earphone's other features, including its audio quality. A detailed analysis of co-sensing crosstalk can be found in Supplementary Note 4.

Previous research has found elevated broad EEG frequency bands including theta (4–8 Hz), alpha (8–12 Hz) and beta (13–30 Hz) during and right after acute exercise, driven by changes in peripheral physiology and not only within the brain itself, and returning to pre-exercise baseline resting-state EEG levels approximately 10 min after exercise[43,44]. In this combined sensing experiment, five healthy participants were instructed to conduct a 20-min cycling exercise, along with one pre-exercise alpha modulation and three post-exercise alpha modulation measurements at four timings: $t_0$ (pre-exercise), $t_1$ (post-exercise-immediate), $t_2$ (post-exercise-after 3 min) and $t_3$ (post-exercise-relaxed) shown in the timeline of Fig. 4. Concurrent

EEG and lactate sensing were conducted throughout the exercise experiment. Pre- and post-exercise alpha modulation analysis and brain-state classifications were conducted to characterize the participants' brain-state variations throughout the entire session. Figure 4a–d demonstrated the participant-averaged alpha modulation results at the four times ($t_0$ through $t_3$). Extended Data Fig. 10 shows pre-exercise baseline and post-exercise alpha modulation spectra of all individual participants. Consistent with previous observations[43,44], the EEG PSD transitions from the pre-exercise baseline levels (Fig. 4a,e) to elevated levels across the entire theta, alpha and beta bands after the exercise (Fig. 4b,e), finally returning to pre-exercise baseline levels (Fig. 4c–e). These EEG PSD variations in brain activity were accompanied by concurrent changes in other body physiological indicators, including heart rate, ventilation and breathing rate. Returning to the resting state ~10 min after the exercise, all participants showed comparable pre- and post-alpha modulation ratios (pre-exercise $R_{AM} = 3.20 \pm 1.10$ (V²/V²)

**Table 1 | Accuracy of FBCSP-based brain-state classification across five participants' pre- and post-exercise in-ear EEG data**

| Frequency bands (Hz) | Accuracy (%) | | | |
|---|---|---|---|---|
| | Eyes open | | Eyes closed | |
| | Fully relaxed | Immediately post-exercise | Fully relaxed | Immediately post-exercise |
| 4–28 (3 bands) | 64.06 | 84.40 | 63.25 | 88.45 |
| 4–36 (4 bands) | 63.37 | 86.45 | 64.11 | 90.34 |
| 4–44 (5 bands) | 63.94 | 89.77 | 66.11 | 91.48 |
| 12–44 (4 bands) | 62.74 | 89.77 | 65.31 | 90.91 |
| 20–44 (3 bands) | 63.25 | 89.71 | 66.06 | 90.11 |

Each of the participants' brain-state classification data are given in Supplementary Table 2.

vs post-exercise-relaxed $R_{AM}$ = 3.00 ± 1.14 ($V^2/V^2$)) and re-emerging baseline alpha-band brain activity when the participants were fully rested (Fig. 4d). In addition, filter-bank-based common-spatial-pattern (FBCSP) analysis[45,46], which combines spatially diverse recordings across channels and extracts EEG-related features across different frequency bands, was conducted to classify brain states before and after exercise. For every participant, both the post-exercise-immediate brain state and post-exercise-relaxed brain state were classified with respect to the pre-exercise brain state as the baseline.

Table 1 shows that the post-exercise-immediate brain state demonstrated a considerably higher prediction accuracy (on average, eyes open: 88.02%, eyes closed: 90.26%) than the post-exercise-relaxed state (on average, eyes open: 63.47%, eyes closed: 64.97%), confirming clear differences among all participants' brain states before and immediately after exercise, before returning to resting conditions afterwards. Classification accuracy varied when more frequency bands were used in the FBCSP pipeline, indicating that the recorded signals possess some discriminative characteristics in both the fast waves (beta, gamma) and the mid-band slow waves (theta, alpha).

Lactate concentration was recorded to show the secretion of sweat and the intensity of physical activity during the exercise. Figure 4e shows the grand average of the 25-min lactate sensing current throughout the exercise session (with individual participants' lactate sensing current given in Extended Data Fig. 10c,f,i,l,o). The initial stage of the experiment showed low and stable current values, reflecting the absence of lactate in sweat. In contrast, the exercise-induced change in the current profile was measured ~12 min after exercise onset, showing a −0.47 ± 0.10 μA incremental current Δ*i* on average (for the five participants: −0.58 μA, −0.34 μA, −0.41 μA, −0.42 μA and −0.58 μA, respectively). Across participants, the sweat lactate CA current remained for a variable duration, extending beyond exercise due to post-exercise sweat residue before recovering at rest after exercise. An extended 60-min recording of the temporal profile of the lactate current is shown in Supplementary Fig. 11. Although different participants showed different timings of sweat onset, dynamic alignment of the onset timing and level of the recorded sweat lactate current across participants in Fig. 4f shows a clear and consistent current increment pattern immediately upon sweat onset.

## Outlook

We have reported results from fully in-ear integrated sensors for monitoring brain-state and dynamic lactate-concentration changes. The continuous and simultaneous sensing of both modalities was achieved after rejecting the onset transient artefact of the chronoamperometric driving potential. Overall, the sensors displayed an observable degree of changes after exercise compared with the pre-exercise baseline. This work extends previous in-ear systems (Supplementary Table 1) by demonstrating integrated brain-state and dynamic chemical monitoring

in one fully integrated, user-generic device completely in the ear. The proof-of-principle demonstration presented here used off-the-shelf data-acquisition systems not optimized for power and size, but rather for the dependable performance required to characterize the sensors. Further advances in sensor–electronics integration with our low-power, low-noise analogue front-end signal-amplification, filtering and acquisition integrated-circuit designs[47,48], as well as clinical validation across a large population pool, promise to bring about an abundance of wearable diagnostic and therapeutic applications.

## Methods

### Materials and reagents

Ecoflex 00–30 was purchased from Smooth-on. TPU film was purchased from Lubrizol. Silver flakes, SEBS, toluene, Prussian blue (soluble), chitosan, acetic acid, potassium hydroxide (KOH), PVA (MW ~89,000), phosphate buffer solution (PBS) (1 M, pH 7.4), uric acid, l-lactic acid, D(+)-glucose, acetaminophen, ascorbic acid, sucrose, sodium chloride, bovine serum albumin (BSA) and potassium chloride were purchased from Sigma-Aldrich. Graphite powder was purchased from Acros Organics. Super-P carbon black was obtained from MTI. Silver conductive epoxy adhesive was purchased from MG Chemicals. LOx (activity 101 U mg⁻¹) was purchased from Toyobo. Mould release spray (Smooth-on) was purchased from Amazon.

### Sensor design and fabrication

**Sweat-gland mapping.** Locations where the sweat secretion was higher after physical activities were evaluated by using custom silicon pieces. Extended Data Fig. 1 demonstrates the preparation of the custom silicon pieces, which were modified by incrusting filter paper traces (2 × 4 mm). A small section (2 × 1 mm) of the filter paper was modified with edible blue dye. As shown in Extended Data Fig. 1f, the area modified with blue was exposed to the outside (skin), while the un-modified area was exposed to the inside of the silicon piece. As shown in Supplementary Fig. 1, by performing individual experiments on three participants, it was observed that the areas with the highest sweat volumes were found towards the tragus area of the ear channel. The geometry of the in-ear electrochemical sensors was then designed to match such sweat secretion distribution.

**Sensor fabrication, electrode modification and earphone assembly.** The in-ear integrated sensors were fabricated on the basis of the layer-by-layer screen printing method using a semi-automatic MPM-SPM printer (Speedline Technologies) and custom 10 × 10-inch stainless-steel stencils (MetalEtch). Starting from the 150-μm-thick TPU substrate, stretchable Ag interconnection layer, stretchable SEBS layer and stretchable PB electrochemical electrodes were sequentially printed and cured. After the printing of a layer, the patch element was cured in a hot-air oven for 10 min each at 80 °C, 80 °C and 60 °C, respectively, which stabilizes the printed polymeric materials and evaporates out the toluene traces from the thin layer. 3D stretchable Ag electrophysiological electrodes were then built using a 3D-printed 750-μm-thick tough polylactic acid mould (S5, Ultimaker). The sensor was then bonded to a flexible printed circuit board (PCB) (PCBWay) using silver liquid solder for electrical connections. The sensor, along with the flexible PCB, was then assembled onto a generic in-ear earphone and secured using the pre-applied double-sided adhesive (3M medical 1509) as well as a silicone earphone hook. Details of the ink formulation, printing parameters, 3D electrophysiological electrode fabrication, bonding and assembling processes, and electrode modification method are given in Supplementary Note 1.

**3D electrophysiological electrode design.** 3D electrophysiological electrodes were built to provide tight contact between the electrophysiological electrode and the ear canal. To overcome the geometrical variations of participants' ears, the electrophysiological electrodes'

3D structure was featured with an electrode thickness that was 5 times of the TPU substrate (750 μm stretchable Ag thickness, 150 μm TPU thickness) and a 'spring-loaded backing' structure made with SEBS (Extended Data Fig. 2r). Such a structure was realized by using the thermal coefficient mismatch between the TPU substrate and the stretchable Ag ink at the curing step to create a curvature shape under the electrode. The concept of using the thermal coefficient mismatch was demonstrated with a finite element model (Supplementary Note 1), which showed a deflection of 53% of the TPU thickness (80 μm, 150 μm TPU thickness) after the curing process. This deflection created the space underneath the electrophysiological electrodes, which was then filled with elastomer material SEBS to function as the 'spring-loaded' support for the electrophysiological electrodes.

**Participant-specific tight fitting inside the ear.** To account for the anatomical variation in ear shape between participants and the geometrical mismatch between the round-arc earphone tip and the ear canal, three sizes of generic silicone earphone tips were used in the assembly of the sensors for a tight fit across different human participants during on-body experiments, with small (11.6 mm), medium (12 mm) and large (13.5 mm) diameters. The outline contour of the integrated ear sensors was customized to match the contour of the earphone and its silicone tip. Moreover, the electrophysiological stretchable Ag electrodes adopted a 3D structure (Extended Data Fig. 2q–v), whereas the three planar electrochemical electrodes were coated with a piece of PVA hydrogel (Extended Data Fig. 3c–e) to cushion the gap between the electrochemical electrodes and the ear canal, in addition to enhancing sweat collection due to its hydrophilicity and porosity. Such soft structures offered additional cushioning to accommodate the anatomical variation of the ear in addition to the existing tight fit between the earphone tip and ear canal. Lastly, the sensors were assembled to the earphone using the bottom adhesive. In this work, a silicone earphone hook was used to provide further mechanical anchoring to the auricle (Extended Data Fig. 2f). These three design factors ensured interference fit between the sensors and the ear canal as well as mechanical stability when the participant moved. More intricate designs of the earphone and sensor enclosure in flexible and stretchable form factors may provide equal or greater levels of mechanical stability with greater user comfort, conforming to various ear sizes and shapes. Likewise, the fabrication and assembly procedures employed in this work can be further mechanized for large-scale, low-cost production by using 3D-printed custom moulds, spinning machines and automated cutters.

**PVA gel characterization.** The suitability of the PVA interfacial hydrogel for skin and on-body application was evaluated. As the PVA hydrogel is in contact with the skin, its potential skin-irritant components were evaluated. Since KOH was used to prepare the sensor, there is a possibility of its retention in the hydrogel due to its spongy texture. To characterize the presence of KOH, we optimized the post-synthetic washing steps and confirmed the results with infrared tests, where the infrared spectra of the PVA hydrogel were recorded immediately after preparation and after subsequent washing (Supplementary Fig. 5a). The disappearance of the peak at 670 cm$^{-1}$ and fainting of the peak at 2,940 cm$^{-1}$ of the spectrum after washing was attributed to the absence of the KOH in the hydrogel. In addition, the PVA hydrogel was evaluated by measuring the pH at its surface before each experiment, where a neutral pH ensured and confirmed the elimination of the KOH irritant from the PVA hydrogel.

## Electrophysiological sensing

The on-body tests of the in-ear integrated sensors were conducted on healthy consenting individuals with no previous history of hearing damage, heart conditions or chronic pain and in strict compliance with the protocol approved by the Institutional Review Board of the University of California, San Diego.

**Electrode–ear impedance characterization.** The electrode–ear impedance measurements were performed with a potentiostat (PalmSens4, PalmSens) in a three-electrode setup, which characterized the impedance at the working electrode–ear interface. Here, all three electrodes were made of the same stretchable Ag material. The working electrode and reference electrode were two adjacent electrophysiological electrodes in the ear canal, and the reference electrode was the concha cymba REF electrode, as shown in Fig. 1i. The continuous impedance testing used the galvanostatic impedance spectroscopy method with an applied current range of 10 μA ($i_{ac} = 0.01 \times 10 = 100$ nA), a total duration of 120 s and a fixed frequency at 10 Hz, which is a representative EEG frequency. The EIS testing also used the galvanostatic impedance spectroscopy method with an applied current range of 100 μA ($i_{ac} = 0.01 \times 100 = 1$ μA) and a frequency range of 1 Hz–1 kHz. The galvanostatic impedance spectroscopy method strictly clamped the current level running into the body to ensure safety. Figure 2a–d show the grand average for both the continuous impedance and the EIS experiments; 6 electrophysiological electrode–ear impedance recordings across 2 participants were obtained and averaged to produce the results.

**Electrophysiological measurement system integration and on-body setup.** The electrophysiological measurement system consisted of the in-ear integrated sensors' electrophysiological electrodes and a wireless DAQ (BioRadio, Great Lakes Neurotechnologies, Bluetooth low energy). For on-body electrophysiological measurements, the electrophysiological electrodes on the in-ear integrated sensors were connected to the input channels of the DAQ through the bonded flexible PCB (Extended Data Fig. 2y) and connector cables (Supplementary Fig. 2). During measurements, the participants wore the in-ear integrated sensors assembled on earphones. The flexible PCB was adhered to the mastoid and the hindneck. The DAQ was attached to the collar at the participant's back (Extended Data Fig. 7a). According to the continuous electrode–ear impedance measurement results, the participant was instructed to wear the in-ear integrated sensors and the DAQ for more than 2 min to stabilize the electrode–ear interface and reach a magnitude of less than 1 MΩ at 10 Hz. The DAQ employed the following configurations: sampling rate, 500 Hz per channel (two ears, 6-channel input) or 1 kHz per channel (1 ear, 3-channel input); measurement mode, single-ended; resolution, 1 μV; input range, ±187 mV. Live streaming raw EEG data (Unit V) were transmitted via Bluetooth to the host computer and were saved with the DAQ-bundled software (BioCapture, Great Lakes Neurotechnologies). The raw EEG data were then exported as .csv files for processing.

The DAQ system in this work had an integrated 24-bit high-precision TI ADS1299 acquisition chip. The ADS1299 chip was pre-programmed to have the lead-off detection functionality, which injected a current of 6 nA at minimum and 31.2 Hz alternating current signal to the body for impedance measurement purposes[49]. The 6 nA injected resulted in ~30 μV of artefact signal at 31.2 Hz, which is within the frequency range of the EEG signals. For all the EEG spectral analyses of alpha modulation and the ASSR measurements, the calculated 31.2 Hz data point was rejected from the EEG PSD spectra.

**Electrophysiological electrode DC offset characterization.** No skin cleaning preparation was conducted before the electrophysiological EDO experiments. The EDO experiment was conducted on 2 participants' 4 ears; each electrophysiological electrode took the EDO recording 8 times, with a time separation of ~5 min between recordings. Each EDO recording measured the potential on one electrophysiological electrode against the REF electrode for 10 s. The EDO value of one recording 'edo ($s$)' was the average potential over the 10-s duration, calculated using $edo(s) = \frac{1}{10,000} \sum_{n=1}^{10,000} v_s(n)$. Here, $n$ is the time sequence: $n = 1$–10,000 since the sampling rate of the 10-s EDO recordings was

1 kHz. $s$ is the recording index: $s = 1–96$ since 96 EDO recordings were taken. The 96 EDO waveforms were averaged in Fig. 2e and the EDO values were normally distributed in Fig. 2f. Here, the normal distribution fit returns the probability density function of the 96 EDO values with the mean $\mu = \frac{1}{96} \sum_{s=1}^{96} \text{edo}(s)$ and the standard deviation $\sigma = \sqrt{\frac{1}{96} \sum_{s=1}^{96} (\text{edo}(s) - \mu)^2}$ as follows:

$$f(\text{edo}(x), \mu, \sigma) = \frac{1}{\sigma\sqrt{2\pi}} e^{\frac{-(x-\mu)^2}{2\sigma^2}} \tag{1}$$

**Alpha modulation measurement protocol and analyses.** The alpha modulation measurements employed the following protocol: the alpha baseline measurement was conducted when the participant stayed relaxed and kept eyes open for 1 min. The alpha modulation measurement was taken when the participant stayed relaxed and kept eyes closed for 1 min. By pressing the triggering button on the DAQ, the timings for the start and end of the alpha baseline or alpha modulation measurements were recorded as event markers along with the EEG data stream. Both time-frequency analyses and spectral analyses were conducted. For the time-frequency analyses, the participant conducted a 4-min experiment with the following sequence: eyes open, eyes closed, eyes open, eyes closed. The raw EEG streaming data were then used to generate a spectrogram with the following parameters in Matlab (MathWorks): window segment length, 3,000 (3 s); overlapped points, no overlap; sampling frequency, 1 kHz. For the spectral analyses, the participant first conducted a 1-min eyes-open alpha baseline measurement and then conducted a 1-min eyes-closed alpha modulation measurement. The raw EEG streaming data were then used to generate the power spectral density with the Matlab EEGLAB toolbox (spectopo function) using the following parameters: window segment length, 10 s; window overlapping, 0.5 s. Here, the PSD was calculated using a time-domain averaging method. Raw EEG data were first segmented and the PSD was the averaged results of individual window segments, which helped smooth the noise. The alpha modulation ratio $R_{AM}$ characterized the modulated alpha-band power relative to the eyes-open baseline over the 8–12 Hz frequency alpha band. Here, $R_{AM}$ was defined as the ratio of the 1-min eyes-closed EEG power $P(\text{Alphaband}_{\text{Eyes open}})$ and the 1-min eyes-open EEG power $P(\text{Alphaband}_{\text{Eyes closed}})$:

$$R_{AM} = \frac{P(\text{Alphaband}_{\text{Eyes closed}})}{P(\text{Alphaband}_{\text{Eyes open}})} \tag{2}$$

$$P(\text{Alphaband}) = \sum_{n=\frac{8\,\text{Hz}}{\Delta f}+1}^{\frac{12\,\text{Hz}}{\Delta f}} \text{PSD}(n)\,\Delta f \tag{3}$$

As an overall evaluation, the alpha modulation results were characterized by statistically averaging PSDs from 4 participants' 12 electrophysiological sensors in total, as shown in Fig. 2h.

**ASSR measurement protocol and analyses.** ASSR is an important auditory brain signal pattern and has found applications in hearing threshold estimation and brain–computer interfaces. ASSR can be evoked by presenting ASSR sound stimuli to the participants, which commonly takes the form of amplitude-modulated white noise signals[24]. The ASSR stimuli were generated in Matlab with the following processing steps (Supplementary Fig. 4): uniformly distributed Gaussian white noise was first generated and then convolved with a set of sinusoids used in the ASSR experiments (25 Hz, 40 Hz, 55 Hz, 70 Hz), producing an amplitude-modulated white noise signal. Superimposed waveforms were separated from the modulated noise signal into frequency-specific amplitude modulations. A Blackman envelope was applied to the separated waveforms, producing an appropriately

shaped and timed final ASSR stimulus. The ASSR stimuli were played on the Bluetooth-connected earphones, integrated with the in-ear integrated sensors. The sound pressure level (SPL) of the ASSR stimuli was adjusted to 75 dB by putting the earphone tip in proximity to a high-sensitivity electric sound gauge. The ASSR measurement employed the following protocol: the participant stayed relaxed, kept eyes closed and was played one of the ASSR stimuli (25 Hz, 40 Hz, 55 Hz, 70 Hz) at both ears for 1 min. The raw EEG streaming data taken during the ASSR measurement were then used to generate the PSD with the Matlab EEGLAB toolbox (spectopo function) using the following parameters: window segment length, 10 s; window overlapping, no overlap. The ASSR SNR was calculated as the ratio of the power at the ASSR frequency $f_{\text{ASSR}}$ to the averaged power of noise from $f_{\text{ASSR}} - 5\,\text{Hz}$ to $f_{\text{ASSR}} + 5\,\text{Hz}$, excluding $f_{\text{ASSR}}$ power:

$$\text{SNR}_{f_{\text{ASSR}}} = \frac{P(f_{\text{ASSR}})}{P_{\text{Avg}}(f_{\text{ASSR}} - 5\,\text{Hz to } f_{\text{ASSR}} + 5\,\text{Hz, excluding } f_{\text{ASSR}})} \tag{4}$$

$$P(f_{\text{ASSR}}) = \text{PSD}\left(\frac{f_{\text{ASSR}}}{\Delta f}\right)$$

$$P_{\text{Avg}}(f_{\text{ASSR}} - 5\,\text{Hz to } f_{\text{ASSR}} + 5\,\text{Hz, excluding } f_{\text{ASSR}}) =$$

$$\frac{1}{10\,\text{Hz}-\Delta f}\left(\sum_{n=\frac{f_{\text{ASSR}}-5\,\text{Hz}}{\Delta f}+1}^{\frac{f_{\text{ASSR}}}{\Delta f}-1} \text{PSD}(n)\,\Delta f + \sum_{n=\frac{f_{\text{ASSR}}}{\Delta f}+1}^{\frac{f_{\text{ASSR}}+5\,\text{Hz}}{\Delta f}} \text{PSD}(n)\,\Delta f\right) \tag{5}$$

As an overall evaluation, the ASSR results were characterized by statistically averaging PSDs from 4 participants' 12 electrophysiological sensors in total, as shown in Fig. 2i–l.

**Eye movement measurement protocol and analyses.** Eye movement measurements were taken with in-ear integrated sensors in both ears. Here, the REF electrode of one ear was used as the common REF for the 6 electrophysiological electrodes in both ears. Ipsilateral referencing refers to measurements taken from the electrophysiological electrodes at the REF electrode's side. In contrast, contralateral referencing refers to measurements taken from the electrophysiological electrodes at the other ear. During the eye movement measurements, the participant was instructed to sit relaxed in front of a monitor and perform the following eye movements with a timer hint on the screen: glancing up, glancing down, glancing left, glancing right, normal eye blinking and heavy eye blinking. The corresponding 6-channel, synchronous raw EEG data stream was processed on the host computer. The DC offset of the EEG data stream was removed by subtracting the mean of the time series. The power line interference was removed by applying a 60 Hz notch filter. For individual eye movement characterization, the peaks of the corresponding eye movement signals were detected and centred in a 1-s window segment. The signal amplitude of each eye movement was characterized by calculating the absolute value of the difference between the averaged peak potential over the 200-ms interval ($V_{400-600\text{ms}_{\text{avg}}}$) centred within the 1-s window segment, and the averaged baseline potential over the leading and trailing 100-ms intervals ($V_{0-100\text{ms},900-1000\text{ms}_{\text{avg}}}$) of the 1-s window segment:

$$A = \left| V_{400-600\text{ms}_{\text{avg}}} - V_{0-100\text{ms},900-1000\text{ms}_{\text{avg}}} \right| \tag{6}$$

As an overall evaluation, the eye movement measurement results were characterized by statistically averaging EEG time-series data from 2 participants' both ears (6 sensors per participant, 12 electrophysiological sensors in total) as shown in Fig. 2m,n and Extended Data Fig. 5.

### Electrochemical lactate sensing

The electrochemical characterization of the lactate biosensor was performed and recorded using a PalmSens4 potentiostat (Supplementary

Fig. 2) and PSTrace software (PalmSens), respectively. In all experiments using the electrochemical sensor, CA was used as the sensing method, applying a potential step of −0.2 V vs Ag (RE).

**In vitro characterization.** In standard addition, selectivity and stability tests, CA scans were applied for 60 s. A solution of PBS (0.1 M, pH 7.3) was used as the electrolyte covering the electrochemical sensor. After adding the desired concentration to the PBS aliquot, the solution was mixed 10 times using a regular pipette, followed by an incubation time of 60 s. This period was followed by the recording step using CA. The relative response (%) of each run performed in the stability test (Fig. 3c) was calculated by dividing the current of each amperogram (run $n$) by the initial amperometric scan (run 0). The resulting number was then multiplied by 100 to acquire the percentage difference. The electrochemical sensors were also characterized for stability in repeated scans and long-term scanning, variable temperature and variable humidity. For evaluating repeated scanning stability, subsequent scans were recorded after every 10 min at 10 mM of lactate for 18 times. Thereafter, the assessment in the long-term scan was performed by scanning the sensor at 0 mM and 10 mM of lactate. Further, the temperature- and humidity-dependent stability of the sensors was assessed by evaluating the dose-dependent study at various temperatures (25 °C, 30 °C, 32.5 °C, 35 °C, 37.5 °C and 40 °C) and relative humidity levels (40%, 50%, 60% and 70%), respectively. These dose-dependent studies were performed by scanning the CA after sequential addition of 5 mM lactate aliquots at four concentrations (5–20 mM).

**On-body characterization.** Before transferring the device, the ear of each volunteer was cleaned with alcohol prep pads. After cleaning, the area was left to dry for 2 min. Meanwhile, a 0.5 × 0.5-cm piece of PVA hydrogel was placed on top of the electrochemical sensor. Next, the hydrogel layer was pressed for 30 s to ensure good contact between the gel and the sensor. Next, the device was transferred inside the ear of the individual. The test started by continuously applying CA for 45 min. In the first stage of the experiment, 3 volunteers were asked to stay still for 20 min to stabilize the sensors. The recorded CA current during the stabilization phase was not used in the analysis. Next, the stationary exercise was performed for 20 min at a fixed level. After concluding the last stage of the experiment, volunteers were asked to remain still for 5 min. Simultaneously, blood samples were collected 10, 30 and 43 min after starting the experiment using a blood lactate metre (NOVA Biomedical). The recorded CA currents from the electrochemical sensors were temporally averaged over a 50-point sliding rectangular window to filter out high-frequency noise due to PVA hydrogel electrochemical fluctuations at the electrochemical sensor–skin interface. The incremental current induced at sweat onset was calculated by taking the difference between the average current of the sweating session and that of the 30-s pre-sweating session, the timing of the sweat onset was variable across participants.

**Co-sensing crosstalk characterization**
The crosstalk between the electrophysiological and electrochemical signals was analysed on-body by monitoring the changes in one signal while the other signal was generated. Co-sensing experiments combined the experimental setups of both the electrophysiological and electrochemical measurements. Specifically, the participants wore one of the in-ear integrated sensors in one ear. Three electrophysiological electrodes were in direct contact with the ear canal and were connected to the DAQ, and 3 electrochemical electrodes contacted the ear canal through a piece of PVA hydrogel and were connected to the potentiostat. Both the EEG potentiometric and lactate amperometric data streams were transmitted to the host PC wirelessly via separate Bluetooth connections.

Two aspects were considered in analysing co-sensing interference: the sweat condition and the effect of the electrochemical CA

measurement setup. Sweat concentration inside the ear was an influencing factor because it affected the conductivity of the skin, the electrode–ear interface and the PVA hydrogel. Sweating conditions were simulated with two ideal settings. First, before each co-sensing crosstalk experiment, the participant's ear was cleaned with alcohol to remove sweat residues. Then, one setting simulated the less-sweaty condition by drop casting 10 µl of deionized water to the PVA hydrogel, which diffused to the skin after insertion (Extended Data Fig. 9b). The other setting simulated the extremely sweaty condition by drop casting 10 µl of 0.1 mol $l^{-1}$ PBS to the PVA hydrogel (Extended Data Fig. 9c). Under normal conditions, the expected amount of sweat in the ear is ~20% of the 10 µl PBS solution[42] used in the sweaty setting. To demonstrate the effect of the electrochemical CA measurement setup, after considering real-life usage conditions, a typical auditory EEG paradigm ASSR was used for the EEG measurement. Participants were presented with a 40 Hz ASSR stimulus for 70 s. During the period, an EEG measurement session was taken. After 5 s from the onset of the EEG measurement, a lactate measurement session was started by applying the −0.2 V potential between the electrochemical WE and RE, resulting in a CA current between the electrochemical WE and CE. The lactate measurement session lasted for 60 s and ended 5 s before the end of the EEG measurement session. Crosstalk was characterized by statistically analysing the time series and spectra of EEG and lactate measurements across the 3 ears of 2 participants.

As shown in Extended Data Fig. 9d–j, the co-sensing crosstalk experiment results show that the potentiometric and CA measurements could operate simultaneously in the ear despite the presence of transient artefacts, which took place right after the driving voltage was applied to the electrochemical electrodes. Such artefacts are location-dependent, which can be explained by the different amounts of voltage drop across the skin given different separation distances. There are several ways to avoid or reject such artefacts. The first solution is to separate the electrophysiological and electrochemical electrodes even further, which is not ideal for in-ear applications given insufficient space. The second solution is to build insulating structures between the electrophysiological and electrochemical electrodes, as shown in previous work[8]. However, this may result in challenges due to the limited space in the ear. The third solution is to apply front-end mitigation techniques[50]. For the application in this work, the CA-induced artefact was controllable and sporadic, and the artefact blanking method was used without adding complexity to the circuit. Specifically, a portion of the EEG time series data after the onset of the lactate measurement was rejected for both time domain and frequency domain analyses to maintain the fidelity of the EEG features. A detailed discussion of the EEG spectral analyses regarding artefacts in this work can be found in Supplementary Note 4.

**Combined EEG and lactate sensing**
The experiment involved an exercise session, during which five participants experienced resting state, sweating, exciting state and return to resting state. The combined sensing experiment integrated the experimental setups of both the electrophysiological and electrochemical measurements. Throughout the experiment, every participant wore the in-ear integrated sensors in one ear, which connected to both the DAQ and the potentiostat for EEG and lactate data streaming and wireless transmission. Before transferring the device, the ear of each volunteer was cleaned with alcohol prep pads. After cleaning, the area was left to dry for 2 min. Meanwhile, a 0.5 × 0.5-cm piece of PVA hydrogel was placed on top of the electrochemical biosensor. Next, the hydrogel layer was pressed for 30 s to ensure good contact between the gel and the sensor. Afterwards, the device was transferred inside the ear of the individual.

The combined sensing experiment used the following protocol as shown in the timeline in Fig. 4: The EEG recording first started and the participant then remained seated at a table for more than 2 min to relax

thoroughly. Following that, the 30-min lactate sensing with CA began. A 2-min pre-exercise alpha modulation experiment (consisting of a 1-min eyes-open alpha baseline experiment and a 1-min eyes-closed alpha modulation experiment) was conducted and time-stamped sequentially by pressing the trigger buttons on the DAQ (denoted as $t_0$). At $t = 0$ min, the participant moved to a nearby stationary bike and started a 20-min cycling session (room temperature). At $t = 20$ min, the participant stopped cycling, moved back to the table and sat down. Immediately after the participant sat down, the first 2-min post-exercise alpha modulation experiment was conducted and time-stamped (denoted as $t_1$). After 2 min, the second 2-min post-exercise alpha modulation experiment was conducted and time-stamped (denoted as $t_2$). When the participant returned to normal breathing rhythm and self-reported the relaxing state, the third 2-min pre-exercise alpha modulation measurement was conducted and time-stamped (denoted as $t_3$). The third alpha modulation measurement was repeated to ensure that participants reached consistent fully relaxed state. All the post-exercise alpha modulation measurements were conducted using the same instrumental setup and followed the same procedures as the pre-exercise alpha modulation measurements. During the exercise session, motion artefacts were present for the EEG recording, while all alpha modulation data were collected when the participant remained motionless, avoiding the introduction of motion artefacts. Signal processing methods including automatic subspace reconstruction may be applied to reduce the motion artefacts recorded from the electrophysiological sensors. Details of the automatic subspace reconstruction algorithm used for in-ear sensors can be found in Supplementary Note 3. To characterize the brain states before and after exercise, Extended Data Fig. 10b,e,h,k,n and d,g,j,m,p show the measured pre-exercise and post-exercise alpha modulation results from each participant, respectively, on the basis of which the alpha modulation ratios were calculated. Figure 4a–d show the calculation of participant-averaged alpha modulation results at four different timings: $t_0$ (pre-exercise), $t_1$ (post-exercise-immediate), $t_2$ (post-exercise-after 3 min) and $t_3$ (post-exercise-relaxed), respectively. Figure 4d summarizes the results of participant-averaged EEG alpha band (8–12 Hz) power and participant-averaged alpha modulation ratios, both of which were calculated from each participants' measured PSD in the alpha band and then statistically averaged. For sweat monitoring, Extended Data Fig. 10c,f,i,l,o show the measured lactate sensing currents from each participant, which again were temporally averaged over a 50-point rectangular sliding window to filter out high-frequency noise. Figure 4f summarizes the results of participant-averaged lactate sensing current. Figure 4g further characterizes the change in lactate sensing current by dynamically aligning the current waveform at the onset of measured sweating from each participant.

**FBCSP method for EEG feature classification.** The EEG recording was further used to classify the pre- and post-exercise brain states from a broader frequency scope, apart from the alpha modulation, which only focused on the 8–12-Hz frequency band and resting states. The FBCSP approach has been widely deployed as a classification algorithm for scalp-EEG motor imagery tasks[45,46]. This work used the FBCSP method to extract features and classify physiological and brain-state changes acquired via in-ear EEG sensing.

FBCSP used a two-stage data processing pipeline for two-class feature extraction and classification (Supplementary Fig. 12). The three participants' continuous EEG data before, during and after the exercise were used as inputs to the pipeline. In both cases, the eyes-open-session data and eyes-closed-session data were used to investigate the brain-state changes. The first stage of the pipeline consisted of multiple bandpass filters (4–12 Hz, 12–20 Hz, 20–28 Hz, 28–36 Hz and 36–44 Hz). These bandpass-filtered signals went through a common-spatial-filter transformation where a projection matrix was calculated, maximizing the variance of class-specific samples in a spatial dimension and maximizing the other class variance in a different spatial dimension.

In this stage, we also extracted the reduced-dimension CSP features for all windows and generated the final CSP feature vector and associated their class labels. Further, a mutual information-based feature extraction technique selected the most discriminative features for classification. The final stage was a support-vector-machine classifier, which calculated the decision boundary for classification.

Combining the FBCSP, an in-ear EEG data processing pipeline consisting of signal processing and machine learning methods was used to determine the participants' physiological states accurately. In this work, the same datasets from all participants' alpha modulation sessions at $t_0$, $t_1$, $t_2$ and $t_3$ in Fig. 4a–d were used. Here, the classification of the pre-exercise brain state at $t_0$ and post-exercise-immediate state at $t_1$ was chosen as one task, while the classification of the pre-exercise state at $t_0$ and post-exercise-relaxed state at $t_3$ was chosen as the other task. Table 1 shows the averaged accuracy of brain-state classification, with each individual participant's classification results shown in Supplementary Table 2.

### Reporting summary

Further information on research design is available in the Nature Portfolio Reporting Summary linked to this article.

## Data availability

All data supporting the results in this study are available within the paper and its Supplementary Information. Source data for the figures are provided with this paper, and are available in figshare at https://doi.org/10.6084/m9.figshare.22829051.

## Code availability

Custom codes for electrophysiological signal analysis, for the automatic subspace reconstruction (ASR) algorithm and for the filter-bank-based common-spatial-pattern (FBCSP) method are available at https://doi.org/10.1038/zenodo.8193117.

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

## Acknowledgements

We thank the UCSD Center of Wearable Sensors (CWS) for providing generous support to this research. E.D.l.P. acknowledges support from a UC MEXUS–CONACYT collaborative fellowship (2017–2022).

## Author contributions

Y.X., E.D.l.P., A.P., J.R.S., P.M., S.X., J.W. and G.C. conceived the idea. Y.X., E.D.l.P., A.P., J.R.S. and M. Lin contributed to the design of the in-ear integrated sensors. Y.X., E.D.l.P., A.P., J.R.S., S. Deiss and B.L.W. contributed to the interface instrumentation design. Y.X. designed and simulated the 3D electrophysiological electrodes, and designed the flexible PCB. N.T., Y.X., E.D.l.P., K.M., J.R.S., M. Lin, S. Dua, L.Y. and W.C. fabricated the sensors. Y.X., A.P., M. Lee and G.H. designed and carried out the electrophysiological characterization experiments. E.D.l.P., K.M. and J.R.S. designed and carried out the electrochemical characterization experiments. Y.X., E.D.l.P., K.M., N.T. and W.C. designed and carried out the co-sensing experiments. Y.X., A.P. and M. Lee analysed the electrophysiological data and developed the code for electrophysiological analysis. A.P. developed the code for ASSR stimuli. A.U. and Y.X. conducted the EEG motion-artefact analysis. G.H. and Y.X. performed the FBCSP analysis. E.D.l.P., K.M., J.R.S. and Y.X. analysed the electrochemical data. Y.X. and E.D.l.P. analysed the co-sensing crosstalk data. Y.X., E.D.l.P. and J.R.S analysed the combined EEG and lactate-sensing data. Y.X. and E.D.l.P. drafted the manuscript. A.P., K.M., J.R.S., M. Lee, A.U. and G.H. contributed to the schematics and photographs. G.C, J.W, S.X. and P.M. planned and guided the experiments, analysed data, participated in figure design and in the writing of the manuscript, and provided guidance and infrastructure for the projects. All authors contributed to discussing the data and commented on the manuscript.

## Competing interests

The authors declare no competing interests.

## Additional information

**Extended data** is available for this paper at https://doi.org/10.1038/s41551-023-01095-1.

**Correspondence and requests for materials** should be addressed to Patrick Mercier, Sheng Xu, Joseph Wang or Gert Cauwenberghs.

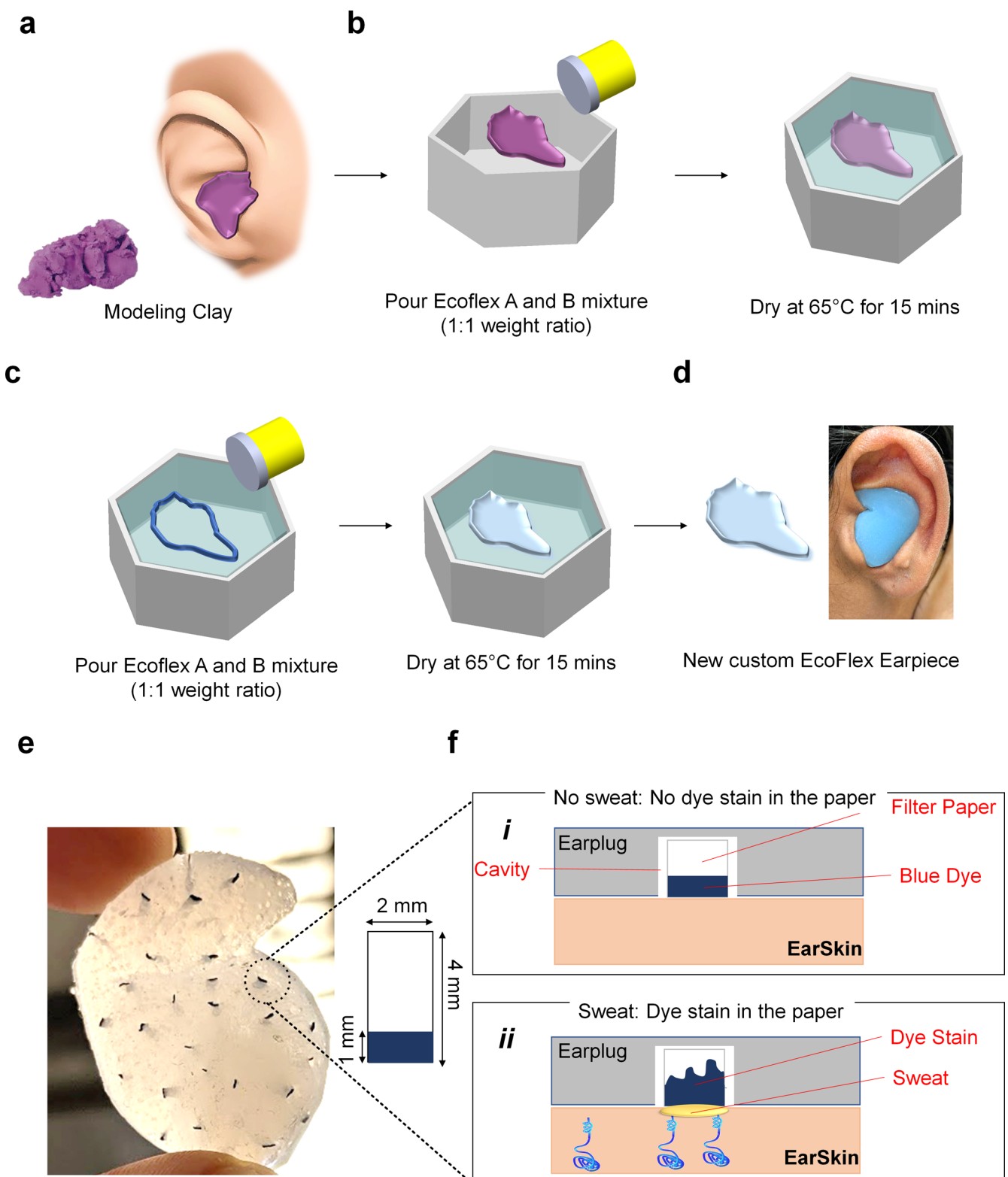

**a.** Modeling Clay

**b.** Pour Ecoflex A and B mixture (1:1 weight ratio) → Dry at 65°C for 15 mins

**c.** Pour Ecoflex A and B mixture (1:1 weight ratio) → Dry at 65°C for 15 mins

**d.** New custom EcoFlex Earpiece

**e.**

2 mm / 4 mm / 1 mm

**f.**

*i* No sweat: No dye stain in the paper — Earplug, Cavity, Filter Paper, Blue Dye, EarSkin

*ii* Sweat: Dye stain in the paper — Earplug, Dye Stain, Sweat, EarSkin

**Extended Data Fig. 1 | Fabrication of the custom Ecoflex earpiece used for sweat mapping. a.** The modeling clay was inserted and pressed gently inside the ear to obtain the inner shape of the participants. Next, the clay with the shape of the inner ear was removed from the participants. **b.** After removal, the clay with the inner ear canal shape was sprayed with mold release. Afterward, the piece was placed inside a receptacle in which a mixture of Ecoflex mixture was poured. Next, the receptacle and its content were dried at 65 °C for 15 minutes. **c.** After drying, the clay was removed from the receptacle, leaving an Ecoflex mold in the shape of the clay. The clay's void space was first sprayed with mold release and then filled with a mixture of Ecoflex and dried following the same conditions used in the previous step. **d.** The resulting Ecoflex earpiece was removed from the mold to use in further experiments. **e**–**f.** Sweat mapping using a costume Ecoflex earpiece. (**e**) With the help of a tweezer, small cavities were made in the resulting Ecoflex earpiece. Next, pieces of filter paper (2×4 mm) were modified with an edible blue dye covering a certain area (2×1 mm). These pieces were inserted inside the cavities of the earpiece, exposing the area with blue dye to the ear skin. (**f**) Three participants ($n = 3$) wore the resulting earpiece to perform 30 minutes of stationary cycling at a fixed level. Before exercise, no dye stain was observed due to the absence of sweat (*i*). After concluding the 30 minutes of exercise, dye spread across the filter paper was observed, indicating areas of higher sweat secretion (*ii*).

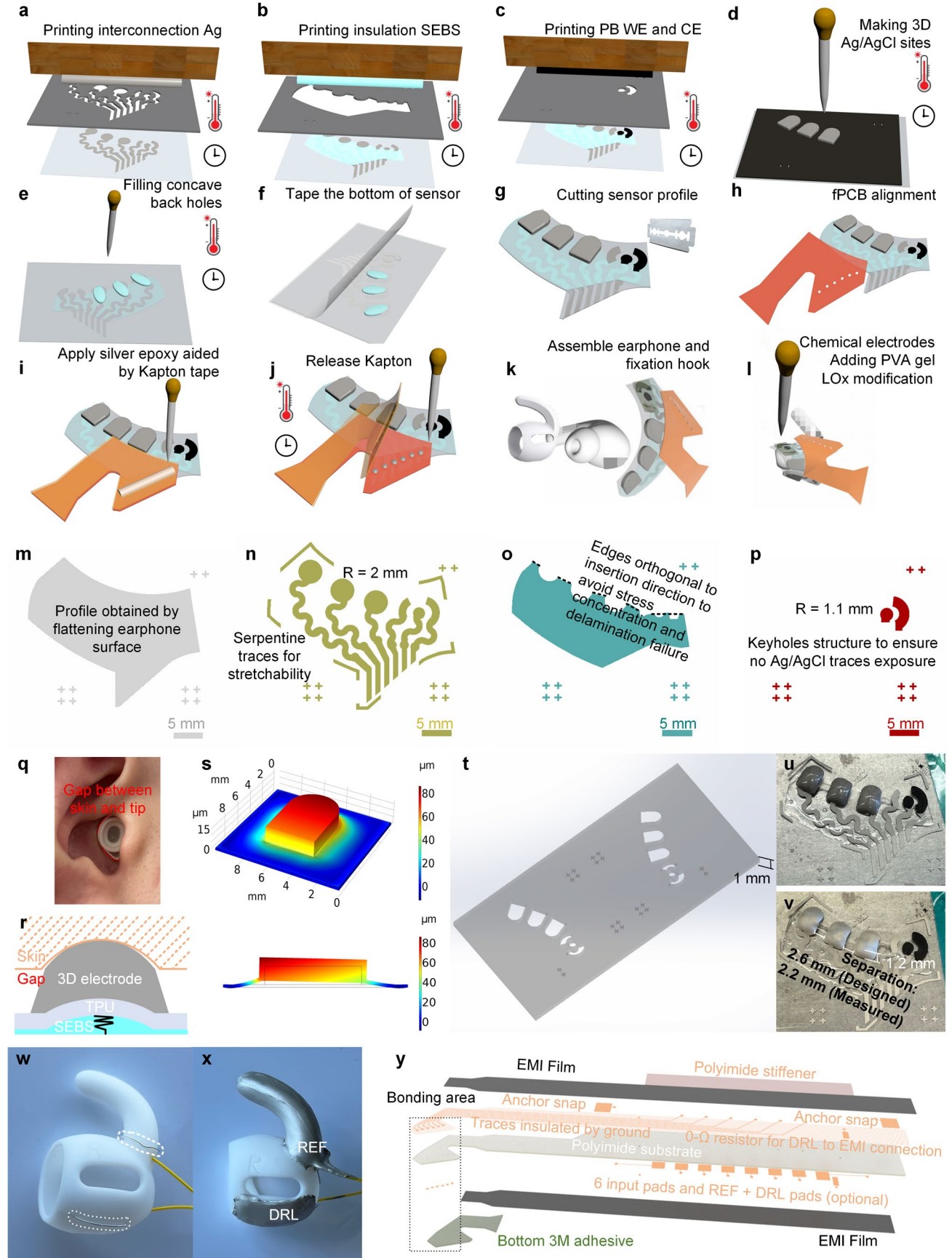

**a** Printing interconnection Ag
**b** Printing insulation SEBS
**c** Printing PB WE and CE
**d** Making 3D Ag/AgCl sites
**e** Filling concave back holes
**f** Tape the bottom of sensor
**g** Cutting sensor profile
**h** fPCB alignment
**i** Apply silver epoxy aided by Kapton tape
**j** Release Kapton
**k** Assemble earphone and fixation hook
**l** Chemical electrodes Adding PVA gel LOx modification

**m** Profile obtained by flattening earphone surface
5 mm

**n** R = 2 mm
Serpentine traces for stretchability
5 mm

**o** Edges orthogonal to insertion direction to avoid stress concentration and delamination failure
5 mm

**p** R = 1.1 mm
Keyholes structure to ensure no Ag/AgCl traces exposure
5 mm

**q** Gap between skin and tip

**r** Skin / Gap / 3D electrode / TPU / SEBS

**s**

**t** 1 mm

**u**

**v** Separation: 2.6 mm (Designed) 2.2 mm (Measured) 1.2 mm

**w**

**x** REF / DRL

**y** EMI Film / Polyimide stiffener / Bonding area / Anchor snap / Anchor snap / Traces insulated by ground / 0-Ω resistor for DRL to EMI connection / Polyimide substrate / 6 input pads and REF + DRL pads (optional) / Bottom 3M adhesive / EMI Film

**Extended Data Fig. 2 | See next page for caption.**

**Extended Data Fig. 2 | Fabrication and design specifications of the in-ear integrated sensors. a–l.** Fabrication procedure. (**a**) Print and cure (60 °C for 10 minutes) the stretchable Ag interconnection layer with the corresponding stencil. (**b**) Print and cure (60 °C for 10 minutes) the SEBS insulation layer with the corresponding stencil. (**c**) Print and cure (60 °C for 10 minutes) the stretchable PB layer for lactate sensing WE and CE with the corresponding stencil. (**d**) Mold and cure (room temperature for 30 minutes) the 3D electrophysiological electrodes with a 3D printed PLA mold. (**e**) Drop cast and fill the 3D electrophysiological electrode back holes with the SEBS. (**f**) Adhere a piece of double-sided medical tape onto the back of the sensor enclosing the SEBS backing. (**g**) Cut the profile of the sensor. (**h**) Align and adhere the flexible PCB to the bonding pads. (**i**) Fill silver epoxy onto a piece of Kapton tape with openings aligned to the sensor's bonding pads. (**j**) Release the Kapton tape to cure and pattern the silver epoxy onto corresponding bonding pads. (**k**) Peel off the release liner of the medical tape at the back, assemble the sensor onto the targeted location of the earphone, and install the earphone hook as fixation. (**l**) Adhere the PVA hydrogel and modify the lactate electrodes with LOx. **m–p.** Pattern design of each layer. (**m**) Substrate TPU layer. (**n**) Interconnection stretchable Ag layer. (**o**) Insulation SEBS layer. (**p**) Electrochemical electrode stretchable PB layer. **q–s.** 3D electrophysiological electrode design. (**q**) Air gap observed for the earphone's silicone tip to the ear canal. (**r**) Design principle of the 3D electrophysiological electrode with 'spring-loaded' backing. (**s**) Finite element simulation of the 3D electrophysiological electrode structure showing the curved back hole formation via thermal expansion coefficient mismatch between the TPU and the stretchable Ag ink. **t.** 3D printed PLA mold structure. **u–v.** Molded 3D electrophysiological electrode (**u**) before and (**v**) after curing. **w–x.** Silicone earphone hook stretchable Ag ink modification, (**w**) REF and DRL exposed wire location. (**x**) Cured stretchable Ag electrodes on the earphone hook. **y.** Connecting flexible PCB layer by layer design.

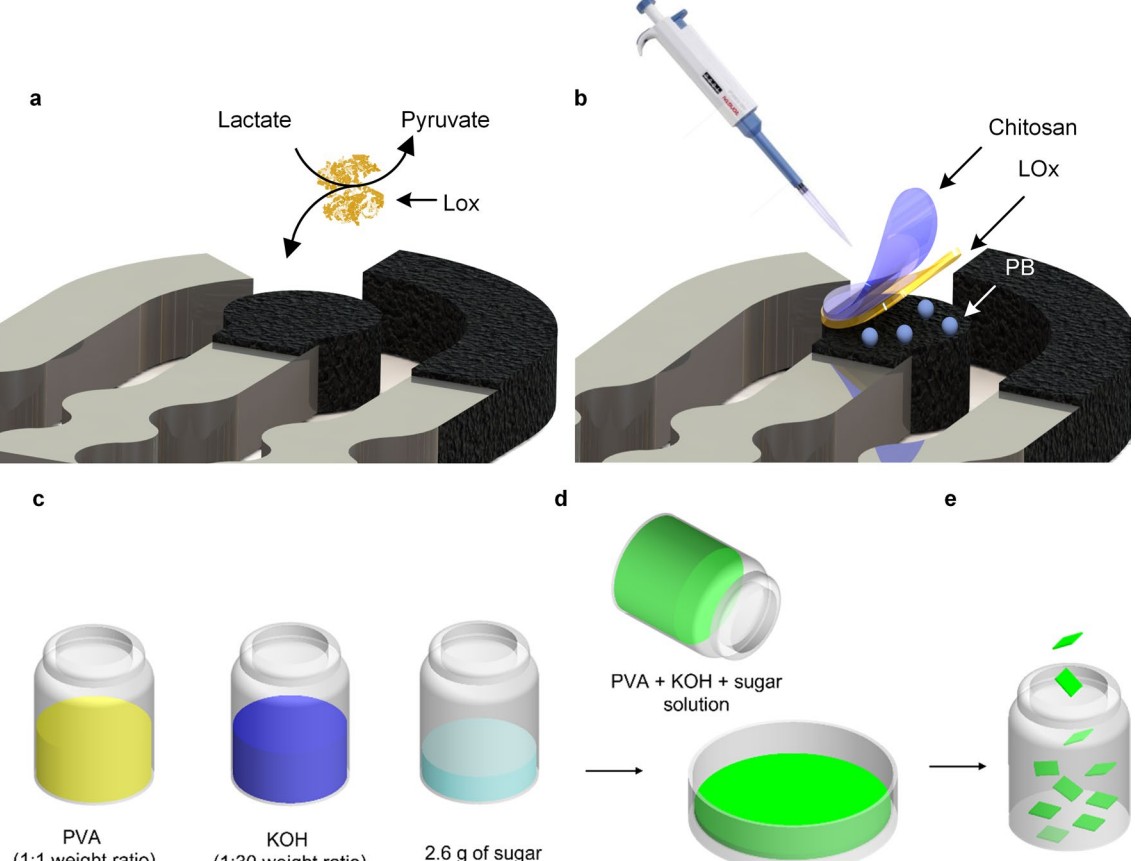

**Extended Data Fig. 3 | Electrochemical electrode modification preparations.**
**a**–**b**. electrochemical lactate electrodes mechanism and modification.
(**a**) Lactate sensing was based on detecting the biocatalytic oxidation of lactate, glucose, or alcohol molecules to pyruvate along with the production of hydrogen peroxide. (**b**) The Prussian-blue working electrode was modified first by drop-casting 1.5 µL of a solution of LOx (40 mg/mL) containing BSA (10 mg/mL) in 0.1 M PBS pH 7.3. The sensor was left at room temperature to allow the drying of the enzymatic layer. After the layer was dried, 2 µL chitosan (0.5 wt% in acetic acid) was drop-casted on the sensor to stabilize the biocatalyst layer on the PB

surface. The modified electrochemical sensor was then stored at 4 °C overnight. The modification was made after the sensor was assembled onto the earphone. **c**–**e**. PVA hydrogel preparation. (**c**) A mixture of solutions of PVA, KOH, and sugar were prepared separately in water. (**d**) Next, a mixture of 10 g of PVA, 14 g of KOH, and 2 mL of sugar solutions were mixed inside a glass container under mild stirring. A volume of 15 g of the resulting solution was poured into a glass slide and left inside a vacuum desiccator overnight. (**e**) The formed hydrogel (thickness: 0.5 mm) was immersed in 0.1 M PBS pH 7 to remove the residue. After washing, the hydrogel disk was sliced and stored in 0.1 M PBS buffer.

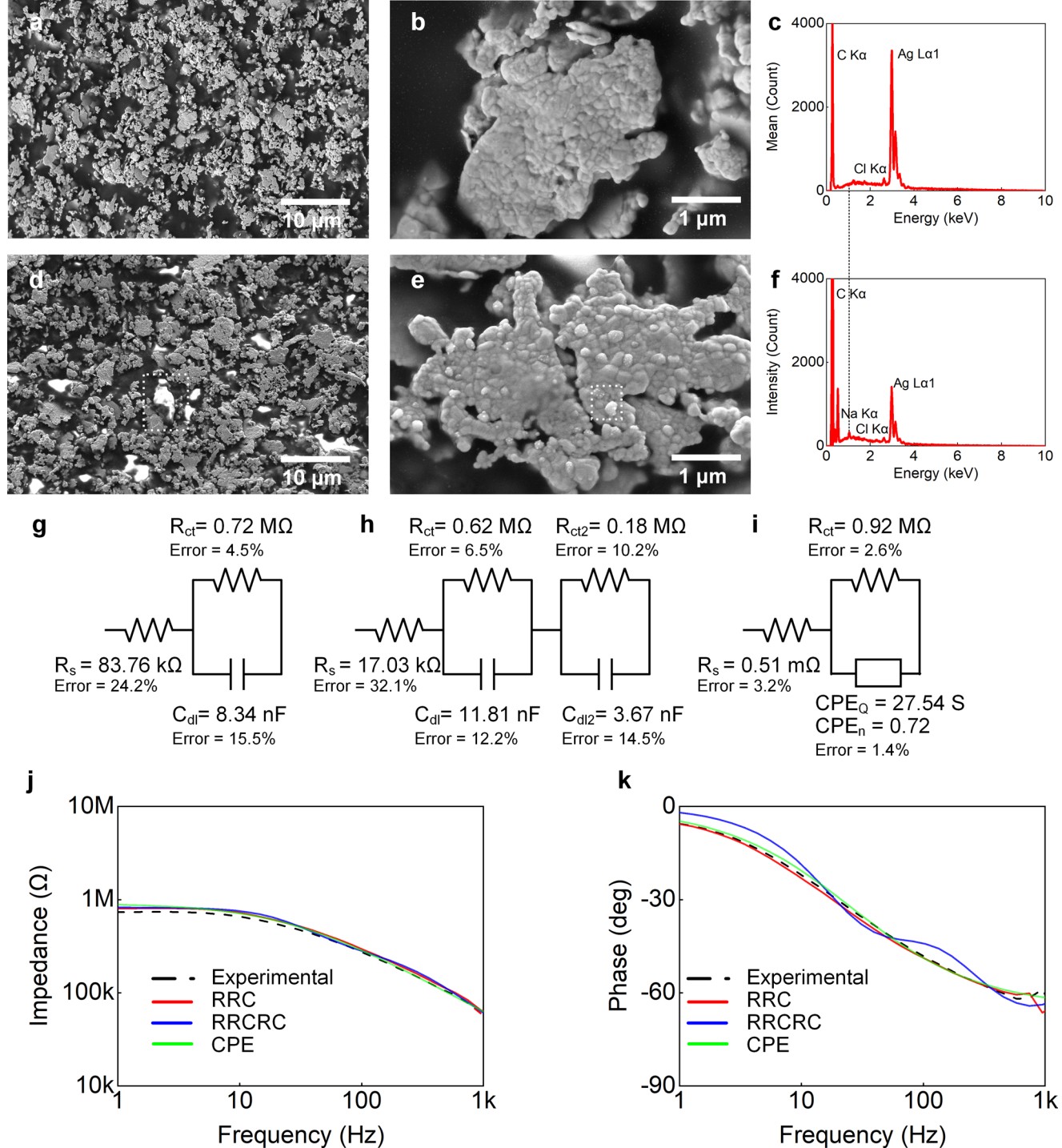

**Extended Data Fig. 4 | Electrophysiological electrode-ear impedance analysis. a–f.** Electrophysiological electrode composition characterization before and after interfacing with the ear canal. Pre-insertion SEM (scanning electron microscope) at two magnifications (**a**) and (**b**), and (**c**) EDS (energy-dispersive X-ray spectroscopy) of the electrophysiological electrodes. The EDS is the average of three scanned surface locations on the electrophysiological electrode. Post-insertion SEM at two magnifications (**d**) and (**e**), and (**f**) EDS of the electrophysiological electrodes. Sweat residue was observed in the SEMs and indicated by a 176% increase in the intensity of $Na^+$ in the EDS. **g–k.** Equivalent model fitting results with the (**g**) RRC model, (**h**) the RRCRC model, and (**i**) the constant phase element (CPE) model. (**j**) Magnitude fitting results. (**k**) Phase fitting results.

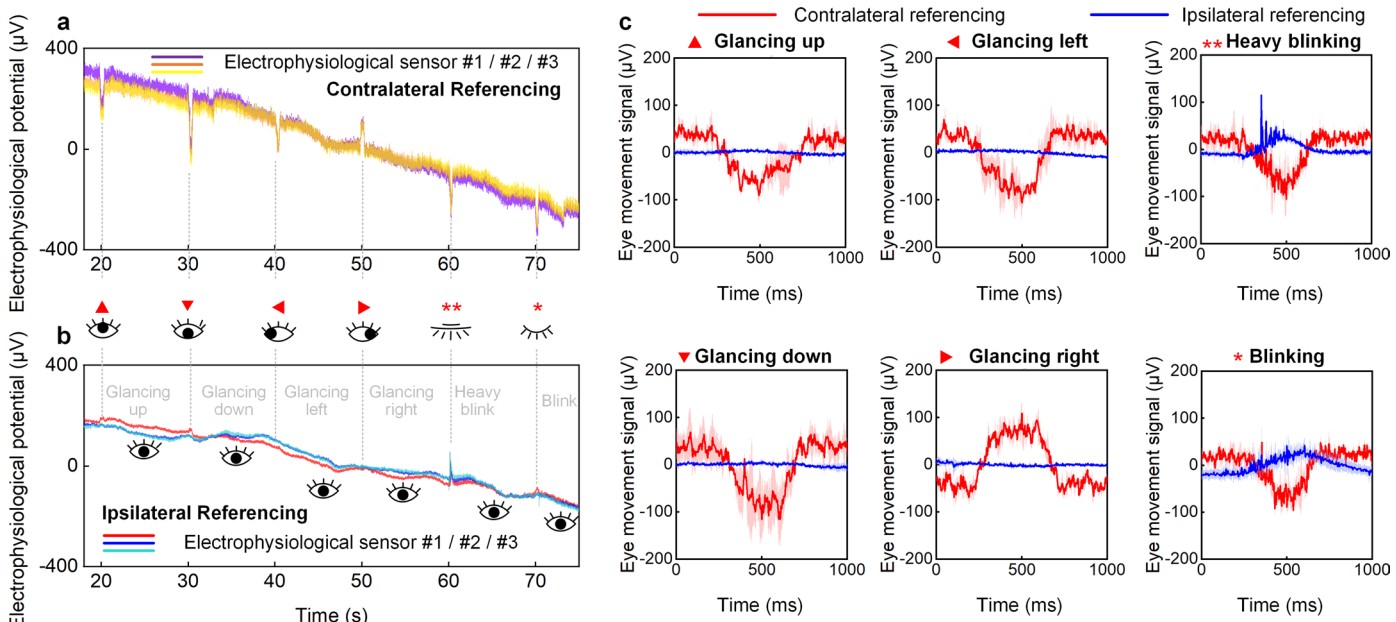

**Extended Data Fig. 5 | Characterization of the EOG.** Transient response to various eye movements recorded with **a**. contralateral and **b**. ipsilateral referencing. Transient eye movements performed by the participants included 'glancing up', 'glancing down', 'glancing left', 'glancing right', 'heavy blinking' and 'blinking', each returning to centered gaze before the next eye movement. **c**. Time-averaged transient response for each eye movement. The mean and standard deviation are shown in solid lines and shaded bands for all participant-averaged data.

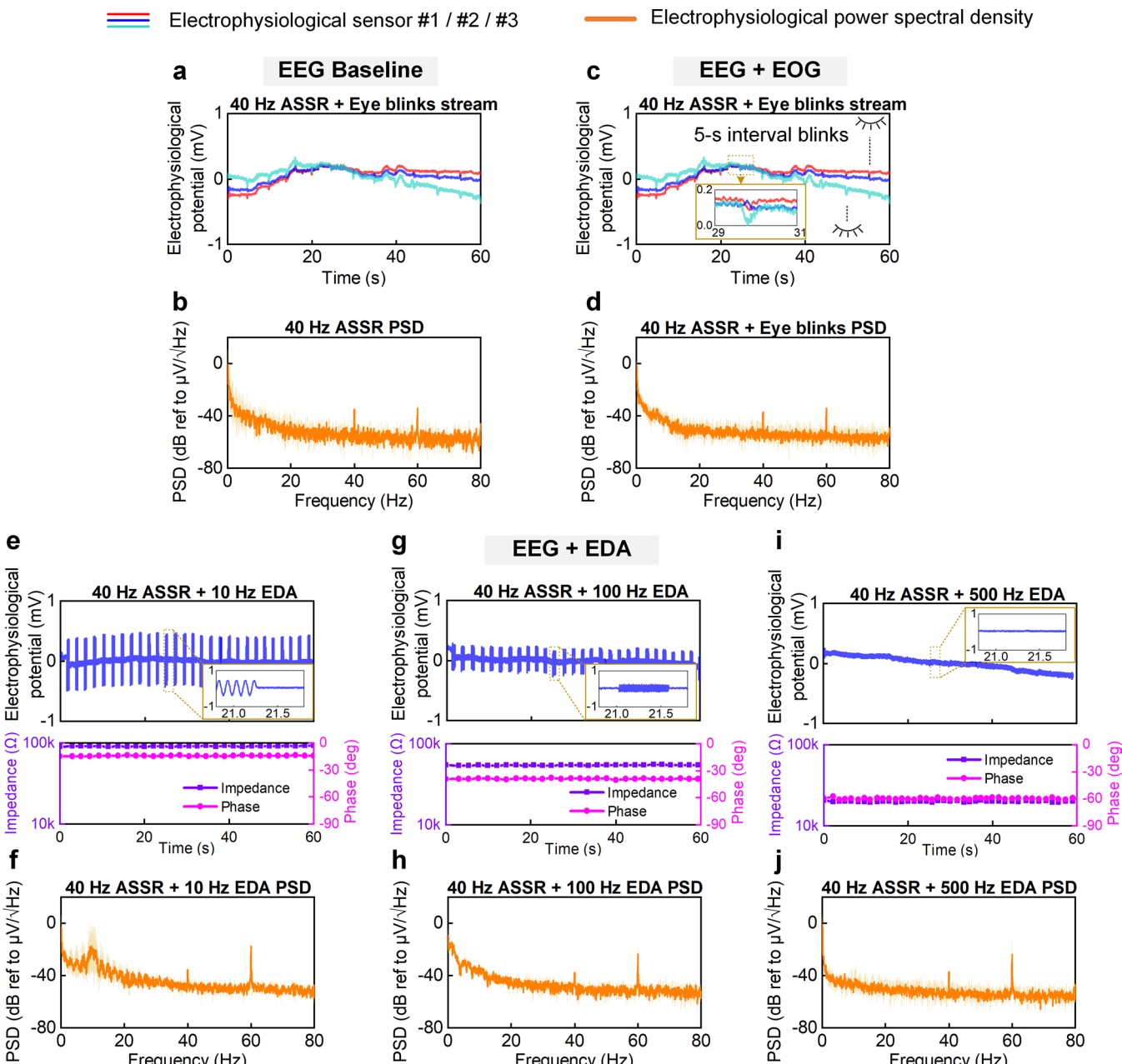

**Extended Data Fig. 6 | Characterization of combined multimodal electrophysiological in-ear sensing capability. a–b.** EEG-only baseline recording of 40 Hz ASSR (**a**) time series data, and (**b**) its PSD. **c–d.** Simultaneous EEG + EOG recording of 40 Hz ASSR (**c**) time series data and (**d**) its PSD under eye blinks at 5-s intervals. **e–j.** Simultaneous EEG + EDA measurement for the same auditory stimulus. The ear-electrode impedance was continuously measured at 10 Hz (**e,f**), 100 Hz (**g, h**), and 500 Hz (**i,j**), with simultaneous EEG and impedance time series data shown in (**e,g,i**) and the corresponding EEG PSD shown in (**f,h,j**). The mean and standard deviation of the measured 40 Hz ASSR PSDs across all 3 electrophysiological sensors are represented by solid lines and shaded bands, respectively.

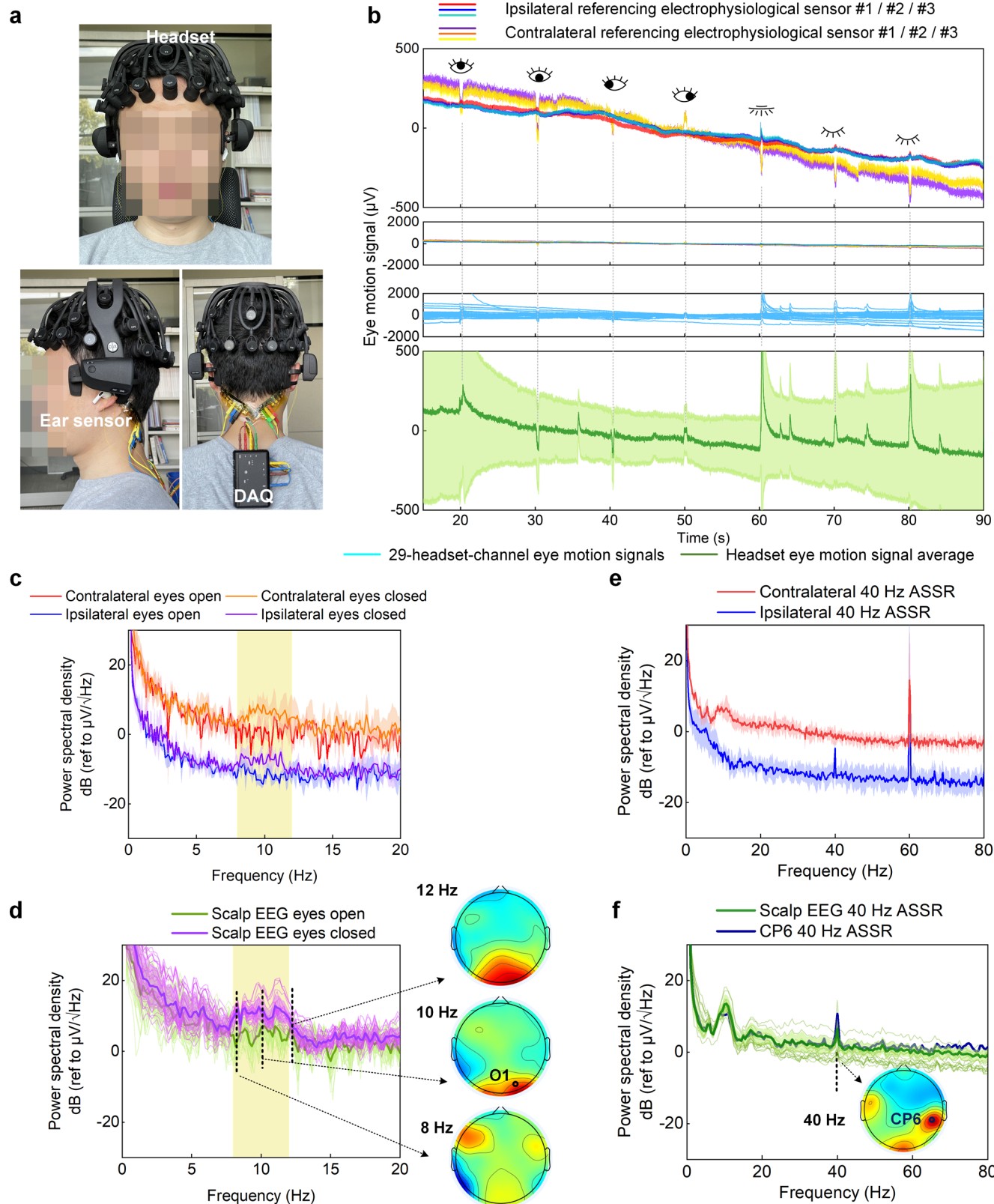

**Extended Data Fig. 7 | Controlled validation of the in-ear integrated sensors' electrophysiological sensing performance with a commercial dry contact EEG headset. a**. Experimental setup with both the in-ear integrated sensors and the EEG headset. Participants wore a pair of in-ear integrated sensors in both ears, with the REF and DRL on the left side (ipsilateral referencing for the left ear sensor, and contralateral referencing for the right ear sensor). **b**. Eye movement signals from one participant simultaneously measured with the in-ear integrated sensors and the EEG headset. **c**–**d**. Controlled alpha modulation measurement with participant-averaged PSDs obtained from measurement with (**c**) 6 electrophysiological sensors in each pair of ears, and (**d**) 27 channels of the EEG headset. O1 is the channel at the occipital lobe that recorded a significant alpha modulation signal. The yellow shaded interval illustrates the 8–12 Hz alpha band. **e**–**f**. Controlled 40 Hz ASSR PSDs obtained from measurement with (**e**) 6 electrophysiological sensors in each pair of ears, and (**f**) 19 channels of the EEG headset. CP6 is the channel nearest to the auditory cortex that recorded a significant ASSR signal. For each in-ear and headset electrophysiological sensor, the participant-averaged mean and standard deviation of the measured time series and PSD, are represented by solid lines and shaded bands, respectively.

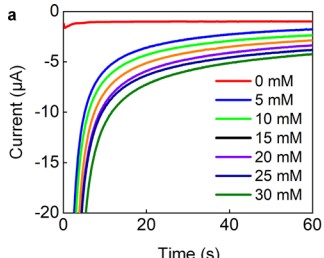 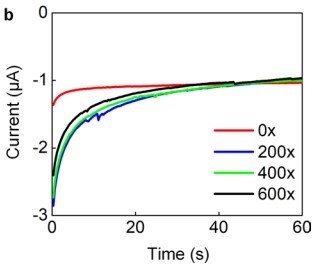

**Extended Data Fig. 8 | Characterization of the in-ear integrated sensors' lactate sensing performance. a**. In vitro lactate sensing: calibration experiment with increasing lactate concentration. **b**. Lactate in-vitro characterization before and after mechanical stretching of the in-ear integrated sensors for 200, 400, and 600 cycles, respectively.

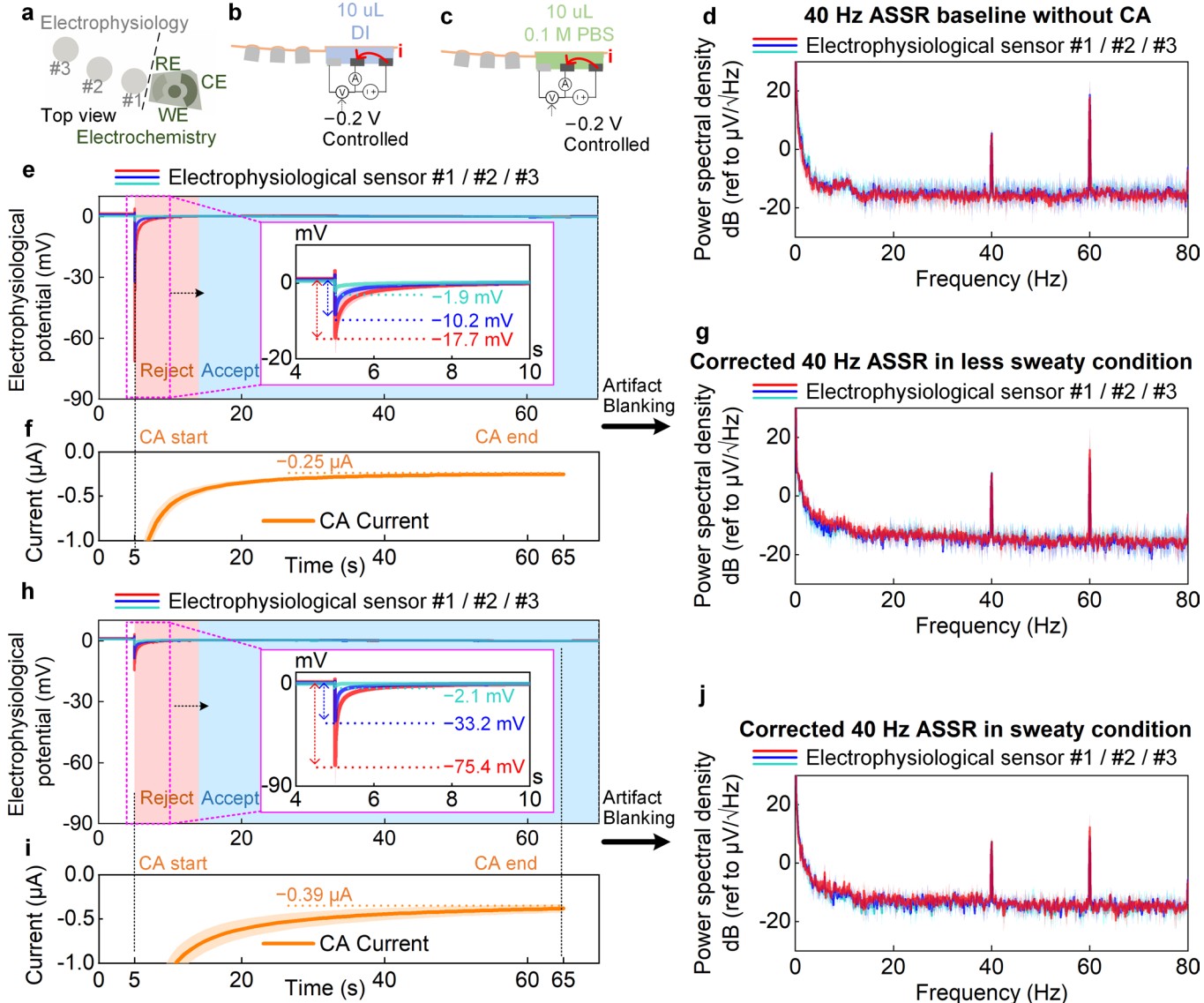

**Extended Data Fig. 9 | Co-sensing crosstalk characterization of the in-ear integrated sensors. a.** Top view of the electrode layout of the in-ear integrated sensors. **b–c.** Co-sensing setup, with testing performed under (**b**) less sweaty and (**c**) more sweaty conditions. **d.** ASSR PSD of the three 70-s 40-Hz ASSR baseline measurements in the absence of electrochemical recording. **e–j.** Effect of lactate chronoamperometry (CA) current recording on simultaneous EEG recording under less sweaty (**e–g**) and more sweaty (**h–j**) conditions. (**e,h**) Time traces of the 40-Hz ASSR measurements, with light red and blue shaded regions representing the data rejection and data acceptance intervals used in the artifact blanking method. (**f,i**) Currents recorded for CA measurements initiated at $t = 5$ s and completed at $t = 65$ s into the simultaneous EEG recording. (**g,j**) PSD of the 40 Hz ASSR after artifact blanking (averaged rejection interval: 9.63 s under less sweaty conditions, and 10.78 s under more sweaty conditions; see Supplementary Note 4 for more details). For each electrophysiological and electrochemical sensor, the participant-averaged mean and standard deviation of the measured time series and PSD, are represented by solid lines and shaded bands, respectively.

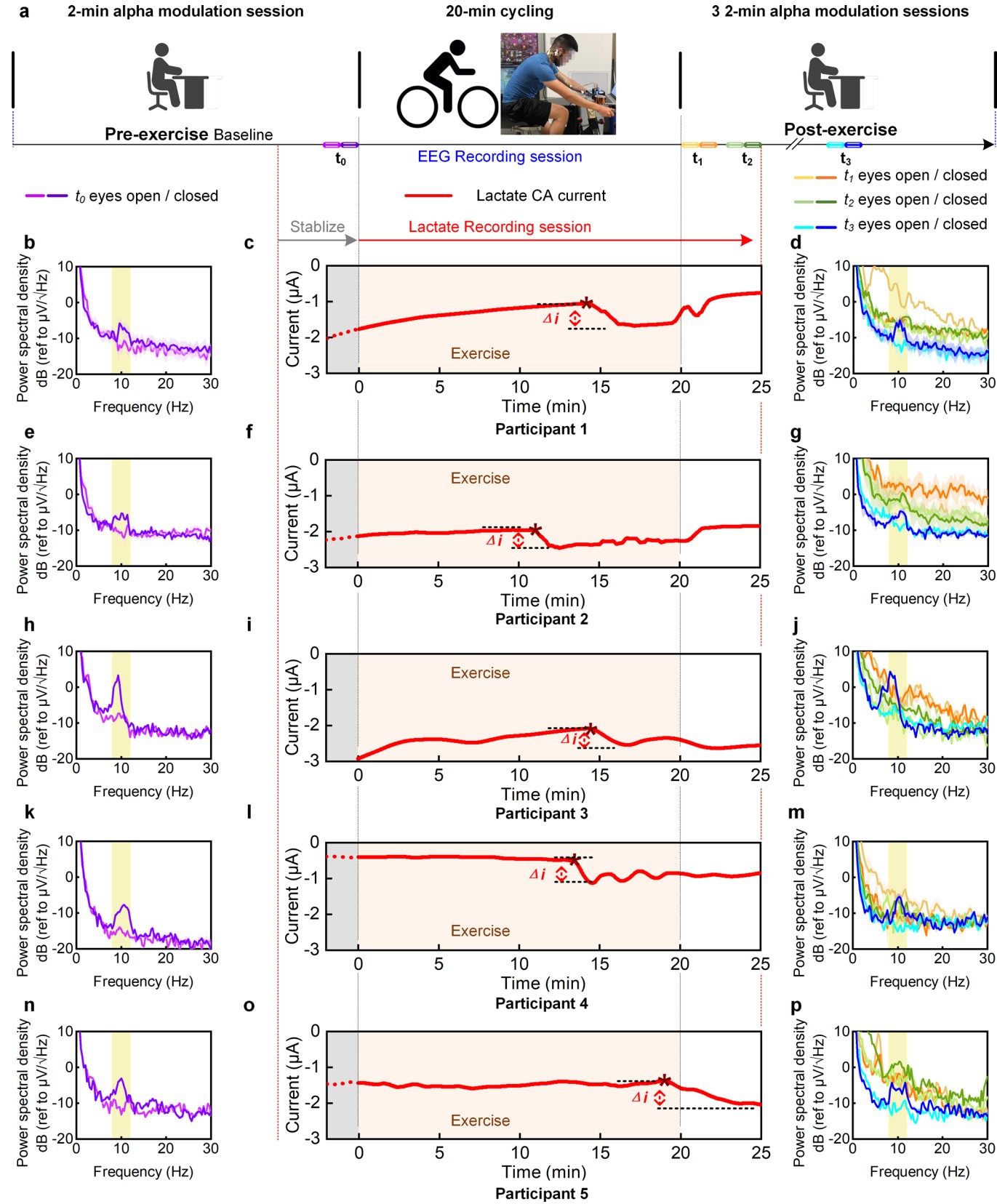

**Extended Data Fig. 10 | Individual participant EEG and lactate integrated sensing data before, during, and after exercise. a.** Experimental timeline used for the integrated experiments. EEG and lactate recordings from participants 1 (**b**–**d**), 2 (**e**–**g**), 3 (**h**–**j**), 4 (**k**–**m**) and 5 (**n**–**p**). (**b,e,h,k,n**) Pre-exercise alpha modulation data from individual participants. (**c,f,i,l,o**) Lactate CA current measurements from individual participants, showing lactate concentration variation during the experiments. (**d,g,j,m,p**) Post-exercise alpha modulation

data from individual participants. The yellow shaded interval illustrates the 8–12 Hz alpha band. The pre-and post-exercise alpha modulation ratios for the five participants were 2.13 (V²/V²) vs 2.25 (V²/V²), 2.21 (V²/V²) vs 1.58 (V²/V²), 5.04 (V²/V²) vs 4.91 (V²/V²), 3.77 vs 3.24 (V²/V²), and 2.88 vs 3.10 (V²/V²), respectively. For alpha modulation results, the mean and standard deviation of the measured EEG PSD across all 3 electrophysiological sensors per participant are represented by solid lines and shaded bands, respectively.

# Reporting Summary

## Statistics

For all statistical analyses, confirm that the following items are present in the figure legend, table legend, main text, or Methods section.

| n/a | Confirmed | |
|---|---|---|
| ☐ | ☒ | The exact sample size (*n*) for each experimental group/condition, given as a discrete number and unit of measurement |
| ☐ | ☒ | A statement on whether measurements were taken from distinct samples or whether the same sample was measured repeatedly |
| ☒ | ☐ | The statistical test(s) used AND whether they are one- or two-sided<br>*Only common tests should be described solely by name; describe more complex techniques in the Methods section.* |
| ☒ | ☐ | A description of all covariates tested |
| ☐ | ☒ | A description of any assumptions or corrections, such as tests of normality and adjustment for multiple comparisons |
| ☐ | ☒ | A full description of the statistical parameters including central tendency (e.g. means) or other basic estimates (e.g. regression coefficient) AND variation (e.g. standard deviation) or associated estimates of uncertainty (e.g. confidence intervals) |
| ☒ | ☐ | For null hypothesis testing, the test statistic (e.g. *F*, *t*, *r*) with confidence intervals, effect sizes, degrees of freedom and *P* value noted<br>*Give P values as exact values whenever suitable.* |
| ☒ | ☐ | For Bayesian analysis, information on the choice of priors and Markov chain Monte Carlo settings |
| ☒ | ☐ | For hierarchical and complex designs, identification of the appropriate level for tests and full reporting of outcomes |
| ☐ | ☒ | Estimates of effect sizes (e.g. Cohen's *d*, Pearson's *r*), indicating how they were calculated |

*Our web collection on statistics for biologists contains articles on many of the points above.*

## Software and code

Policy information about availability of computer code

| Data collection | Electrophysiological data (EEG, EOG, Electrode DC offset) were collected from the sensors using the BioCapture software (version 5.5.640) (Great Lakes Neurotechnologies). In the validation test, electrophysiological data (EEG, EOG) collected from the Cognionics headset used the CGX 2021 (Cognionics) software.<br><br>Electrode-ear impedance and chronoamperometry (including lactate measurement) data were collected from the sensors using the PSTrace 5.9 (Palmsense). In the validation test, lactate data collected from the NOVA Biomedical blood-lactate meter used the CHI instruments. |
|---|---|
| Data analysis | Electrophysiological data (EEG, EOG, Electrode DC offset), electrode-ear impedance data and chronoamperometry data (including lactate measurement) were analysed and visualized using Matlab R2021b (Mathworks) and OriginPro 2021 (Originlab). ASR EEG artifact algorithm was implemented using Matlab R2021b.<br>Filter-bank-based common-spatial-pattern method for EEG cognitive-state classification used scripts coded in Python.<br><br>Custom codes for electrophysiological signal analysis, for the automatic subspace reconstruction (ASR) algorithm, and for the filter-bank-based common-spatial-pattern (FBCSP) method, are available at https://doi.org/10.1038/zenodo.8193117. |

For manuscripts utilizing custom algorithms or software that are central to the research but not yet described in published literature, software must be made available to editors and reviewers. We strongly encourage code deposition in a community repository (e.g. GitHub). See the Nature Portfolio guidelines for submitting code & software for further information.

## Data

Policy information about <u>availability of data</u>

All manuscripts must include a <u>data availability statement</u>. This statement should provide the following information, where applicable:

- Accession codes, unique identifiers, or web links for publicly available datasets
- A description of any restrictions on data availability
- For clinical datasets or third party data, please ensure that the statement adheres to our <u>policy</u>

All data supporting the results in this study are available within the paper and its Supplementary Information. Source data for the figures are provided with this paper, and are available in figshare at https://doi.org/10.6084/m9.figshare.22829051.

## Human research participants

Policy information about <u>studies involving human research participants and Sex and Gender in Research.</u>

| | |
|---|---|
| Reporting on sex and gender | The gender of all participants is indicated in Supplementary Table 3. |
| Population characteristics | In-ear sweat gland mapping: 3 participants and 3 recording sessions. <br> Ear-electrode impedance experiment: 2 participants and 6 measurement sessions for in total 6 electrophysiological electrodes. <br> Electrode DC offset and EOG: 2 participants and 12 measurement sessions for in total 12 electrophysiological electrodes (two ears per participant). <br> Alpha modulation and auditory steady state response: 4 participants and 12 measurement sessions for in total 12 electrophysiological electrodes. <br> Simultaneous multi-modal electrophysiological recording: 1 participant and 1 recording session for in total 3 electrophysiological electrodes (EEG + EOG, EEG + EDA). <br> Electrophysiological validation with the dry EEG headset: 2 participants and 4 recording sessions for in total 12 electrophysiological electrodes (Alpha, ASSR, EOG). <br> Motion artifact analysis: 1 participant and 1 recording session for in total 3 electrophysiological electrodes. <br> On-body lactate sensing assessment: 3 participants and 3 recording sessions. <br> Crosstalk analysis of electrophysiological and electrochemical sensing: 2 participants and 3 recording sessions (one participant with both ears). <br> EEG and lactate co-sensing experiment: 5 participants and 5 recording sessions. |
| Recruitment | Recruitment information was shared via flyers and an online platform. No age, race/ethnicity or gender-based participant-exclusion criteria were used. The participants recruited were healthy, and were asked to fill out an online qualification form before participating in the experiment. |
| Ethics oversight | University of California San Diego Institutional Review Boards. |

Note that full information on the approval of the study protocol must also be provided in the manuscript.

# Field-specific reporting

Please select the one below that is the best fit for your research. If you are not sure, read the appropriate sections before making your selection.

☒ Life sciences ☐ Behavioural & social sciences ☐ Ecological, evolutionary & environmental sciences

For a reference copy of the document with all sections, see nature.com/documents/nr-reporting-summary-flat.pdf

# Life sciences study design

All studies must disclose on these points even when the disclosure is negative.

| | |
|---|---|
| Sample size | Please see 'population characteristics' above. |
| Data exclusions | To avoid co-sensing crosstalk, we didn't include EEG data immediately after the onset of lactate chronoamperometry measurements. Detailed justification of the exclusions can be found in the Supplementary Information. |
| Replication | The study validated the integration of electrophysiological and electrochemical sensing in the ear. The result was dependent on the body condition for each experiment session and was thus not replicated. However, for all the electrophysiological and electrochemical validation experiments and combined EEG and lactate-sensing experiments, the measurements were taken from multiple participants, to demonstrate repeatability. |
| Randomization | The human participants for each experiment were selected randomly. |

| Blinding | No blinding experiments were carried out. The data were directly processed after collection. |

# Reporting for specific materials, systems and methods

We require information from authors about some types of materials, experimental systems and methods used in many studies. Here, indicate whether each material, system or method listed is relevant to your study. If you are not sure if a list item applies to your research, read the appropriate section before selecting a response.

## Materials & experimental systems

| n/a | Involved in the study |
|-----|----------------------|
| ☒ ☐ | Antibodies |
| ☒ ☐ | Eukaryotic cell lines |
| ☒ ☐ | Palaeontology and archaeology |
| ☒ ☐ | Animals and other organisms |
| ☒ ☐ | Clinical data |
| ☒ ☐ | Dual use research of concern |

## Methods

| n/a | Involved in the study |
|-----|----------------------|
| ☒ ☐ | ChIP-seq |
| ☒ ☐ | Flow cytometry |
| ☒ ☐ | MRI-based neuroimaging |

