## [Peer Review File · Nature Biomedical Engineering]

In-ear integrated sensor array for the continuous monitoring of brain activity and of lactate in sweat

Corresponding author: Gert Cauwenberghs

Editorial note

This document includes relevant written communications between the manuscript's corresponding author and the editor and reviewers of the manuscript during peer review. It includes decision letters relaying any editorial points and peer-review reports, and the authors' replies to these (under 'Rebuttal' headings). The editorial decisions are signed by the manuscript's handling editor, yet the editorial team and ultimately the journal's Chief Editor share responsibility for all decisions.

Any relevant documents attached to the decision letters are referred to as **Appendix #**, and can be found appended to this document. Any information deemed confidential has been redacted or removed. Earlier versions of the manuscript are not published, yet the originally submitted version may be available as a preprint. Because of editorial edits and changes during peer review, the published title of the paper and the title mentioned in below correspondence may differ.

Correspondence

Sat 16 Jul 2022

Decision on Presubmission Enquiry nBME-22-1647-PE

Dear Dr Cauwenberghs,

Thank you for submitting to *Nature Biomedical Engineering* your Presubmission Enquiry, "Unobtrusive In-Ear Integrated Physiological and Metabolic Sensors for Continuous Brain-Body Activity Monitoring".

We find the integration of the electrophysiological and metabolic sensors into the ear canal and the testing in volunteers a compelling engineering advance, and hence we will be glad to consider the work for peer review.

I should ask you to please fill in our reporting summary and policy checklist. (Please note that these forms are dynamic PDF files that can only be properly visualized and filled in by using Acrobat Reader.) Both documents are aimed at ensuring good reporting standards and at easing the interpretation of results, and will be available to any reviewers. Should the manuscript be eventually published, the reporting summary will be attached to the published PDF of the paper and will also be available as supplementary information. More information is available on the editorial policies page.

When you are ready to submit the manuscript, please upload the manuscript files as well as the reporting summary and policy checklist.

Best wishes,

Pep

—
Pep Pàmies
Chief Editor, Nature Biomedical EngineeringSat 21 Jan 2023

Decision on Article nBME-22-1647A

Dear Dr Cauwenberghs,

Thank you again for submitting to *Nature Biomedical Engineering* your manuscript, "Unobtrusive In-Ear Integrated Physiological and Metabolic Sensors for Continuous Brain-Body Activity Monitoring". As I communicated in previous e-mail correspondence, the manuscript has been seen by three experts, yet one reviewer has not delivered a report (and, despite our chasing efforts, it is unlikely that they will). You will find the reports that I had already forwarded to you at the end of this message.

You will see that the reviewers appreciate the work. However, they express concerns about the degree of support for the claims, and provide useful suggestions for improvement. We hope that with significant further work you can address the criticisms and convince the reviewers of the merits of the study. In particular, we would expect that a revised version of the manuscript provides:

- * Improved design of the device, towards improved usability, robustness and user-friendliness .
- * Validation of the performance of the device, in particular of simultaneous measurements and via relevant physiologically relevant quantitative metrics, in a larger number of individuals.
- * Improved background context, and extended discussion of the most promising applications and of the current limitations of the integrated device.
- * Thorough methodological details.

I would also like to mention that, depending on the improvements in the revision, we may seek to enrol an additional expert in the next round of review.

When you are ready to resubmit your manuscript, please upload the revised files, a point-by-point rebuttal to the comments from all reviewers, the reporting summary, and a cover letter that explains the main improvements included in the revision and responds to any points highlighted in this decision.

Please follow the following recommendations:

- * Clearly highlight any amendments to the text and figures to help the reviewers and editors find and understand the changes (yet keep in mind that excessive marking can hinder readability).
- * If you and your co-authors disagree with a criticism, provide the arguments to the reviewer (optionally, indicate the relevant points in the cover letter).
- * If a criticism or suggestion is not addressed, please indicate so in the rebuttal to the reviewer comments and explain the reason(s).
- * Consider including responses to any criticisms raised by more than one reviewer at the beginning of the rebuttal, in a section addressed to all reviewers.
- * The rebuttal should include the reviewer comments in point-by-point format (please note that we provide all reviewers will the reports as they appear at the end of this message).
- * Provide the rebuttal to the reviewer comments and the cover letter as separate files.

We hope that you will be able to resubmit the manuscript within 16 weeks from the receipt of this message. If this is the case, you will be protected against potential scooping. Otherwise, we will be happy to consider a revised manuscript as long as the significance of the work is not compromised by work published elsewhere or accepted for publication at *Nature Biomedical Engineering*.

We hope that you will find the referee reports helpful when revising the work. Please do not hesitate to contact me should you have any questions.

Best wishes,

Pep

Pep Pàmies
Chief Editor, Nature Biomedical Engineering

Reviewer #1 (Report for the authors (Required)):

In this paper, the authors present a pioneering integration of two dimensions of electrophysiological brain state monitoring and health-related metabolite monitoring into an unobtrusive wearable sensor. It makes full use of the extremely limited space in the ear canal and provides an elegant solution for user monitoring. The authors solved the problem of measurement signal interference and optimal synergy between sensors with different sensing methods. This work breaks through traditional challenges such as sensors' size and area limitations and achieves brain state and metabolite monitoring in an unobtrusive manner. Overall, this manuscript demonstrates an interesting design with advantages over existing state-of-the-art technologies. However, there are several issues that the authors must address to improve the quality and clarity of their work.

Major concerns:

1. Ingenuity compared to previous work? Both electrophysiological brain status monitoring (IEEE Trans. Biomed. Circuits Syst. 2020, 14(4), 727) and health-related metabolite monitoring (Sci. Robot. 2020, 5(41), eaaz7946) (Nat. Biomed. Eng. 2021, 5, 737) dimensions have been reported before. Among these reported research, electrophysiological brain state monitoring has been integrated into sensors of similar size to monitor physiological signals. The chemical monitoring of metabolites has also achieved the monitoring of biomarkers such as lactate through electronic skin (e-skin) and patches. Please compare this work with previous work and explain the unique strengths, breakthroughs, and ingenuity of this work.
2. How to fit users' different ear canals? Due to the extremely limited space in the ear and the large variation in anatomy between the ears of different users (lines 51-52), which can lead to a geometric mismatch between the in-ear sensor and the ear canal (lines 95-96). The authors proposed the solution by applying the ePhys stretchable Ag electrode with a three-dimensional structure (lines 97-99). However, the solution principle for solving the adaptation problem is not detailed, please clarify the principle.
3. It seems that the placement of eChem electrodes towards the tragus is based solely on the sweat profile of a single volunteer, as detailed in Extended Data Fig. 1, Fig. 1f, and lines 81-84. Are the in-ear sweat profiles of different people different, which would make the sensor less effective for others? If not, is this backed up by either a) the author's own experimental data or b) previous literature?
4. Will the in-ear sensor slip off? The sensor is primarily made from TPU and SEBS, both of which are hydrophobic, and the two surfaces are very smooth. Does this hydrophobic and smooth material surface cause any effect on the in-ear sensor if it conforms to the skin over time? In practice, will the sweat of the exercise cause the in-ear sensor to slip out of the ear canal? Many times, when people are exercising, they enjoy listening to music or podcasts. With the sensor design are the headphones still functional enough for such tasks?
5. Can the sensor be produced on a large scale? In this paper, the sensor is customized, designed, and processed according to the area with users' high sweat secretion, including printing and cutting sensor profile preparation. The process is complicated and requires a lot of manpower and resources to provide customized services. Will these processes limit the large-scale production of modified in-ear sensors? Will these complex processes lead to excessive production costs to make an in-ear sensor?
6. The evaluation of in-ear sensor wearing comfort? In this work, the ePhys stretchable Ag electrode of in-ear sensor adopts a three-dimensional structure to fit the contour of the ear canal (lines 97-99). However, there is a lack of a quantitative definition of fit. If the fit is low, the sensor will fall off during exercise, and

excessive fit will cause the burden and discomfort of wearing. Therefore, it is necessary to use objective comfort evaluation criteria to measure the sensor to prove the comfortable wearing and excellent performance. Please design and provide a quantitative definition of the in-ear sensor fit.

7. The authors should likewise describe how the sensor was mounted to different sites, and how the sensor connected to their peripheral electronics for wireless data transmission. The peripheral electronics should also be pictured.

8. Is the double-sided adhesive biocompatible? In this work, the in-ear sensor only verified its mechanical properties in the silicone eardrum (SI, line 174). Whether the long-term use of double-sided adhesive will bring discomfort to the skin needs further biological verification. Furthermore, could it be replaced with a more stable, reliable, and convenient sensor adhesive method?

9. Skin irritation and skin allergy considerations? The sensor is fabricated using the organic solvents toluene, as well as an alkaline solution like KOH. They are often strongly irritating to the skin and have an unpleasant odor. Does this have any potential dangers to user health when the device adapts to the skin over time? Please supplement a pathological test to certify the safety and ensure wearer comfort.

10. Will the in-ear sensor keep firm in a humid environment? During the processing of the sensor, the authors used manual cutting double-sided adhesive to assemble the sensor (254 lines). Whether will there be a defect where the adhesion is not tight enough during the manual cutting of the double-sided adhesive? In exercise, these double-sided adhesive defects might encounter water (sweat), and the adhesion will decrease, so the fastness of the sensor under different humidity needs to be further verified.

11. Why is there a need to combine both in the first place? If both metabolites and brain state demonstrate the same thing, why do we need to measure it twice? They need more background on each modality to argue this. Can more biomarkers be monitored? As mentioned in review comment 10, the combination of two dimensions of electrophysiological brain state monitoring and health-related metabolite monitoring is still lacking in sensor comprehensiveness. It can be considered that this in-ear sensor can be developed to monitor more physiological signals, thereby endowing it with higher integration and more comprehensive monitoring performance. For example, a pressure sensor can be integrated into the in-ear sensor to monitor the change of sound pressure in the ear canal to realize the judgment of the source of the sound position (J. Acoust. Soc. Am. 1989, 86, 89).

12. Could the in-ear sensor keep robust against temperature changes? At present, the authors only test the stability of the sensor under the specific temperature and humidity of the laboratory, but in the actual use scenario, the ear canal is different from the outside temperature, and the temperature fluctuates throughout the year when the sensor is used. Will the difference in thermal expansion coefficients of the TPU and SEBS materials that make up the sensor cause the sensor structure to shrink and expand to varying degrees between layers when the temperature fluctuates? As a result, the device of the in-ear sensor might be damaged. Once it falls off inside the ear canal, the damaged sensor part may fall into the human body along the cochlea, with disastrous consequences. Therefore, it needs to be supplemented within a specific temperature range. The device stability test needs to diversify the test environment and consider the impact of the actual environment.

13. Lines 168-171 state that "running 10 repetitive CA scans at a fixed concentration displayed in Fig. 3c, showed minimal changes (<4%), demonstrating the efficient entrapment of the enzymatic layer on the eChem transducer." The device was only run for 10 CA scans, which is very little compared to what is expected of continuous monitoring. Additionally, how long is considered one "scan"?

14. The reproducibility of the sweat lactate CA current data. In line 213 of the manuscript, it is stated, "The sweat lactate CA current remained for ~2 mins after the exercise stopped because of the post-exercise sweat residue". However, according to the corresponding Fig.4c, f, and i, subject 3 showed a CA current retention much more significant than 2 minutes. Therefore, please repeat the verification of the sensor performance to make it stable.

15. Lack of support from clinical data. Lines 72-74 of the manuscript mention that "The implementation of both modalities into a miniaturized in-ear non-invasive platform could thus facilitate the process of using multiple instruments for assessing these features during neurological monitoring and potentially allow self-monitoring in patients." Here, it is necessary to provide supporting information on monitoring specific

physiological disease indicators. The significance of physiological signal monitoring through sensors will be greatly reduced without clinical data support.

16. Identically, when the text discusses the usefulness of electrooculography (EOG) signature eye movements in brain-computer interface (BCI) applications, sleepiness detection, and mobile ophthalmology treatment (lines 155-156) are mentioned. Please use the in-ear sensor to collect and compare the physiological indicators of patients and healthy people, which will further prove the potential and value of the in-ear sensor in biomedicine.

17. Current electrochemical energy storage devices (e.g., batteries) were limited by energy and power density, and thus cannot power the electronics over an extended operational time. In this work, the sensors consume a lot of power, and the input battery energy storage is limited. Will these cause the issue of power supply time deficiency?

18. This in-ear sensor's future potential and potential value should be briefly explained at the end of the abstract and conclusion part. The current narrative (lines 22-23) is too bland without emphasis on the potential and application value when the interaction between the two dimensions of electrophysiological brain state monitoring and health-related metabolite monitoring is observed. For example, the authors should focus more on the profound influence on the sensor application of early disease detection, health monitoring, physical performance improvement, and virtual/augmented reality applications.

Minor concerns:

19. In the abstract, the originality and breakthrough of the work should also be highlighted, such as "the first breakthrough in combining brain state and health-related metabolite monitoring in a small size". Along the same note, the authors should also briefly discuss such limitations of EEG/metabolite and the necessity to combine it.

20. When the abbreviation of a specific noun appears for the first time in the manuscript, it is necessary to explain its full name clearly, even if it is explained in the supporting information (SI). If it is not explained clearly for the first time, it will lead to poor article readability. For example: PCB (252 lines), ADC/PGA/AVDD/AVSS (Fig.S1) and AC/AA/Gluc/UA (166 lines)

21. The article numbering is confusing: the "Supplementary Text Note B" mentioned in line 123 cannot be found.

22. Unclear referring: When referring to "a flat bottom with an adhesive layer" (lines 84-85), it is hard to know which layer of Fig.1d it is, please specify it.

23. Regarding "a fast and low-cost printing-bonding-assembly process" (lines 85-86), Fig. 1e cannot explain the whole process alone. Please refer together to Extended Data Fig. 2, which can help to detail the total printing process.

24. For all Fig: The figures quality of the article is low at present, and there are a lot of defects in the text format, schematic drawing, and picture layout. If the quality of the figures can be improved as follow comments (22-29), it will greatly improve the readability of the article and help this work to be more impressive.

All the outline boxes of all figures and text, including solid line boxes and dotted line boxes, contain no scientific information. Please remove them. For example, the gray dotted line box in Fig. 1b, the yellow/red/gray solid line box in Fig. 1f, the gray line box for printing/bonding/assembly, the black solid line in Extended Data Fig. 1f, the black solid line in Extended Data Fig. 2, etc.

Fig. 1: The organization of the pictures is too compact, which will cause certain reading comprehension obstacles. There should be a certain distance between each small picture (such as the distance between Fig. 1a, c, d, and e)

Fig. 1a: Please keep the style (realistic or anime) and color of each element consistent, including sweat, brain signals, and ear pictures. And the font size in Fig. 1a should be consistent with other parts in Fig. 1.

Fig. 1c: The thickness of all lines and the interval of dashed lines should be kept as consistent as possible (for example, the dashed lines drawn from Fig. 1a to c should be the same as the dashed box style in Fig. 1c). The layout of Fig. 1c is also confusing. The dashed lines should correspond to areas on the images in Fig. 1b, rather than Fig. 1a, as it suggests that the electrodes are on the outer ear rather than the headphone.

Fig. 1e: Each font style and weight should be consistent with the rest part of Fig. 1.

All the line icons in Fig. 2 should be placed inside the corresponding pictures. This can help readers understand the meaning of the icons more clearly and reduce misunderstandings. For example, the red line icon indicating continuous in-ear impedance should be placed in Fig. 2a, and so on for other parts. In addition, the text description of the picture can be shortened, it is a bit too long now, and it can be described in detail in the caption below the picture.

In Fig. 2, all borders on the left and right sides should be kept in a line, and the second and fourth lines have obvious indents, which should be adjusted accordingly. This adjustment will make the figure clean and improve its quality.

Fig. 2v: The number is wrong, please revise v to b.

Fig. 2m: The grey font is too inconspicuous, which may cause reading difficulty. Please use another color like black.

Fig. 2n: The font styles of "Look upward" and "Blink" should be consistent with other parts.

The icons in Fig. 3 and Fig. 4 should be placed inside the figure (just like Fig. 3b).

Fig. 3b and h: the font size is slightly smaller and should remain the same as the other parts. the icons should be placed inside the figure.

Extended Data Fig. 2: all text in Extended Data Fig. 2a-l should not be underlined.

Extended Data Fig. 3a: the font size is slightly bigger; it should remain the same size as the other parts.

Extended Data Fig. 4: As with the previous review comment 25, all the icons should be placed inside the figure. Label g, h, and i should be placed at the top left of a figure.

25. Overall, the manuscript language and grammar could be improved. There are numerous misplaced prepositions (e.g. "in" in line 55, lack of "the" in front of "exercise" in line 21, "higher" in line 78), incorrect spelling (line 105), and confusing sentence structures throughout the manuscript (e.g. the sentence of lines 34-38, 42-46, 53-56). Please seek detailed editing to ensure that the quality of writing reflects the quality of this work.

Reviewer #2 (Report for the authors (Required)):

Summary:

Ear EEG is an exciting and emerging wearable neural technology. This work integrates wet electrochemical and electrophysiological sensors onto a novel flexible earpiece. The earpiece is combined with a DAQ system that includes a commercial off the shelf chip and combines EEG, ASSR, EDA, EOG and eChem recording. Prior art combines electrophysiological recordings, therefore the main contribution of this work is in the integration of electrochemical measures in the ear, including a novel earpiece fabrication process. Verification performs sequential/time-multiplexed electrochemical and electrophysiological sensing on 3 user subjects in an exercise task in addition to separate electrophysiological recordings and verifications.

Major technical criticisms/questions:

1. The introduction provides little to no background or motivation for this integrated device.

a. You should redefine and describe acronyms in the body of the text. In particular, you only mention EDA in

the abstract and never discuss it in terms of findings.

b. Lines 17-20. You claim to simultaneously monitor EEG, ASSR, EOG, and EDA. ASSR is typically a cortical response and is considered evoked EEG. EOG is also considered an artifact that appears over EEG and therefore is not a separable metric. It's also unclear if EEG, EOG, and EDA are truly recorded simultaneously. Reported measurements are time-multiplexed, but details on the acquisition are not provided. Are they all continuous? Do you have data that shows their continuous, simultaneous acquisition?

c. Lines 25-56. There is little/no background or introduction that would be important for understanding the work as a whole. What are existing methods of metabolite monitoring on the skin (generically saying tomography imaging is too broad)? Are there commercial or widespread devices that perform metabolite sensing on skin? What are common targets for metabolite sensing? How accurate are existing metabolite sensing platforms? EEG is similarly left unexplained. Are you claiming that in ear EEG will be just as good as scalp EEG? That is implicit in the current introduction which is wrong because no data suggests in ear EEG provides the same coverage/SNR as scalp EEG.

d. Lines 38-40. From my understanding, the density of sweat glands is highly variable around the human and thus different parts of the human body will have different sweat responses [1]. Is the ear canal a better or worse place to monitor sweat than other parts of the body?

e. Line 47. What is the reasoning for going inside the ear? Is the claim that the ear will provide better SNR than other locations for integrated sensors? Is the claim that it's more discreet? Is the claim that the ear provides a trade-off between EEG, heart rate, and blood ox sensing?

f. Line 68. Isn't lactate primarily a physical stress marker (from exercise)? Or is it also a marker of emotional and psychological stress as well?

g. Line 98. Does this hydrogel require periodic reapplication? What would have happened if no hydrogel was applied (doesn't have to go in the introduction).

h. Was the sweat study (with the ecoflex earpiece) performed only on a single subject? Is that enough?

i. Line 103. Does this mean the ePhys electrodes have hydrogel as well? Were there concerns of 'bridging' (where electrodes are shorted by hydrogel) between all the different sensors?

j. Line 108. More information should be given on the DAQ. How much power does it dissipate? What is the data acquisition rate, datarate, and how does it perform multiplexing? (apologies if I missed this somewhere).

k. Line 129. In ear electrode characterization results are unclear.

i. Impedance/area claim is unclear. This work's electrodes are indeed smaller but this should be more clearly stated. Furthermore, you should state the area of reference 18's electrodes.

ii. Population size and number of averaged trials is not stated

iii. Figure 2 shows a 2-minute settling time. This seems very long and is not discussed. Is this comparable to the state of the art? Is there a reason? Was this the case across all users? Did users have to wait several minutes before starting a trial? How many users? How many trials?

iv. Are the in-ear impedance measurements normally distributed? How is the standard deviation region calculated?

v. Reported EDO is a positive number but figure 2f makes it seem like the mean should be negative? Was this a typo? Was a negative dropped?

l. Line 160. Are there any quantitative metrics to compare the in-ear eChem sensors to the commercial blood sensor? I imagine the blood sensor will be more accurate but is there a way to compare measured trends quantitatively?

m. Line 168. 4% is fairly significant drift for only 10 CA scans. What is the expected drift over longer periods of time? What is required for a given application and how much error can be tolerated? Does the device need to be recalibrated? What is the known lifetime?

n. Line 199. Why are the pre/post exercise Alpha experiments different? It is best not to compare two different experimental methods and not address the reasons for doing so? Was no alpha measured in the first eyes open/closed session?

o. Line 222. What is this analysis and what does it mean that they have different cognitive states? Was there a control experiment performed? Alpha modulation can vary across 30 minutes and hyperventilating. Table s1 should be moved to the results because otherwise your primary claims of being able to do in ear EEG to show the change in cognitive states aren't supported by the main manuscript. Even with the table it is unclear if it's a fair comparison given the differences in experimental procedure before and after exercise.

p. Line 232. It looks like the devices are modeled on Apple airpods pro with afterparty ear wings. These ear wings usually come in multiple sizes (small, medium, large). Were all of these experiments performed with one size of earpiece? If so, does that mean this device is truly user-generic?

q. Figure captions must clearly state how many subjects were involved in the experiment and whether their data is plotted separately or averaged. For example are parts 3 d-g single user data or are they averaged?

General comments:

This work provides a first step to building a new class on integrated electrochemical and electrophysiological sensors. The earpiece in this design is interesting but requires wet hydrogel electrodes, which are impractical and not user-friendly (no one wants something wet in their ear). Thus, why would this feature be added to a headphone rather than a more comfortable patch system elsewhere on the body? The data acquisition system is also large and cumbersome. The introduction/background should properly motivate the uses of such a device and the motivation for this combination, which is unclear in the manuscript. Since this device is used in an exercise task, no mention is made of motion artifacts and how to mitigate their effects in a real application. Furthermore, the results and discussion do not state what such an integrated device could enable. Does it provide better performance over alternatives? Is there new utility that is provided from this form factor?

Minor technical criticisms/questions:

1. Line 140. Did all users use the same earpiece?
2. Line 169. Please define CA, it's only defined in the figure caption. Please also be sure that all acronyms are defined throughout the text.
3. In the sweat detection video – why is there such a steep drop off in the CA curve? Shouldn't there be a slightly more gradual response?

Missing details regarding statistics:

1. In ear electrode impedance measurements lack important statistical information
2. Lactate sensing also seems to lack population size

Missing citations:

Optional suggestions for improvement:

1. Rewrite the introduction to motivate the work and explain more in depth applications of such an integrated device.
2. Perform simultaneous EEG and EDA measurement.
3. Discuss the impact of motion artifacts in the experiment.
4. Provide quantitative metrics for lactate sensing if possible.
5. Add a conclusion to tie up the findings and reconnect it to the motivation of your work.

Stylistic issues:

1. Figure 1 is a bit hard to follow visually (which I realize may be due to the figure count limitation). Would it be possible to place boxes to at least visually separate disparate parts of the figure? For example, a box could be placed around a-d, a separate box around f, and a third box around g-j. Just a thought
2. A more traditional 2D fabrication process guide may be more intelligible than the 3D one currently in Figure 1. It would also be significantly more space efficient.

References:

[1] Baker LB. Physiology of sweat gland function: The roles of sweating and sweat composition in human health. *Temperature (Austin)*. 2019 Jul 17;6(3):211-259. doi: 10.1080/23328940.2019.1632145

Sat 03 Jun 2023

Decision on Article nBME-22-1647B

Dear Dr Cauwenberghs,

Thank you for your revised manuscript, "Unobtrusive In-Ear Integrated Physiological and Metabolic Sensors for Continuous Brain-Body Activity Monitoring", which has been seen by the original reviewers. In their reports, which you will find at the end of this message, you will see that the reviewers acknowledge the improvements to the work and raise a few additional technical questions and suggestions that should help you improve the discussion and reporting quality of the work.

As before, when you are ready to resubmit your manuscript, please upload the revised files, a point-by-point rebuttal to the comments from all reviewers, the reporting summary, and a cover letter that explains the main improvements included in the revision and responds to any points highlighted in this decision.

We look forward to receive a further revised version of the work. Please do not hesitate to contact me should you have any questions.

Best wishes,

Pep

Pep Pàmies
Chief Editor, Nature Biomedical Engineering

Reviewer #1 (Report for the authors (Required)):

I highly appreciate the authors' comprehensive response that provides detailed insights into highlight the unique strengths, breakthroughs, and creativity of the work. The detailed comparative analysis of the reported methodology with previous approaches is appreciated by the reviewer. It is a great pleasure to contribute to the refinement and improvement of such an interesting study. However, to further improve the clarity and depth of the current manuscript, there are a few additional questions and concerns that needs to be addressed:

- 1.The discussion on the clinical relevance of combined brain state and metabolite monitoring was insightful. However, can the authors provide any direct, experimental evidence from the work that supports the synergistic effect of these two modalities in diagnosing disorders?
- 2.The authors have thoroughly described the advantages of the ear as a site for integrated sensing, from its anatomical stability and proximity to the brain to its high density of blood vessels and sweat glands. Regarding the combination of brain state and metabolic monitoring, I found your argument compelling. Can the authors elaborate on the trade-offs, if any, of combining electrophysiological and electrochemical sensing in your in-ear sensors, given the ear's complex and enclosed geometry? For example, does the addition of one sensing modality negatively impact the performance or accuracy of the other in any way?
- 3.Although the authors have mentioned the sizes (small, medium, large) of the earphone silicone tip, could the authors specify the range of sizes these categories cover? This will help to understand if there are any limitations in the user base due to size restrictions. In the experiments, did the authors find any differences in sensor readings based on the size of the earphone silicone tip used? If so, how could the author account for these differences in your analysis?
- 4.The design integrates a sensor into an earbud, presumably to be used in conjunction with a personal audio device. It's good to see that the PVA-hydrogel interfacing to the eChem sensor provides a spongy microporous rough surface to help stability. But are there any effects? Have the authors done any testing to ensure that the added sensor does not interfere with the audio quality of the earbuds?

- 5.Regarding the supplementary video, is it possible to add a quantitative measure, such as a stability index, to show the firmness of the sensor in place during intense physical motion? This would lend more credibility to the demonstration.
- 6.The authors have mentioned that the DAQ was attached to the subject's collar. Was this the most effective position for data collection? Was comfort or interference ever an issue with this setup?
- 7.The authors mentioned using a silicone hook for attaching the sensors and flex PCB tightly to the earphone, but also noted that it could be eliminated in future designs. Could you elaborate on the reasons why you consider this improvement, and what alternatives are you contemplating?
- 8.It's good to see that the authors are considering industrial consortiums like NEXTFLEX for scaling up solutions. While the large-scale production feasibility is addressed in theory, have any practical tests been performed yet to confirm the success of this automated process? The authors mention using 3D-printed custom molds for future production. Could the authors provide more detail on how this would work and any potential benefits or challenges you foresee? How will the automation process maintain the customization aspect of the sensor, particularly in relation to the variations in users' ear sizes and shapes?
- 9.The authors mentioned that the impedance-checking procedure is typical for EEG instruments. While this makes sense for professionals used to handling EEG equipment, how would a layperson, an everyday user, navigate this procedure? Do the authors anticipate developing a user-friendly interface or guide for this?
- 10.It's great to see the questionnaire results showing high ratings for usability and wearability. Can the authors provide more details about the six categories used for assessment? Were there any outliers or common issues identified in the questionnaire results that could lead to improvements in future iterations of the sensor design?

Reviewer #2 (Report for the authors (Required)):

Thank you for your updated manuscript, it is substantially improved with updated experimental results to showcase a significantly more compelling piece of work. Some minor comments are written below.

The introduction much more clearly describes the different modalities and benefits of integrating multiple physiological modalities into a single device. The authors mention that while both EEG and chemical sensing can both monitor cognitive changes. EEG is better suited for acute neuromodulation, rehabilitation, and brain-machine interfacing while metabolite monitoring can provide insight into longer term changes. It stands to reason that having both measures can help account for day-to-day variation in user-specific EEGs (a problem for many epilepsy monitoring applications) and provide vital training data for different machine learning algorithms. I would suggest the authors add a sentence around lines 33-45 to specify a precise example for how these measures can be used to solve a current problem as opposed to 'generally providing better monitoring solutions'.

I appreciate the additional writing that highlights specific issues with commercial metabolic health monitoring solutions (invasiveness or bulky optical equipment requirements) and state of the art wearable platforms (cross talk/bridging + location requirements that negate possibilities for better sensor fusion). I would consider adding a point around lines 73 - 90 that specifies that minimized crosstalk is a benefit of this work over the state of the art. I realize it comes in the subsequent paragraph, but is phrased as just a design parameter as opposed to a system-level improvement over existing solutions.

The adjustments to the pre/post exercise experimental procedure make for a more compelling demonstration.

I appreciate the difficulty in training subjects/redoing analysis for the full EEG & EEG + Lactate trials, but electrode-ear impedance characterization is a straight forward experiment. Having six trials from two subjects isn't much of an average. At the very least the plots should be labeled with (n = 6) so that readers are aware of the small number of trials. The DC offset characterization plots should also be labelled with an (n = 96) - which is a much more useful amount of samples. Future works should prioritise more in-ear

characterization trials across more than 2 users.

I'm a little confused by Fig 3 H & J. After staring at it for a while (and reading lines 207-225), I recognize that you are plotting the change in sweat lactate measurements in each trial relative to the 'ground truth' blood-lactate meter. What the plot makes it seem like is that there is no change in sweat-lactate after enzyme modification? I think it's a scaling issue (there is currently some change across the 40 min trial - but it's less than 0.25 uA after you've improved the sensors...). S11 is significantly more clear than fig 3j. Is there a reason that isn't a main figure? I would suggest double checking the plotting code and addressing any possible confusion about the lack of change in sweat lactate around lines 207-225.

Thu 06 Jul 2023

Decision on Article nBME-22-1647C

Dear Dr Cauwenberghs,

Thank you for your revised manuscript, "Unobtrusive In-Ear Integrated Physiological and Metabolic Sensors for Continuous Brain-Body Activity Monitoring". Having consulted with Reviewers #1 and #2 (whose comments you will find at the end of this message), I am pleased to write that we shall be happy to publish the manuscript in *Nature Biomedical Engineering*.

We will be performing detailed checks on your manuscript, and in due course will send you a checklist detailing our editorial and formatting requirements. You will need to follow these instructions before you upload the final manuscript files.

Best wishes,

Pep

Pep Pàmies
Chief Editor, Nature Biomedical Engineering

Reviewer #1 (Report for the authors (Required)):

Based on the authors' response to my previous comments, it is clear that they have adequately addressed all of my concerns. They have provided additional information and evidence to support their claims and have demonstrated that the integration of multiple sensing modalities in the ear does not negatively impact the performance or accuracy of each other. They have also addressed the issue of earphone tip sizes and ensured that the sensor readings remain consistent across different sizes. The authors have taken measures to preserve the audio quality of the earbuds and have shown that the addition of the sensor does not interfere with the sound pathway. Furthermore, the authors have provided a video demonstrating the mechanical stability of the sensor during intense physical motion and have included a stability index to enhance the credibility of their demonstration. They have explained the reasons for using the current DAQ position and have indicated that future developments will allow for greater flexibility in mounting positions. The authors have successfully addressed all my concerns and have presented a well-designed and promising integrated sensing system for in-ear monitoring. Based on the authors' comprehensive response and the evidence provided, I only have several very minor suggestions regarding the figures in the current manuscript:

1. Ensure that the legends are placed within the respective areas of the labels. This will help readers easily identify which legends correspond to each label. In Figure 2, the legends "ijkl" extend beyond the label area, causing confusion regarding their association with the corresponding figure. Similarly, in Figure 3, the cartoon patterns should be placed within the corresponding label areas.
2. Removal of Meaningless Elements: In Figure 3, it is noted that the dashed lines within subfigure "k" lack meaningful representation. It is suggested to remove these lines unless they serve a specific purpose to avoid any confusion or misleading interpretation.
3. Improving Data Representation in Figure 4: The representation of data in Figure 4, specifically subfigure "f," requires optimization for enhanced readability.

Reviewer #2 (Report for the authors (Required)):

My comments have been adequately addressed.
No further comments.

Rebuttal 1

Response to referees

We sincerely appreciate the editorial team and reviewers for their constructive comments. Here, we provide point-to-point responses to address each of the comments with explicit reference to the changes made accordingly in the revised manuscript.

Editor Comments

You will see that the reviewers appreciate the work. However, they express concerns about the degree of support for the claims, and provide useful suggestions for improvement. We hope that with significant further work you can address the criticisms and convince the reviewers of the merits of the study. In particular, we would expect that a revised version of the manuscript provides: Improved design of the device, towards improved usability, robustness and user-friendliness.

We have made substantial changes in the revised manuscript conducting several rounds of fabrication of new devices for improved usability and robustness, and performed an array of experiments with several new subjects to more comprehensively validate the sensors, in addition to many other substantive improvements to the manuscript addressing the reviewers' comments below.

Validation of the performance of the device, in particular of simultaneous measurements and via relevant physiologically relevant quantitative metrics, in a larger number of individuals.

The protocol for device handling including the ear insertion and removal, cleaning, and storage have been revisited in order to optimize user comfort (further evaluated by a questionnaire), maintain consistency between measurements, minimize wear and tear on the device over repeated use, and streamline cleaning and storage steps.

We have added several performance validation experiments for the device, as listed below:

- Verification on mechanical stability of the device in the ear.
- Longer chronoamperometric scanning of the eChem sensors to verify long-term stability.
- Component analysis of the device to demonstrate safety for skin contact.
- Characterization of the device under different temperature and humidity conditions.
- Motion artifact analysis based on the data obtained from the device.
- Simultaneous EEG + EDA, EEG + EOG experiments to verify the multiplexed electrophysiological sensing capability.

- Combined EEG + Lactate sensing experiments on 2 more subjects. Updated analysis of the exercise brain state and lactate variations based on all the subject data.

Specific details are provided in subsequent responses to reviewer comments below.

Improved background context, and extended discussion of the most promising applications and of the current limitations of the integrated device.

We have updated the introduction section and have extended the discussion of the applications and limitations of the device. We highlighted the unique breakthrough of this work, which is to report for the first time integrated electrophysiological and electrochemical sensing inside the ear.

For the promising applications:

We listed broadly a few applications of this work: early disease detection, health monitoring, body performance improvement, and virtual/augmented reality.

- We expanded our discussion on the role of the device on neurodegenerative diseases. Since clinical evidence has shown that diseases such as epilepsy and Alzheimer's disease can trigger characteristic patterns in both electrophysiological brain state monitoring and produce abnormal metabolic profiles in an individual. We have then highlighted such application in the Abstract "Such simultaneous and continuous unobtrusive monitoring of brain and body biomarkers permits observing their dynamic and synergic interactions in highly mobile settings, significantly expanding the functionality of the aural device for long-term neurodegenerative disease detection, daily health monitoring, and beyond".
- As for the limitations, we explicitly included in the revised manuscript discussion on:
 - The feasibility of mass production in updated manual fabrication procedures.
 - The power consumption and the size of the sensing system. Specifically, a discussion on how these limitations can be overcome by further sensor-electronic integration, namely with the incorporation on-going of low-power IC design efforts in the near future.

Thorough methodological details.

We have thoroughly reviewed and improved the Methods section to cover several unclear sections mentioned by the reviewers.

The following sections have been added to the Methods section:

- Sweat mapping methods inside the ear.
- Subject-specific tight fitting inside the ear
- PVA-gel characterization for skin safety

The following sections have been revised for clarity in the Methods section:

- Added clarity and reference to Extended Data Fig. 2 in the “Sensor fabrication, electrode modification, and earphone assembly”.
- Improved clarity on ePhys electrode DC offset characterization methods.
- Additional description for the robustness testing in the eChem in-vitro characterization methods.
- Additional details for eChem on-body characterization methods.
- Reorganized method description for the combined EEG and lactate sensing to expand on the how motion artifacts were avoided, and further analysis based on the statistics across all subjects.

The following sections have been added to the Supplementary Information to strengthen degree of support for the claims:

- Methods and results of the sensors’ chemical component analysis to support the device’s use on the skin safely.
- Methods and results of the sensors’ robustness under different temperatures and humidity to support the robustness of the device.
- Methods and results of validating the ePhys sensors’ capability to perform simultaneous electrophysiological sensing.
- Methods and results of motion artifact reduction based on the in-ear ePhys sensors data input.
- Methods and results of the questionnaire to evaluate usability and wearability.

Clearly highlight any amendments to the text and figures to help the reviewers and editors find and understand the changes (yet keep in mind that excessive marking can hinder readability).

We have provided a version with all changes clearly tracked and another *clean* version of the manuscript, supplementary information.

If you and your co-authors disagree with a criticism, provide the arguments to the reviewer (optionally, indicate the relevant points in the cover letter). If a criticism or suggestion is not addressed, please indicate so in the rebuttal to the reviewer comments and explain the reason(s).

We agree with all the instructive comments from the reviewers and have made our best efforts to address every comment.

Reviewer 1 commented that “Please use the in-ear sensor to collect and compare the physiological indicators of patients and healthy people, which will further prove the potential and value of the in-ear sensor in biomedicine”. We agree with the reviewer’s comments on demonstrating the potential BCI applications of the current

study. This would add a lot of weight to this work for sure. However, we may think that specific applications of EOG are not too closely aligned with the main focus of this work: integrated electrophysiological and electrochemical sensing in the ear. We would like to add here, the application of EOG and EEG has been established in the research field including drowsiness detection¹, eye vergence therapy², and motor control^{3,4}, which we have described in the manuscript. These can be valuable future directions of the applications of our in-ear sensors.

Reviewer 2 commented that “Provide quantitative metrics for lactate sensing if possible.” We appreciate the suggestion, in this work, blood lactate was used as validation however not as quantitative metrics. Based on the control assessment (Fig. 3h), there is a strong correlation of the blood lactate and the sweat lactate (n=3).

Consider including responses to any criticisms raised by more than one reviewer at the beginning of the rebuttal, in a section addressed to all reviewers.

We have addressed several common criticisms raised by both reviewers at the beginning of the rebuttal. These include:

- Better motivation of the work.
- Why choosing the ear as the sensing location.
- More clarity on the in-ear sweat mapping
- Repetitive chronoamperometry scans to demonstrate the robustness of the eChem sensors.

The rebuttal should include the reviewer comments in point-by-point format (please note that we provide all reviewers will the reports as they appear at the end of this message). Provide the rebuttal to the reviewer comments and the cover letter as separate files.

In the rebuttal, we have addressed the referees’ and editors’ comments in a point-by-point manner.

Response to all reviewers

1. Reviewer 1: Ingenuity compared to previous work? Both electrophysiological brain status monitoring (IEEE Trans. Biomed. Circuits Syst. 2020, 14(4), 727) and health-related metabolite monitoring (Sci. Robot. 2020, 5(41), eaaz7946) (Nat. Biomed. Eng. 2021, 5, 737) dimensions have been reported before. Among these reported research, electrophysiological brain state monitoring has been integrated into sensors of similar size to monitor physiological signals. The chemical monitoring of metabolites has also achieved the monitoring of biomarkers such as lactate through electronic skin (e-skin) and patches. Please compare this work with previous work and explain the unique strengths, breakthroughs, and ingenuity of this work.

Reviewer 1: Why is there a need to combine both in the first place? If both metabolites and brain state demonstrate the same thing, why do we need to measure it twice? They need more background on each modality to argue this.

Reviewer 2: The introduction provides little to no background or motivation for this integrated device.

Reviewer 2: Line 47. What is the reasoning for going inside the ear? Is the claim that the ear will provide better SNR than other locations for integrated sensors? Is the claim that it's more discreet? Is the claim that the ear provides a trade-off between EEG, heart rate, and blood ox sensing?

The unique breakthrough of this work is to report for the first time integrated electrophysiological and electrochemical sensing inside the ear. We would like to explain the significance of such an effort from three aspects:

How our efforts compare with previous electrophysiological sensing approaches:

This work took a systematic approach to extend the scope of previous works on in-ear electrophysiological sensing, to be combined with electrochemical sensing all inside the ear canal.

Indeed, there have been an increasing number of reports of electrophysiological sensing in the ear, mostly focusing on ear-EEG^{5,6} with applications based on it^{7,8}. A significant portion of ear-EEG research is focused on the feasibility analysis of recording EEG from the ear, as well as the application of ear-EEG devices for scenarios including stress monitoring, epilepsy monitoring, sleep staging, etc⁹. Among the sensor or system design approaches in the ear, the main goal was to conduct electrophysiological measurements, especially in-ear EEG. With that goal, established previous in-ear sensors took the form of the custom-fit earpiece, generic earpiece, or electrode grid behind the ear¹⁰.

This work built an electrophysiological sensing system that differs from any previous approach. Among other novel approaches, this study employs flexible 3D Ag/AgCl electrodes with a geometry designed to coexist with three other electrodes for electrochemical lactate sensing, as well as a device structure mapped to a commercial earphone. While the methods of this work extend the capabilities of in-ear sensors beyond those previously published (Table S1), the electrophysiological

performance of the sensors in this study was at least comparable to ear-EEG methods described in the literature.

How our efforts compare with previous electrochemical sensing approaches:

Chemical monitoring has also been demonstrated as e-skin and patches with the capability of functioning as non-invasive sensing tools¹¹. Still, despite major advantages shown with such platforms, the demonstration of their performance has been limited to body parts such as the arm, wrist, and back, where the simultaneous monitoring of the EEG along with biochemical biomarkers is not feasible, while this work demonstrated the first electrochemical sensing of the ear canal, which has a complex and enclosed geometry. This development has made the sensors completely concealable, allowing for more discreet and comfortable health monitoring. Also, this study marks the first time that electrochemical sensing and electrophysiology sensing were conducted simultaneously in the ear for multimodal health monitoring.

Why integrated sensing in the ear is an innovative approach:

The ear is an important human organ that hosts rich possibilities for wearable sensing. Compared with electrophysiological and electrochemical sensing at conventional sites on the body it offers the following advantages:

- It is relatively stable and resistant to motion artifacts in the signals for both modalities, which is supported by the natural mechanical anchoring points to stabilize the sensor during movement.
- It is located in a very unique position in the human body, close to the brain (especially the auditory cortex), and has blood vessels and sweat glands at high density in close proximity¹². Such property makes it the most suitable sensing location for electrophysiological (brain) and electrochemical (on-body) sensing in an unobtrusive simultaneous manner.
- The sensing location is highly concealable and widely accepted by users, owing to the prevalent use of hearing aids, earphones, and other similar auditory devices in modern society.

Comparison with previous multi-modal sensing approaches in the ear:

Multi-modal sensing in the ear has been demonstrated. Previous approaches include integrating off-the-shelf sensors such as SpO₂ sensors, and temperature sensors for such targets. However, the electrochemical side of the ear, especially in combination with brain signal monitoring, has rarely been explored. Combined brain state and metabolite monitoring covers a greater range of real-life applications owing to the orthogonality of electrophysiological measurement and electrochemical measurement targeting a different set of human biomarkers. Despite the rich electrophysiological and biochemical skin processes (i.e. apocrine glands, eccrine glands, sebaceous glands, etc.), reports of sensing of biomarkers in the auditory canal using electrochemical detection have been very limited^{13,14}.

Why there is a need to combine brain state and metabolite monitoring:

There is a need to combine the brain state and metabolic monitoring. The brain state and metabolic monitoring has different implications on the conditions of the body functioning, while combining them provides a more comprehensive indication of the human body functioning.

Monitoring the brain's condition, particularly using EEG, was the approach widely used to investigate brain activity. Due to the brain's widespread involvement in almost all real-world human actions, brain status monitoring has proved very useful in several ways. EEG has not only been utilized for the diagnosis of diseases such as epilepsy, seizure, etc. but also for practical applications such as rehabilitation engineering, cognitive state classifications, mental state monitoring, etc.

Monitoring critical metabolic biomarkers gives real-time analytical data on the dynamically changing health state of humans. Important biophysical and biochemical characteristics, these biomarkers are identified as an indication of certain biological, physiological, or pathological processes or as pharmacological reactions to therapeutic treatments.

In addition to their individual signals of brain and body function from brain state and metabolite monitoring, it has been shown that the combination of these two distinct modalities may have a synergistic effect on the diagnosis of several disorders. Stress and emotions, as well as neurodegenerative disorders such as epilepsy and Alzheimer's disease^{15,16}, may generate specific patterns in both electrophysiological brain state monitoring and aberrant metabolic profiles.

Clinical evidence showing brain state and metabolite monitoring are synergistic:

EEG + sweat monitoring, which correlates to the variations of lactate, several studies touched upon the synergistic applications of EEG and sweat measurement to monitor stress and tension^{17,18}, seizure and epilepsy^{15,19}, and Alzheimer's disease¹⁶. These disorders can trigger characteristic patterns in both electrophysiological brain state monitoring and produce abnormal metabolic profiles in an individual.

Additionally, the role of EEG and lactate monitoring for the early detection of several disorders including stress, epilepsy, and Alzheimer's disease have been demonstrated from isolated clinical studies with individual sensing modalities. For EEG, EEG has been widely used as the original dataset to further investigate neuro-disorders including depression²⁰, epilepsy²¹, Alzheimer's disease²², and beyond. With further signal processing and classification algorithm, modern EEG-based systems can achieve more than 90% detection accuracies for the abovementioned neuro-disorders. For lactate, recent studies have revealed the variation of lactate correlated to physical exercise and its conducive role to improve brain function^{23,24}, such a mechanism helps the treatment of neurodegenerative disorders including the Alzheimer's disease. In another study²⁵, the authors found that "Insufficient lactic acid production results in an inadequate neuronal energy supply, which

affects normal physiological responses and results in brain dysfunction. Conversely, the buildup of lactic acid can lead to abnormal activity in brain areas that cause lactic acid to rise, which leads to brain dysfunction". Therefore, an in-depth understanding of the molecular mechanisms by which lactic acid regulates brain function is of great value for the early diagnosis and prevention of neuropsychiatric diseases". These clinical requirements are among the primary motivations for this study. As a summary, these isolated studies with EEG and lactate as biomarkers for different neuro-disorders revealed that:

- EEG signals reflect the electrical activities of brain behaviors while such electrical activities are directly related to the disorders of neuron activities in the brain.
- Lactate is related to vascular endothelial growth factor (VEGF) and brain-derived neurotrophic factor (BDNF) expression and, by regulating the function of hippocampal mitochondria, can improve brain functions such as angiogenesis and neuroplasticity as well as stress-related symptoms such as depression. As Alzheimer's disease preventative and to aid individuals with brain function issues and the elderly, lactate-related compounds will be in the limelight. If lactate levels, which can be regulated via exercise, are well-controlled, it will become an area of interest for future healthcare studies.

In summary, the sensors presented in our manuscript achieved combined electrophysiological and electrochemical sensing completely inside the ear, which enabled integration with a ubiquitous earphone platform. Apart from sensing, we have contributed several design innovations to ensure a mechanically stable interface between the sensors and the ear skin, such as the PVA-hydrogel. Placed on the top of the eChem sensor, between the ear canal and the earplug, this PVA-hydrogel offers not only spongy interfacial connections to fasten the earplug in the canal but also offers the sweat sample collection by incorporating hydrophilic porous membranes. This allowed the continuous collection of chemical sensing data for 40 min during physical exercise. Hence the data collected was not disturbed by the collection of electrophysiological brain data.

We have updated the introduction section of the manuscript accordingly, including additional references mentioned by the reviewer. Manuscript lines 33–45 discusses the motivation to combine the two sensing modalities, lines 46–61 discusses the challenges for combining brain state and health-related metabolic sensing, and then lines 62–72 described why in-ear sensing was the optimal solution. Overall, we highlighted the importance of combined brain-state and metabolite sensing and introduced the ear as a uniquely fit location on the body surface to sense these two modalities simultaneously, while also addressing the challenges for integrated sensing in the ear.

2. Reviewer 1: It seems that the placement of eChem electrodes towards the tragus is based solely on the sweat profile of a single volunteer, as detailed in Extended

Data Fig. 1, Fig. 1f, and lines 81-84. Are the in-ear sweat profiles of different people different, which would make the sensor less effective for others? If not, is this backed up by either a) the author's experimental data or b) previous literature?

Reviewer 2: Was the sweat study (with the ecoflex earpiece) performed only on a single subject? Is that enough?

To find the optimal location for placement of the electrochemical sensors to perform sweat analysis in a subject independent manner, we evaluated the areas of elevated sweat secretion after physical exercise by applying custom Ecoflex pieces in the ear of three subjects. As shown in Revision Figure 1 **Error! Reference source not found.** (Fig. S1 in the supplementary information), we consistently observed the areas with the highest sweat volumes were found toward the tragus area of the ear channel across all three subjects. We did not encounter any challenges in sweat collection and sensing lactate during any of our experiments using the electrochemical sensors in the tragus area.

Revision Figure 1. Sweat profiles on 3 different volunteers. The spreading of commercial dye deposited on traces of filter paper pieces was evaluated on 3 healthy individuals (a-c). The left side of the figure presents the filter paper pieces before 30 mins of exercise. The right side of the figure shows the spreading of the blue dye after

exercise.

Accordingly, we have added a new section about sweat mapping in the method section (manuscript lines 309–318).

3. Reviewer 1: Lines 168-171 states that “running 10 repetitive CA scans at a fixed concentration displayed in Fig. 3c, showed minimal changes (<4%), demonstrating the efficient entrapment of the enzymatic layer on the eChem transducer.” The device only run for 10 CA scans, which is very little compared to what is expected of continuous monitoring. Additionally, how long is considered one “scan”?

Reviewer 2: Line 168. 4% is fairly significant drift for only 10 CA scans. What is the expected drift over longer periods of time? What is required for a given application and how many errors can be tolerated? Does the device need to be recalibrated? What is the known lifetime?

The stability tests were performed every 10 minutes, which sums up sensor stability of the 100 minutes and confirms sensor probe development is robust and stable even after the repeated removal of the PVA-gel from it. The drift below 5% is acceptable and shows the sensors’ stability for the application in this work. Accordingly, we have added extended experiments to further demonstrate sensors’ stability, 18 repetitive cycles were run to check for stability, and we observed similar responses (< 5%) (Fig 3c) validating the sensors’ stability for approximately 180 mins, which is sufficient for each in-ear usage. The sensors do not require recalibration during use and have been tested for 180 mins. The operational lifetime of sensors using similar design approaches can be as long as 8 hours which have been demonstrated on our previous work²⁶.

Reviewer #1

In this paper, the authors present a pioneering integration of two dimensions of electrophysiological brain state monitoring and health-related metabolite monitoring into an unobtrusive wearable sensor. It makes full use of the extremely limited space in the ear canal and provides an elegant solution for user monitoring. The authors solved the problem of measurement signal interference and optimal synergy between sensors with different sensing methods. This work breaks through traditional challenges such as sensors' size and area limitations and achieves brain state and metabolite monitoring unobtrusively. Overall, this manuscript demonstrates an interesting design with advantages over existing state-of-the-art technologies. However, there are several issues that the authors must address to improve the quality and clarity of their work.

We again thank the reviewer for the constructive comments. Accordingly, we have revised the manuscript. Here, we also provide point-to-point responses to address each of the comments.

Major concerns:

1. Ingenuity compared to previous work? Both electrophysiological brain status monitoring (IEEE Trans. Biomed. Circuits Syst. 2020, 14(4), 727) and health-related metabolite monitoring (Sci. Robot. 2020, 5(41), eaaz7946) (Nat. Biomed. Eng. 2021, 5, 737) dimensions have been reported before. Among these reported research, electrophysiological brain state monitoring has been integrated into sensors of similar size to monitor physiological signals. The chemical monitoring of metabolites has also achieved the monitoring of biomarkers such as lactate through electronic skin (e-skin) and patches. Please compare this work with previous work and explain the unique strengths, breakthroughs, and ingenuity of this work.

Please refer to our 1st response in the “Response to all reviewers” section. We provided comparison to previous work and the motivation of this work.

2. How to fit users' different ear canals? Due to the extremely limited space in the ear and the large variation in anatomy between the ears of different users (lines 51-52), which can lead to a geometric mismatch between the in-ear sensor and the ear canal (lines 95-96). The authors proposed the solution by applying the ePhys stretchable Ag electrode with a three-dimensional structure (lines 97-99). However, the solution principle for solving the adaptation problem is not detailed, please clarify the principle.

Two considerations were made to address the geometrical variation in ear anatomies: 1) The sensors were designed to have an outline profile that closely matched the structural profile of the earphone silicone tip. Such mapping was achieved by flattening the scanned 3D structure of the earphone used in this work. Beyond that, the earphone silicone tip comes in three different sizes (small, medium, and large). For different subjects, a suitable silicone tip size was chosen before

sensor assembly and on-body experiments. By choosing the appropriate earphone silicone tip, an initial tight fitting to the ear canal was achieved. 2) The sensors were further designed to have 3D ePhys structures as well as a PVA gel covering the eChem sensors. Both structures used soft materials, serving as cushioning layers between the already tight-fitted ear canal-sensor interface, further achieving an interference-fit.

From our experiments, the customized-size earphone tip plus cushioning morphology for each volunteer obtained a mechanically stable interface between the ear and the sensors, and no incidence of loose contact or slip-off was observed with the sensors in the ear.

Accordingly, we have added a paragraph on design considerations to provide tight contact in the ear and prevent sensors from slipping off (manuscript lines 346–363). In the supplementary information Note 1, we further provided a detailed description of how the 3D ePhys and PVA gel were prepared. We have also recorded a supplementary video to show how the in-ear sensors along with the earphones stay firmly in place during intense physical motion.

3. It seems that the placement of eChem electrodes towards the tragus is based solely on the sweat profile of a single volunteer, as detailed in Extended Data Fig. 1, Fig. 1f, and lines 81-84. Are the in-ear sweat profiles of different people different, which would make the sensor less effective for others? If not, is this backed up by either a) the author's experimental data or b) previous literature?

Thanks for pointing this out, it is backed up by our experimental data. Please refer to our 2nd response in the "Response to all reviewers" section.

4. Will the in-ear sensor slip off? The sensor is primarily made from TPU and SEBS, both of which are hydrophobic, and the two surfaces are very smooth. Does this hydrophobic and smooth material surface cause any effect on the in-ear sensor if it conforms to the skin over time? In practice, will the sweat of the exercise cause the in-ear sensor to slip out of the ear canal? Many times, when people are exercising, they enjoy listening to music or podcasts. With the sensor design are the headphones still functional enough for such tasks?

Throughout several on-body testing sessions, we did not observe any sensor detachment. The longest experimental session lasted between 40 and 60 minutes, with the subjects doing continuous activity.

As we described in our response to the comment 2 above, the sensors were first designed to encapsulate a generic earbud with 3 different sizes for different subjects, which itself has already been mechanically stable inside the ear. Then, the sensors also had 3D ePhys electrodes in the current design. This design effectively formed an interference-fit between the sensors and the ear canal. In addition, the PVA-hydrogel interfacing to the eChem sensor offers a spongy

microporous rough surface that helps to provide stable adherence of the ear plug during motion. Additionally, the assembly procedure included a silicone hook attaching the sensors and flex PCB tightly to the earphone. Another effect of such an approach was that the earphone, along with the sensors were further anchored by the auricle. Though noted here the earphone hook can be improved and eliminated in future designs to further reduce the form factor and improve the integration.

Accordingly, we have added a paragraph on design considerations to provide tight contact in the ear and prevent sensors from slipping off (manuscript lines 346–363). We have also recorded a supplementary video to show how the in-ear sensors along with the earphones stay firmly in place during intense physical motion.

5. Can the sensor be produced on a large scale? In this paper, the sensor is customized, designed, and processed according to the area with users' high sweat secretion, including printing and cutting sensor profile preparation. The process is complicated and requires a lot of manpower and resources to provide customized services. Will these processes limit the large-scale production of modified in-ear sensors? Will these complex processes lead to excessive production costs to make an in-ear sensor?

The sensors are capable of being manufactured on a large scale and at a reasonable cost. Screen-printing using stencils was used to fabricate the sensors, which is a low-cost and batch-manufacturing approach. In the reported design, the fabrication of ePhys electrodes, eChem sensors, and the integration of the sensors to the earphones required human labor, while such a design may be automated at the production stage. Indeed, the fabrication of hydrogel, the cutting of Kapton tape, and the modification of the earphone hook were some of the manual procedures employed here for demonstration purposes. In the future, incorporating 3D-printed custom molds, customized spinning machines (e.g. electrospinning and electrospraying tools) and automated cutting machines (e.g. Cricut Explorer or laser cutter) will improve the control and scalability of these fabrication stages. The procedure of attaching the sensors to the earphone may also serve as a fabrication step for the silicone tip of the earphone. It is also worth mentioning that there are industrial consortiums (e.g. NEXTFLEX (<https://www.nextflex.us/>)) that can provide scaling up solutions to wearable research prototype like the one we reported here.

Eventually, the sensors can be manufactured on a big scale and at a low cost. We have included a brief discussion on the potential fabrication improvement in the manuscript.

6. The evaluation of in-ear sensor wearing comfort? In this work, the ePhys stretchable Ag electrode of the in-ear sensor adopts a three-dimensional structure to fit the contour of the ear canal (lines 97-99). However, there is a lack of a quantitative definition of fit. If the fit is low, the sensor will fall off during exercise, and excessive fit will cause the

burden and discomfort of wearing. Therefore, it is necessary to use objective comfort evaluation criteria to measure the sensor to prove comfortable wearing and excellent performance. Please design and provide a quantitative definition of the in-ear sensor fit.

In addition to the design considerations stated in response to the comments 2 and 4 above, we validated the contact between the ePhys electrodes and the ear canal before conducting experiments on the body using impedance as a quantitative metrics.

As described in the Methods section, the subject was instructed to wear the in-ear integrated sensors and the DAQ for over 2 minutes to stabilize the electrode-ear interface and reach a magnitude of less than 1 M Ω at 10 Hz (manuscript lines 404–406). As described in the ePhys sensor validation section (supplementary information lines 353–357), this impedance-checking procedure was also used to characterize the dry contact EEG headset that was used to characterize our in-ear integrated sensors. This preparation procedure is typical for EEG instruments. We would also like to note that such impedance measurement for lead-off detection can be conducted simultaneously with ePhys recordings such as EEG, which means it does not necessarily have to be done before the experiments. The combined impedance and EEG measurement can be found at Supplementary Note 3.

Per the reviewer's suggestion, we conducted further evaluation of the wearing comfort of the sensors. To our best knowledge and throughout literature reviews, direct assessments of wearing comfort for wearable devices especially in-ear devices are rare. A review paper on assessing wearability²⁷ shows that questionnaires and interviews are most commonly used to evaluate wearing comfort. We thus further conducted questionnaires for voluntary wearers with a group size of more than 50. The assessment categories of the questionnaires were adopted from the comfort rating scales (CRSs) reported in previous work²⁸. For all 6 categories of usability and wearability, the sensors in this work recorded a rating of higher than 80 percentile. The evaluation of the wearing comfort has been added to the supplementary information (supplementary information lines 664–698).

7. The authors should likewise describe how the sensor was mounted to different sites, and how the sensor connected to their peripheral electronics for wireless data transmission. The peripheral electronics should also be pictured.

The mounting of the sensors is described in the “Electrophysiological measurement system integration and the on-body setup” and the “Electrochemical lactate sensing” sections in the Methods section. During measurements, the subjects wore the in-ear integrated sensors assembled on earphones. The flexible PCB has adhered to the mastoid and the hindneck. The DAQ was attached to the collar at the subject's back (Extended Data Fig. 7a).

8. Is the double-sided adhesive biocompatible? In this work, the in-ear sensor only verified its mechanical properties in the silicone eardrum (SI, line 174). Whether the long-term use of double-sided adhesive will bring discomfort to the skin needs further biological verification. Furthermore, could it be replaced with a more stable, reliable, and convenient sensor adhesive method?

The double-sided medical tape is a medical-grade product purchased from 3M™ (model 1509). According to their primary irritation tests in the reference below, a piece of double-side adhesive was placed on top of the skin of rabbits for 24 hours. After the time concluded, the skin of the rabbits was examined for 1, 24, 48, and 72 hours to observe any evidence of erythema or edema. It was concluded that there was slight evidence of erythema and no formation of edemas. The Primary Irritation Index for the test article was calculated to be 0.4. The response of the test article was categorized as negligible. In the studies performed in our manuscript, volunteers wore the device for a maximum of 1 hour. Therefore, we would expect a lower primary irritation index. The current adhesive provided enough stability during physical performance. Apart from the abovementioned characteristics of the tape itself, we note that by design, there was no direct contact between the skin and the tape, which was encapsulated between the silicone earphone tip and the sensor substrate. (Reference: https://www.3m.com/3M/en_US/p/d/v000186518/)

For bonding purposes in this work, we did try to use medical grade tape that has been proven for safety. There are recent advances in skin adhesives for wearable applications. We believe our sensors can be integrated with some of these new materials by changing the fabrication methods, which will be future directions of this work.

9. Skin irritation and skin allergy considerations? The sensor is fabricated using toluene as an organic solvent, as well as an alkaline solution like. They are often strongly irritating to the skin and have an unpleasant odor. Does this have any potential dangers to user health when the device adapts to the skin over time? Please supplement a pathological test to certify the safety and ensure wearer comfort.

The sensor patch is assembled with hydrogel on the ear. During the patch fabrication process, the material layers are subsequently heat cured at 80 and 60°C upon the deposition of each layer. For developing the sensor patch, a total of 30 minutes (10+10+10 minutes) of heat treatment has been given, where for the SEBS layer gets approximately 20 minutes of heat exposure (at 80 °C) in the oven, which significantly evaporates the toluene from the patch. In order to verify, we have performed the material characterization using TGA (Revision Figure 2, Fig. S5 in the manuscript), where the SEBS film has been characterized after the curing. TGA curve demonstrates the mass loss at different temperatures. The consistent % weight till 250 °C indicates the SEBS film is devoid of the toluene which has otherwise been seen with mass loss at its boiling point (~110 °C). The mass loss starting from ~250 °C is attributed to the heat degradation of SEBS.

In addition, another skin contact of the gel materials has also been assessed for the KOH contents. The KOH content of the alkaline solution from the hydrogel membranes was removed by performing repetitive washing steps with phosphate buffer (pH 7.0). During every washing cycle, the pH of the solution containing the hydrogel membranes was recorded using commercial pH strips. These washing steps were optimized by examining the KOH content using FTIR spectroscopic methods. (Revision Figure 2, Fig. S5 in the manuscript) The disappearance of the peak at $\sim 670\text{ cm}^{-1}$ confirms the absence of the K-O bond, which corresponds to the absence of the KOH from the hydrogel. In addition, the PVA hydrogel has also been assessed by using the pH measurement at its surface every time before the experimentation, where the neutral pH ensures also corroborates the removal of the KOH irritant from the PVA-hydrogel.

Revision Figure 2. IR spectrum PVA-hydrogel of before and after the wash.

The content of alkaline solution from the hydrogel membranes was removed by performing repetitive washing steps with phosphate buffer (pH 7.0). During every washing cycle, the pH of the solution containing the hydrogel membranes was recorded using commercial pH strips. This washing step was concluded once the pH of the solution reached pH 7.0. The pH 7.0 of the hydrogel confirms the concentration of the KOH is negligible.

Accordingly, we have added a new section on “Sensors’ chemical component analysis” in the supplementary information (supplementary information lines 178–203).

10. Will the in-ear sensor keep firm in a humid environment? During the processing of the sensor, the authors used manual cutting double-sided adhesive to assemble the sensor (254 lines).

Different types of medical-grade adhesives have been tested for the fabrication process. We would like to note here, during the manual cutting step, the liner of the 3M adhesive was not removed. Thus, effectively, the bonding of the sensors to the earphone was after the profile of the in-ear integrated sensors has been cut. We have not experienced any defect or adhesion failure during the fabrication.

Whether will there be a defect where the adhesion is not tight enough during the manual cutting of the double-sided adhesive? In exercise, these double-sided adhesive defects might encounter water (sweat), and the adhesion will decrease, so the fastness of the sensor under different humidity needs to be further verified. Whether will there be a defect where the adhesion is not tight enough during the manual cutting of the double-sided adhesive?

Regarding the breakdown of adhesion due to humidity, we have not seen any such failures during on-body measurements done on various subjects. This was most likely due to two primary factors: 1. As described in response to reviewer's comment 2 and 4, the in-ear sensors were tightly placed in the ear, minimizing adhesion loss due to mechanical instability. 2. In our sweat mapping experiments, we discovered that sweat production in the ear canal was less than in other areas of the body, such as the forehead, chest, and arm. This was one of the reasons why we added hydrophilic PVA hydrogel to the eChem sensors to improve sweat collection. We thus found no evidence that the sweat in the ear would result in adhesion failure. We have also not seen adhesion failures during the in-vitro humidity testing for the ear sensors.

11. Why is there a need to combine both in the first place? If both metabolites and brain state demonstrate the same thing, why do we need to measure it twice? They need more background on each modality to argue this.

Please refer to our 1st response in the "Response to all reviewers" section, we explained the need to integrate both sensing modalities.

Can more biomarkers be monitored? As mentioned in review comment 10, the combination of two dimensions of electrophysiological brain state monitoring and health-related metabolite monitoring is still lacking in sensor comprehensiveness. It can be considered that this in-ear sensor can be developed to monitor more physiological signals, thereby endowing it with higher integration and more comprehensive monitoring performance. For example, a pressure sensor can be integrated into the in-ear sensor to monitor the change of sound pressure in the ear canal to realize the judgment of the source of the sound position (J. Acoust. Soc. Am. 1989, 86, 89).

In this work, we demonstrated synchronous measurement of both modalities in real-time instead of twice that uncover the precise physiological conditions.

It is possible to measure more biomarkers based on the current design framework, by different electrode modification protocols, other eChem biomarkers including metabolites (e.g., glucose), electrolytes (e.g., Na⁺, K⁺, Cl⁻), and substances (e.g., drugs, heavy metals, alcohol) can be monitored.

Adding other physiological sensing modalities is also feasible with moderate design updates to the current design. Most likely, it would involve some degree of customization of the earphone itself to allow for more spatial utilization. Previous arts have already demonstrated the effectiveness of integrating one or several modalities of ECG, BCG PPG, IMU, or sound pressure, etc. as the reviewer mentioned. We have summarized some of these typical attempts in the supplementary table.

12. Could the in-ear sensor keep robust against temperature changes? At present, the authors only test the stability of the sensor under the specific temperature and humidity of the laboratory, but in the actual use scenario, the ear canal is different from the outside temperature, and the temperature fluctuates throughout the year when the sensor is used. Will the difference in thermal expansion coefficients of the TPU and SEBS materials that make up the sensor cause the sensor structure to shrink and expand to varying degrees between layers when the temperature fluctuates? As a result, the device of the in-ear sensor might be damaged. Once it falls off inside the ear canal, the damaged sensor part may fall into the human body along the cochlea, with disastrous consequences. Therefore, it needs to be supplemented within a specific temperature range. The device stability test needs to diversify the test environment and consider the impact of the actual environment.

The in-ear integrated sensors can keep robust against temperature changes. For acute temperature changes, the sensor components were robust even at 80 °C. These have withstood several fabrication steps involving a temperature of 80 °C (each lasting 10 minutes) generating no cracks and specks. Since the material properties have remained intact at the higher temperature, it is most unlikely to happen at the on-body setting at ~36 °C. In addition, the mechanical stretch test has also been performed, which confirmed its mechanical robustness.

In addition, per the reviewer's suggestion, we further characterized the sensors' functionalities in varied temperature (25 °C to 40 °C) and humidity conditions (40% to 70%), where the ePhys sensors have been tested for connectivity, the eChem sensors have been tested for their analytical performance in terms of sensitivity. In the experiments, we expanded the testing condition to cover the reported normal temperature ($36.4 \pm 0.6^{\circ}\text{C}^{29}$) and humidity (relative humidity: 40% - 70%³⁰) profiles inside the ear.

Accordingly, we have demonstrated the mechanical and environmental robustness of the sensors in the Supplementary Note 2 (supplementary information lines 205–

249: mechanical robustness, supplementary information 250–286, temperature and humidity robustness).

13. Lines 168-171 states that “running 10 repetitive CA scans at a fixed concentration displayed in Fig. 3c, showed minimal changes (<4%), demonstrating the efficient entrapment of the enzymatic layer on the eChem transducer.” The device only run for 10 CA scans, which is very little compared to what is expected of continuous monitoring. Additionally, how long is considered one “scan”

We have performed longer CA scans to demonstrate the stability of the eChem sensors in 180 mins, which covers the normal duration of such sensor for one in-ear usage. Please refer to our 3rd response in the “Response to all reviewers” section.

14. The reproducibility of the sweat lactate CA current data. In line 213 of the manuscript, it is stated, “The sweat lactate CA current remained for ~2 mins after the exercise stopped because of the post-exercise sweat residue”. However, according to the corresponding Fig.4c, f, and I, subject 3 showed a CA current retention much more significant than 2 minutes. Therefore, please repeat the verification of the sensor performance to make it stable.

Thanks for pointing it out. We would like to note here the variation between different subjects’ sweat profiles was expected, which was due to the different degrees of exercise intensities, and most importantly the individual’s unique perspiration and the metabolic profiles. Previous work with lactate sensors not in the ear but around the ear has shown similar behavior and reported “with lactate experiencing an abrupt rise at the middle of the indoors cycling event. Moreover, in the 10 min period allowed for recovery, these parameters do not return to their initial levels”¹⁴. In their results, subject 1 and subject 4 have also shown distinct lactate level recovery behavior after the exercise. To further demonstrate how such sweat variation was recorded with our sensors, we have conducted another exercise recording with extended timeframe, here the subject experienced a later sweating onset and also a later retention phase (Fig. S11). We have also added more subjects to repeat the integrated sensing experiments and analyzed the common pattern across subjects (Fig. 4f). As a general pattern, the retention phase and recovery phase of the sweat lactate current are common across subjects.

Accordingly, we have modified the description that reviewer pointed out to:

Across subjects, the sweat lactate CA current remained for a variable duration extending beyond exercise due to post-exercise sweat residue, before recovering at rest after exercise. (manuscript lines 279–281)

15. Lack of support from clinical data. Lines 72-74 of the manuscript mention that “The implementation of both modalities into a miniaturized in-ear non-invasive platform

could thus facilitate the process of using multiple instruments for assessing these features during neurological monitoring and potentially allow self-monitoring in patients.” Here, it is necessary to provide supporting information on monitoring specific physiological disease indicators. The significance of physiological signal monitoring through sensors will be greatly reduced without clinical data support

From a wider scope, EEG + sweat monitoring, which correlates to the variations of lactate, several studies touched upon the synergistic applications of EEG and sweat measurement to monitor stress and tension^{17,18}, seizure and epilepsy^{15,19}, and Alzheimer’s disease¹⁶. These disorders can trigger characteristic patterns in both electrophysiological brain state monitoring and produce abnormal metabolic profiles in an individual.

Additionally, the role of EEG and lactate monitoring for the early detection of several disorders including stress, epilepsy, and Alzheimer’s disease have been demonstrated from isolated clinical studies with individual sensing modalities. For EEG, EEG has been widely used as the original dataset to further investigate neuro-disorders including depression²⁰, epilepsy²¹, Alzheimer’s disease²², and beyond. With further signal processing and classification algorithm, modern EEG-based systems can achieve more than 90% detection accuracies for the abovementioned neuro-disorders. For lactate, recent studies have revealed the variation of lactate correlated to physical exercise and its conducive role to improve brain function^{23,24}, such a mechanism helps the treatment of neurodegenerative disorders including the Alzheimer’s disease. In another study²⁵, the authors found that “Insufficient lactic acid production results in an inadequate neuronal energy supply, which affects normal physiological responses and results in brain dysfunction. Conversely, the buildup of lactic acid can lead to abnormal activity in brain areas that cause lactic acid to rise, which leads to brain dysfunction”. Therefore, an in-depth understanding of the molecular mechanisms by which lactic acid regulates brain function is of great value for the early diagnosis and prevention of neuropsychiatric diseases”. These clinical requirements are among the primary motivations for this study.

As a summary, these isolated studies with EEG and lactate as biomarkers for different neuro-disorders revealed that:

- EEG signals reflect the electrical activities of brain behaviors while such electrical activities are directly related to the disorders of neuron activities in the brain.
- Lactate is related to vascular endothelial growth factor (VEGF) and brain-derived neurotrophic factor (BDNF) expression and, by regulating the function of hippocampal mitochondria, can improve brain functions such as angiogenesis and neuroplasticity as well as stress-related symptoms such as depression. As Alzheimer’s disease preventative and to aid individuals with brain function issues and the elderly, lactate-related compounds will be in the limelight. If

lactate levels, which can be regulated via exercise, are well-controlled, it will become an area of interest for future healthcare studies.

Accordingly, we have added supporting statements and clinical references to the Introduction of the manuscript (manuscript lines 38–45). We have also included this response to the 1st response in the “Response to all reviewers” section as a part of the motivation of this work.

16. Identically, when the text discusses the usefulness of electrooculography (EOG) signature eye movements in brain-computer interface (BCI) applications, sleepiness detection, and mobile ophthalmology treatment (lines 155-156) are mentioned. Please use the in-ear sensor to collect and compare the physiological indicators of patients and healthy people, which will further prove the potential and value of the in-ear sensor in biomedicine

We agree with the reviewer’s comments on demonstrating the potential BCI applications of the current study. This would add a lot of weight to this work for sure. However, we may think that specific applications of EOG are not too closely aligned with the main focus of this work: integrated electrophysiological and electrochemical sensing in the ear. We would like to add here, the application of EOG and EEG for has been established in the research field including drowsiness detection¹, eye vergence therapy², and motor control^{3,4}, which we have described in the manuscript. These can be valuable future directions of the applications of our in-ear sensors.

17. Current electrochemical energy storage devices (e.g., batteries) were limited by energy and power density, and thus cannot power the electronics over an extended operational time. In this work, the sensors consume a lot of power, and the input battery energy storage is limited. Will these cause the issue of power supply time deficiency?

This is a very practical concern. Since this work focused on the sensors’ performance characterization, we have mainly employed verified off-the-shelf instrumentations for sensor verification with high accuracy and also enough flexibility to tune testing conditions.

According to the specs from the instruments, for ePhys measurement, the instrument used in this work can sustain an 8-hour battery time. For the eChem measurement, the instrument used in this work can sustain at least a 4-hour battery time.

(Reference: <https://www.glneurotech.com/products/bioradio/device/specifications/>, <https://www.palmsens.com/product/palmsens4/>)

There is a lot of room to customize the backend data acquisition system to achieve the optimized power efficiency, with potential integration with some low-power IC designs we have been and are currently working on. Moreover, power consumption could be potentially reduced by incorporating a different detection mechanism that

replaces the current eChem potential step detection mechanism. For instance, the integration of a potentiometric biosensor and usage of open circuit voltage (OCV) could enable the detection at lower power demand due to their potential change in the presence of a biomarker without the need for any voltage or current input³¹. It is also worth noting that the users will have full control of when to start either ePhys or eChem or both sensing modalities from the in-ear integrated sensors. And under normal conditions, electronics for both sensing modalities can be put in low-power sleep mode in software settings.

We have added a description of the power limitation of the study to the conclusion (manuscript lines 292–296).

18. This in-ear sensor's future potential and potential value should be briefly explained at the end of the abstract and conclusion part. The current narrative (lines 22-23) is too bland without emphasis on the potential and application value when the interaction between the two dimensions of electrophysiological brain state monitoring and health-related metabolite monitoring is observed. For example, the authors should focus more on the profound influence of the sensor application on early disease detection, health monitoring, physical performance improvement, and virtual/augmented reality applications.

With the review's valuable inputs, we have modified the abstract to highlight the profound influence on the sensor applications: "*Such simultaneous and continuous unobtrusive monitoring of brain and body biomarkers permits observing their dynamic and synergic interactions in highly mobile settings, significantly expanding the functionality of the aural device for long-term neurodegenerative disease detection, daily health monitoring, and beyond.*" In the introduction, we now have a dedicated paragraph to highlight the profound implication of combined monitoring of brain state and health-related metabolite (manuscript lines 33–45).

Minor concerns:

19. In the abstract, the originality and breakthrough of the work should also be highlighted, such as "the first breakthrough in combining brain state and health-related metabolite monitoring in a small size". Along the same note, the authors should also briefly discuss such limitations of EEG/metabolite and the necessity to combine them.

We have incorporated the reviewer's conducive comment to the abstract to highlight the breakthrough of this work. We have also added descriptions regarding the limitations.

20. When the abbreviation of a specific noun appears for the first time in the manuscript, it is necessary to explain its full name clearly, even if it is explained in the supporting information (SI). If it is not explained clearly for the first time, it will lead to poor article readability. For example: PCB (252 lines), ADC/PGA/AVDD/AVSS (Fig.S1) and AC/AA/Gluc/UA (166 lines).

We have updated all the abbreviations and made sure their full name was spelled out at their first occurrence.

21. The article numbering is confusing: the “Supplementary Text Note B” mentioned in line 123 cannot be found.

We have fixed all the mismatching errors related to “Supplementary Text Note”.

22. Unclear referring: When referring to “a flat bottom with an adhesive layer” (lines 84-85), it is hard to know which layer of Fig.1d it is, please specify it.

We have modified Fig. 1d to show the double-sided adhesive layer for assembly to earphone.

23. Regarding “a fast and low-cost printing-bonding-assembly process” (lines 85-86), Fig. 1e cannot explain the whole process alone. Please refer together to Extended Data Fig. 2, which can help to detail the total printing process.

We have added Extended Data Fig. 2 as reference to the description as the reviewer suggested.

24. For all Fig: The figures quality of the article is low at present, and there are a lot of defects in the text format, schematic drawing, and picture layout. If the quality of the figures can be improved as follow comments (22-29), it will greatly improve the readability of the article and help this work to be more impressive.

We genuinely appreciate the reviewers’ instructive comments regarding the figures and have made changes accordingly.

All the outline boxes of all figures and text, including solid line boxes and dotted line boxes, contain no scientific information. Please remove them. For example, the gray dotted line box in Fig. 1b, the yellow/red/gray solid line box in Fig. 1f, the gray line box for printing/bonding/assembly, the black solid line in Extended Data Fig. 1f, the black solid line in Extended Data Fig. 2, etc.

We have removed all unnecessary outline boxes. The remaining outline boxes are kept only because we believe they are informative to indicate the boundary of the sub-figures.

Fig. 1: The organization of the pictures is too compact, which will cause certain reading comprehension obstacles. There should be a certain distance between each small picture (such as the distance between Fig. 1a, c, d, and e)

We have increased the separation between sub-figures for Fig. 1.

Fig. 1a: Please keep the style (realistic or anime) and color of each element consistent, including sweat, brain signals, and ear pictures. And the font size in Fig. 1a should be consistent with other parts in Fig. 1.

We have modified Fig. 1a to be all anime style, and we have kept all brain waves and ePhys electrodes in grey color, all sweat lactate and eChem electrodes in green color. We have updated all font sizes in Fig. 1a to be consistent with others.

Fig. 1c: The thickness of all lines and the interval of dashed lines should be kept as consistent as possible (for example, the dashed lines drawn from Fig. 1a to c should be the same as the dashed box style in Fig. 1c). The layout of Fig. 1c is also confusing. The dashed lines should correspond to areas on the images in Fig. 1b, rather than Fig. 1a, as it suggests that the electrodes are on the outer ear rather than the headphone.

We have modified all line widths in Fig. 1c to be consistent. We have now redrawn dashed lines for ePhys and eChem electrodes to be consistent with Fig. 1b.

Fig. 1e: Each font style and weight should be consistent with the rest part of Fig. 1.

We have modified all font styles and weights to be consistent with the rest of Fig. 1.

In Fig. 2, all borders on the left and right sides should be kept in a line, and the second and fourth lines have obvious indents, which should be adjusted accordingly. This adjustment will make the figure clean and improve its quality.

We have adjusted the alignment of the sub figures. We have also checked all the figures and made our best efforts to align sub figures accordingly.

Fig. 2v: The number is wrong, please revise v to b.

We have updated the index.

Fig. 2m: The grey font is too inconspicuous, which may cause reading difficulty. Please use another color like black.

We have changed the font color as mentioned by the reviewer.

Fig. 2n: The font styles of “Look upward” and “Blink” should be consistent with other parts.

We have changed the font styles as mentioned by the reviewer.

All the line icons in Fig. 2 should be placed inside the corresponding pictures. This can help readers understand the meaning of the icons more clearly and reduce misunderstandings. For example, the red line icon indicating continuous in-ear impedance should be placed in Fig. 2a, and so on for other parts. In addition, the text

description of the picture can be shortened, it is a bit too long now, and it can be described in detail in the caption below the picture.

We have moved all legends to inside the figure if the same type of data was plotted only in one sub figure, for example, Fig. 2h. We would like to explain here, the reason we kept some legends outside the figures was that the same legends were used for several sub figures, for example, Fig. 2i–l. We tried adding legends to each of them but that may lead to a very crowded figure with many repetitive legends in the end. The same situation applies to Fig. 3 and Fig. 4. Thus, we are keeping some legends for multiple sub-figures still outside. But we can make further adjustments if the reviewer recommends that way.

The icons in Fig. 3 and Fig. 4 should be placed inside the figure (just like Fig. 3b).

Please see the response to the previous comment.

Fig. 3b and h: the font size is slightly smaller and should remain the same as the other parts. The icons should be placed inside the figure.

We have changed the font styles as mentioned by the reviewer.

Extended Data Fig. 2: all text in Extended Data Fig. 2a–l should not be underlined.

We have made the corresponding changes.

Extended Data Fig. 3a: the font size is slightly bigger; it should remain the same size as the other parts.

We have changed the font styles as mentioned by the reviewer.

Extended Data Fig. 4: As with the previous review comment 25, all the icons should be placed inside the figure. Label g, h, and l should be placed at the top left of a figure.

We have made the changes as recommended by the reviewer.

25. Overall, the manuscript language and grammar could be improved. There are numerous misplaced prepositions (e.g. “in” in line 55, lack of “the” in front of “exercise” in line 21, “higher” in line 78), incorrect spelling (line 105), and confusing sentence structures throughout the manuscript (e.g. the sentence of lines 34–38, 42–46, 53–56). Please seek detailed editing to ensure that the quality of writing reflects the quality of this work.

We have improved the quality of the writing of the manuscript according to the reviewer’s suggestions.

Reviewer #2

Summary:

Ear EEG is an exciting and emerging wearable neural technology. This work integrates wet electrochemical and electrophysiological sensors onto a novel flexible earpiece. The earpiece is combined with a DAQ system that includes a commercial off the shelf chip and combines EEG, ASSR, EDA, EOG and eChem recording. Prior art combines electrophysiological recordings, therefore the main contribution of this work is in the integration of electrochemical measures in the ear, including a novel earpiece fabrication process. Verification performs sequential/time-multiplexed electrochemical and electrophysiological sensing on 3 user subjects in an exercise task in addition to separate electrophysiological recordings and verifications.

We sincerely appreciate the reviewer for the constructive comments. We have revised the manuscript according to the reviewer's comments. In this letter, we also provide a point-to-point response to address each of the comments.

Major technical criticisms/questions:

The introduction provides little to no background or motivation for this integrated device.

We have added detailed background and motivation for this work. Please find our 1st response from the "Response to all reviewers" section.

a. You should redefine and describe acronyms in the body of the text. In particular, you only mention EDA in the abstract and never discuss it in terms of findings.

Per the reviewer's suggestion, we have examined all the acronyms used in the manuscript and added full forms where needed.

In the context of in-ear sensing, electrodermal activity (EDA) pertains to changes in the conductance of the skin of the ear canal measured as changes in electrode-skin impedance directly correlated with the secretion of sweat from eccrine and apocrine glands present in the ear. In this work, we demonstrated the in-ear sensors' impedance profile corresponding to the changes in sweat formulation, which resulted in the continuous decrease of measured electrode-skin impedance. We have added discussion of the EDA when we presented the ear-electrode impedance measurement results (manuscript lines 143–146).

The metrics of the continuous impedance measurement were mainly used as a contact-checking procedure for this work as our primary focus was the use of electrochemical methods for the investigation of a specific sweat biomarker: lactate, which also corresponds to the secretion of sweat in the ear. Per the reviewer's optional suggestion, we have conducted simultaneous in-ear EEG + impedance testing to show the feasibility of adding EDA as another modality. We demonstrated continuous impedance measurement at 10 Hz, 100 Hz, and 500 Hz given the 500

Hz sampling frequency while obtaining 40 Hz ASSR response from the EEG recording. The combined EEG + EDA experiment and results has been added to the Supplementary Note 3 and are discussed in the body of the paper.

b. Lines 17-20. You claim to simultaneously monitor EEG, ASSR, EOG, and EDA. ASSR is typically a cortical response and is considered evoked EEG. EOG is also considered an artifact that appears over EEG and therefore is not a separable metric. It's also unclear if EEG, EOG, and EDA are truly recorded simultaneously. Reported measurements are time-multiplexed, but details on the acquisition are not provided. Are they all continuous? Do you have data that shows their continuous, simultaneous acquisition?

The measurements of EEG, EOG, EDA are electrophysiological signals derived from biopotential measurements made using the same ePhys sensor, hardware setup, and recording protocol.

The in-ear ePhys sensors are intrinsically multimodal, in that they simultaneously acquire biopotentials from various sources of brain and body electrical activity including, besides EEG and EOG, the electromyogram (EMG), along with electrochemical impedance registering EDA. Nevertheless, the data acquisition did not stop but continuously sense for all the modalities the reviewer mentioned, what differentiated the different recording metrics was the different human activities triggering these biometrics at different timings.

To further demonstrate the capability of the sensors to perform concurrent multi-modal ePhys sensing, we have conducted simultaneous in-ear EEG + EOG testing to show how the ipsilateral referencing setup used by the in-ear sensor helps to reduce EOG artifacts for EEG recording. In the updated Extended Data Fig. 6, we demonstrated 40 Hz ASSR recording with EEG recording while the subjects did eye blinks every 5 seconds, the 40 Hz ASSR result was comparable to the baseline. The eye motions were recorded by the EOG. We have also conducted simultaneous in-ear EEG + EDA testing to show simultaneous impedance measurement which did not affect the EEG ASSR recording. Specially, no effects were observed in case the impedance measurement was taken at a frequency outside the main EEG frequency band.

We agree with the reviewer that ASSR is considered EEG. We have removed the repetitive ASSR description in the Abstract. Details of the combined EEG + EOG, EEG + EDA testing have been added to the Supplementary Note 3 (supplementary information lines 425–485).

c. Lines 25-56. There is little/no background or introduction that would be important for understanding the work as a whole. What are existing methods of metabolite monitoring on the skin (generically saying tomography imaging is too broad)? Are there commercial or widespread devices that perform metabolite sensing on skin?

What are common targets for metabolite sensing? How accurate are existing metabolite sensing platforms?

We have modified the 3rd paragraph of the Introduction section (manuscript lines 46–61), which includes existing methods of EEG and metabolite sensing as well as their limitations.

For metabolite sensing, we have provided more detailed background: *While skin metabolic health monitoring has been demonstrated using skin penetrating tools (small filaments, blood pricking/sampling), noninvasive platforms (epidermal patches) or optical procedures (Near Infrared, UV-Visible or Raman spectroscopy)³², their availability in the market as reliable commercial technologies is limited in the form of skin-penetrating filaments or blood collection approaches that require small or higher volumes of sample to perform metabolite analysis³³. Depending on the collection approach, the metabolites analyzed using such technologies vary. For instance, commercially available devices in the form of small filaments are capable of continuously analyzing glucose from the interstitial fluid, whereas commercial blood analysis techniques provide a broader range of metabolites such as lactate, glucose, ketones, arginine, asparagine, taurine, glutamine, tyrosine among others.*

EEG is similarly left unexplained. Are you claiming that in ear EEG will be just as good as scalp EEG? That is implicit in the current introduction which is wrong because no data suggests in ear EEG provides the same coverage/SNR as scalp EEG.
Introduction

We did not claim that the SNR produced by the sensors in our study was comparable to scalp-EEG. In Supplementary Note 3, we presented the validation studies using a commercial scalp-EEG for the in-ear sensors. With electrodes adjacent to brain areas that generated corresponding EEG signals, such as alpha band signals, scalp-EEG had an SNR that was three to four times higher. In terms of the auditory brain response ASSR, the scalp-EEG channels near the ear performed around 1.3 times better than the in-ear sensors. To clarify, we have added to the manuscript: *The electroencephalogram (EEG) collected on the scalp with gel-based electrodes allows great spatial coverage and high signal-to-noise (SNR) ratio in active brain state monitoring, but at the expense of restricting the user's mobility and comfort. Dry-contact EEG electrodes provide much improved user comfort and reduced setup time, but at a loss in SNR primarily when used over hairy sites on the scalp.*

d. Lines 38-40. From my understanding, the density of sweat glands is highly variable around the human and thus different parts of the human body will have different sweat responses [1]. Is the ear canal a better or worse place to monitor sweat than other parts of the body?

We verified the feasibility of sweat measurement in the ear, and we also located the area where sweat secretion was relatively higher than in other regions in the ear. The outer ear canal is suitable for sweat sensing because it is one of the few

locations on the body (the others being the palms and the underarms) in which there is a concentration of apocrine sweat glands, in addition to the ubiquitous eccrine sweat glands. The EDA phenomenon is similarly only detectable in limited regions of the body where apocrine glands are in high enough density, making the ear canal a uniquely advantageous region for assessment of sweat and sweat metabolites. We did find that sweat secretion in the ear was less in volume compared with other parts of the body. However, our main argument for sweat sensing in the ear was that the ear was the most ideal location for integrated brain signal and metabolite signal sensing. And please see the response to the next comment to see why we selected the ear.

e. Line 47. What is the reasoning for going inside the ear? Is the claim that the ear will provide better SNR than other locations for integrated sensors? Is the claim that it's more discreet? Is the claim that the ear provides a trade-off between EEG, heart rate, and blood ox sensing?

Mainly for 3 reasons: it's mechanically stable; it has access to the brain as well ear sweat glands for multi-modal sensing; It's discreet and concealable. For more details, please find our 1st response in the "Response to all reviewers" section, where we explained why choosing the ear as the sensing location.

f. Line 68. Isn't lactate primarily a physical stress marker (from exercise)? Or is it also a marker of emotional and psychological stress as well?

Previous research has found that lactate is both a marker for physical stress and psychological stress^{34,35}. For psychological stress, these works have used Trier Social Stress Test (TSST), a well-documented psychosocial stressor, and observe that the lactate concentration increases significantly above baseline in response to physiological stress.

g. Line 98. Does this hydrogel require periodic reapplication? What would have happened if no hydrogel was applied (doesn't have to go in the introduction).

The hydrogel does not require any periodic reapplication. The hydrogel acts as a hydrophilic porous membrane that facilitates the collection of sweat on the lactate biosensor. Therefore, in a scenario where the hydrogel membrane was not present, there will be partial or negligible coverage of sweat on top of the biosensor.

h. Was the sweat study (with the ecoflex earpiece) performed only on a single subject? Is that enough?

It was performed on 3 subjects. Please also find our response from the "Response to all reviewers" section.

i. Line 103. Does this mean the ePhys electrodes have hydrogel as well? Were there concerns of 'bridging' (where electrodes are shorted by hydrogel) between all the different sensors?

There was no hydrogel on the ePhys electrodes, which can be demonstrated in Fig. 1b and d. We were aware of the potential crosstalk between sensors since they both contact the skin, which itself is conductive. First of all, the 3D ePhys electrodes and the PVA hydrogel-covered eChem electrodes were physically separated by SEBS-covered substrate in between. Second, the PVA hydrogel was chosen also for its hydrophilicity to collect sweat. Compared with the dome-shaped ePhys electrode, it had a much larger and flatter contact surface with the skin. The hydrophilicity and flat surface made it more likely to collect sweat. Apart from all the design aspects, we further verified the crosstalk between both sensing modalities in detail, showing minor transient crosstalk only at the onset of the lactate sensing session (Supplementary Note 4).

j. Line 108. More information should be given on the DAQ. How much power does it dissipate? What is the data acquisition rate, datarate, and how does it perform multiplexing? (apologies if I missed this somewhere).

Since this work focused on the sensors' performance characterization, we have mainly employed verified commercial instrumentations with BLE wireless capabilities to minimize power consumption. According to the specs from the instrumentations, for ePhys measurement, the instrument used in this work can sustain an 8-hour battery time. For the eChem measurement, the instrument used in this work can sustain at least a 4-hour battery time (Reference: <https://www.glneurotech.com/products/bioradio/device/specifications/>, <https://www.palmsens.com/product/palmsens4/>).

In the "Electrophysiological measurement system integration and the on-body setup" section in Method, we reported using 500 Hz/channel (two ears, 6 channels input) or 1 kHz/channel (1 ear, 3 channels input), measurement mode: Single-ended, resolution: 1 μ V, input range: ± 187 mV. As our response to a previous comment, the electrophysiological multiplexing was dependent on the activities of the subjects, all activities that resulted in biopotential changes in the ear canal were measured. In the signal processing stage, we extracted information for Alpha, ASSR in the frequency domain, and EOG in the time domain. For combination with the electrochemical testing, we have two instruments for individual data acquisition, both streaming wirelessly. On the PC, two processes were run in parallel for data collection.

To add clarity, we have added Fig. S2 to show instrumentations and their connection to the sensors. We have also added a section in Supplementary Note 3 to demonstrate the practical implementation of simultaneous multi-modal ePhys sensing (supplementary information lines 425–485).

k. Line 129. In ear electrode characterization results are unclear.

i. Impedance/area claim is unclear. This work's electrodes are indeed smaller but this should be more clearly stated. Furthermore, you should state the area of reference 18's electrodes.

We have added the electrode area (60 mm²) reported in reference⁵.

ii. Population size and number of averaged trials is not stated.

The impedance results reported in the manuscript were from two subjects, two trials (each subject 1 ear), and 6 ePhys electrodes in total. The statistics of subjects have been mentioned in Methods (manuscript lines 393–395) and Table S4.

iii. Figure 2 shows a 2-minute settling time. This seems very long and is not discussed. Is this comparable to the state of the art? Is there a reason? Was this the case across all users? Did users have to wait several minutes before starting a trial? How many users? How many trials?

We would like to note here it is common for the dry-contact electrode to take time and establish a stable electrode-skin contact. As described in the "Electrophysiological sensing validation" section of the supplementary information, we had to go through a similar impedance verification step before using a commercial dry contact EEG headset. Such a process usually takes minutes for all channels to be "valid". We believe the reason was because of the sweat formulation at the electrode-skin interface which has been reported before³⁶ but here specifically in the ear. We conducted further analysis of this hypothesis to confirm the sweat secretion onto the electrode in Supplementary Note 3.

Users indeed have to wait before starting a trial, this has been reported in the "Electrophysiological measurement system integration and the on-body setup" section in Method. We observed such a trend for almost all users. As a sanity check, all experiments used continuous impedance to verify stable ePhys electrode contact before the experiment was conducted. Such practice is common in EEG measurements and is only required at the onset of the wearing.

The impedance results reported in the manuscript were from two subjects, two trials (each subject 1 ear), and 6 ePhys electrodes in total. The statistics of subjects have been mentioned in Methods (manuscript lines 393–395) and Table S4.

iv. Are the in-ear impedance measurements normally distributed? How is the standard deviation region calculated?

Please see the impedance measurement results of each ePhys electrodes of the 2 trials below. The standard deviation region was calculated in a standard way as follows: for each impedance measurement, the complex impedance value was

sampled a fixed number of times, and the setting was the same across the two subjects. We first extracted the impedance and the phase from the complex impedance, and then at each sample, we had 6 data points for one type of testing (continuous or EIS). For each sample, we calculated the mean among the 6 data points and then calculated the standard deviation following the standard method $s_t = \sqrt{\frac{1}{5} \sum_{i=1}^6 (x_{it} - \bar{x}_t)^2}$. Here s_t is the standard deviation at sample t , x_{it} is 1 of the 6 datapoints at this sample t . The figure was then made by plotting the mean and standard deviation for all samples and interpolating lines between samples. Revision Figure 3 show the impedance measurement results of each trial.

Revision Figure 3. Individual electrode-ear impedance characterization with **a.** continuous electrode-ear impedance, **b.** impedance phase over 120 s after insertion, **c.** impedance spectrum, and **d.** phase spectrum after the electrode-ear interface reached the steady state (inserted for more than 2 mins) at 1–1 kHz.

v. Reported EDO is a positive number but figure 2f makes it seem like the mean should be negative? Was this a typo? Was a negative dropped?

We have examined the electrode DC offset (EDO) data again and can confirm that Fig. 2e, f were plotted correctly.

Fig. 2e intends to show the “averaged waveform” for 96 10-s EDO recordings. For each time index n , the averaged EDO $v[n]$ was calculated as $v[n] = \frac{1}{96} \sum_{s=1}^{96} v_s[n]$, where $n = 1-10000$ for a 10-s duration, since the sampling rate of the EDO

recording was 1 kHz; s is the recording index, $s = 1-96$ since 96 EDO recordings were averaged.

Fig. 2f intends to demonstrate the “normal distribution” of the 10-s averaged EDO of 96 10-s EDO recordings. Each datapoint, the 10-s averaged EDO value $edo[s]$ was calculated by $edo[s] = \frac{1}{10000} \sum_{n=1}^{10000} v_s[n]$. The probability density function used to plot Fig. 2f used $edo[s]$ as the samples for statistical analysis. So effectively, the seemingly negative mean in the reviewer’s comment was $\frac{1}{96} \sum_{s=1}^{96} (\frac{1}{10000} \sum_{n=1}^{10000} v_s[n])$. The actual value of this $\mu = 0.59$ mV, which is the same as the one reported in Fig. 2e, calculated as $\frac{1}{10000} \sum_{n=1}^{10000} (\frac{1}{96} \sum_{s=1}^{96} v_s[n])$.

The reason why Fig. 2f appears to have a negative mean is because there are more than 30 recordings with a -10 mV– 0 mV EDO value, but there are 7 recordings with a 30 mV– 60 mV EDO value, compared with only 1 at the negative end, which results in a positive EDO value mean.

We have updated both the EDO characterization method description (manuscript lines 419–431) as well as further labeling Fig. 2f.

l. Line 160. Are there any quantitative metrics to compare the in-ear eChem sensors to the commercial blood sensor? I imagine the blood sensor will be more accurate but is there a way to compare measured trends quantitatively?

The blood readings collected during each experiment were used to validate the increase in lactate concentration. Given that the readings collected from the electrochemical sensor are in current units (μ A), a potential way in which both eChem sensors and blood readings could be compared is by converting the current readings to blood concentration units (mg/dL). This could be assessed by using an external calibration curve at different lactate concentrations and taking into consideration other correction factors such as the pH and temperature of each volunteer. These factors can be the scope of future work.

m. Line 168. 4% is fairly significant drift for only 10 CA scans. What is the expected drift over longer periods of time? What is required for a given application and how many errors can be tolerated? Does the device need to be recalibrated? What is the known lifetime?

We have performed extended experiments on lactate CA scans. Please refer to our 3rd response in the “Response to all reviewers” section.

n. Line 199. Why are the pre/post-exercise Alpha experiments different? It is best not to compare two different experimental methods and not address the reasons for doing so? Was no alpha measured in the first eyes open/closed session?

Procedure and instrumentation wise, the pre/post Alpha experiments were conducted under exactly the same condition, the only difference here was the subjects themselves, who performed exercise and were in an excitement state along with other body indicators like heart rate and breathing rate. Alpha modulation was measured and analyzed in the first eyes open/closed session (both eyes open and closed period were measured) as described in the “Combined EEG and lactate sensing” section in Methods. In the Results section, the pre/post alpha experimental results were not only compared, showing consistency with previously reported trends^{37,38}, but also further input to a feature extraction pipeline to classify the rested state and excited state after exercise, which we further explain in the next comment.

We have modified the Methods section to clarify the pre/post-exercise measurement conditions and emphasized that the same measurement setup was used (manuscript lines 608–610).

o. Line 222. What is this analysis and what does it mean that they have different cognitive states? Was there a control experiment performed? Alpha modulation can vary across 30 minutes and hyperventilating. Table s1 should be moved to the results because otherwise your primary claims of being able to do in ear EEG to show the change in cognitive states aren't supported by the main manuscript. Even with the table it is unclear if it's a fair comparison given the differences in experimental procedure before and after exercise.

The FBCSP analysis pipeline is a robust signal processing method used to distinguish between two different brain states based on the activity in different EEG frequency bands. Through spatial separation of recorded data from different electrodes, mutual information-based feature selection and a support vector machine classifier³⁹, we obtain the accuracy of predicting the difference between two brain states, pre-exercise vs post-exercise in our case. We show that our classification accuracy improves when using more frequency bands. Also, FBCSP finds the immediate recordings post-exercise to be highly distinctive than pre-exercise recordings, whereas recordings from a relaxed state of the subject post-exercise appear closer to the pre-exercise state.

Control experiments: first, we collected EEG streams from 5 subjects in total for reproducibility. And then, for each subject's EEG data stream, we found, from the alpha modulation analysis in Fig. 4e, that the Alpha modulation results of the pre-exercise and post-exercise relaxed states were similar, indicating similarity between the brain states. This result was in line with the FBCSP processing results we obtained, here using FBCSP method to process the same set of data, we found that the FBCSP method can accurately predict pre-exercise brain state and post-exercise-immediate brain state. As a comparison, the accuracy to predict pre-exercise and post-exercise relaxed brain states was much lower using the FBCSP method (Table 1), which also indicated that the pre-exercise and post-exercise-relaxed brain states were similar.

Accordingly, we have moved the previous Table S1 to the main text as Table 1, we have added a Table S2 to show FBCSP analysis results for each individual subject.

p. Line 232. It looks like the devices are modeled on Apple airpods pro with afterparty ear wings. These ear wings usually come in multiple sizes (small, medium, large). Were all of these experiments performed with one size of earpiece? If so, does that mean this device is truly user-generic?

The claim of this device being user-generic was because we had the two following considerations to account for subjects' ear anatomy variation. The first consideration was also the reviewer's first question, we used 3 different-sized ear wings or silicone earphone tips. For different subjects, a suitable silicone tip size was first chosen before the sensors' assembly as well as the on-body experiments. By choosing the appropriate earphone silicone tip, a first-stage tight fitting to the ear canal was achieved. The sensors used one outline contour, it could fit three sizes of the ear wings and most importantly the earphone body. When inserting it into the ear, we made sure the eChem sensor was facing the tragus. 2), the sensors were further designed to have 3D ePhys structures as well as a PVA gel covering the eChem sensors. Such structures both used soft materials, serving as cushioning layers between the already tight-fitted ear canal-sensor interface, further achieving an interference fit.

q. Figure captions must clearly state how many subjects were involved in the experiment and whether their data is plotted separately or averaged. For example are parts 3 d-g single user data or are they averaged?

We have provided the statistical information in both the Methods section and Table S2.

General comments:

This work provides a first step to building a new class on integrated electrochemical and electrophysiological sensors. The earpiece in this design is interesting but requires wet hydrogel electrodes, which are impractical and not user-friendly (no one wants something wet in their ear). Thus, why would this feature be added to a headphone rather than a more comfortable patch system elsewhere on the body?

First, to answer the reviewer's comment on why integrating with a headphone instead of other parts of the body, please refer to the first response in the "Response to all reviewer" section.

In this work we demonstrated the collection of sweat samples using hydrogel membranes, we understand that the usage of hydrogel membranes might lead to impractical daily usage. However, our goal was to show a proof-of-concept device on which two different sensing modalities can be integrated into a small platform such as an earpiece. And as we introduced for previous comments, ear would be

the ideal location for combined brain state and metabolite monitoring. In future work, these hydrogel membranes can be potentially replaced with a sophisticated microfluidic system that allows the collection of sweat samples without the presence of a hydrogel layer.

The data acquisition system is also large and cumbersome. The introduction/background should properly motivate the uses of such a device and the motivation for this combination, which is unclear in the manuscript.

Since this work focused on the sensors' performance characterization, we have mainly employed verified off-the-shelf instrumentations for sensor verification in a wireless fashion (with Bluetooth low power). There is a lot of room to customize the data acquisition system, this can involve both the hardware and firmware low-power design aspects including integration with low-power analog front-end design^{40,41}, sleep-mode implementation of the microcontroller, and the appropriate wireless protocol implementation. The power consumption could also be potentially reduced by incorporating a different detection mechanism that replaces the current eChem potential step detection mechanism. For instance, the integration of a potentiometric biosensor and usage of open circuit voltage (OCV) could enable the detection at lower power demand due to their potential change in the presence of a biomarker without the need for any voltage or current input³¹. It is also worth noting that the users will have full control of when to start either ePhys or eChem or both sensing modalities from the in-ear integrated sensors. And under normal conditions, electronics for both sensing modalities can be put in low-power sleep mode.

Accordingly, we have mentioned sensor-electronics integration as one future direction of this work in the conclusion (manuscript lines 294–296).

Since this device is used in an exercise task, no mention is made of motion artifacts and how to mitigate their effects in a real application.

Motion artifacts were an important consideration for the exercise experiment conducted in this work. Since the DAQ setup was based on the commercially available BioRadio, there was not much room for optimization on the hardware side. In the current exercise experiment, the EEG analysis was based on data streams taken when the subjects were seated and remained still.

Signal processing techniques can be applied to the in-ear EEG sensors in this work. In supplementary Note 3, we have added a section to demonstrate the feasibility of using the automatic subspace reconstruction (ASR) algorithm for motion artifact reduction on EEG data recorded from the ePhys sensors (supplementary information lines 486–538). With the ASR algorithm, the pre-motion 40 Hz ASSR peak seen in the baseline experiment can be recovered from the elevated noise floor resulting from the subject's motion. The result demonstrates the effectiveness of using ASR to reduce in-ear EEG motion artifacts.

Furthermore, the results and discussion do not state what such an integrated device could enable. Does it provide better performance over alternatives? Is there new utility that is provided from this form factor?

The device in this work enabled simultaneous brain state and health-related metabolite sensing in one integrated device, unobtrusively. Such combined sensing of brain state and metabolite was enabled by the meticulous design of the device to fit inside the ear. Such integrated sensing was first achieved by using the ear as the sensing location, with merits we have provided in the first response in the “Response to all reviewers” section.

The sensing modality of this work is differentiated from most previous works. Multi-modal sensing in the ear has been demonstrated, however, previous approaches were focused on physiological sensing, i.e. ExG, PPG, and temperature. Common approaches include integrating off-the-shelf sensors such as SpO2 sensors, and temperature sensors for such targets. However, the electrochemical side of the ear, especially in combination with brain signal monitoring, has rarely been explored. Brain state and metabolite monitoring, compared with combined physiological sensing, is covering a broader range of real-life applications due to the orthogonality of electrophysiological measurement and electrochemical measurement targeting a different set of human biomarkers. To our knowledge, there is only one group that reported the sensing of biomarkers in the auditory canal using electrochemical detection^{13,14}. According to their description, “the developed earplug device that integrated the chemical sensor was proved to be difficult to hold in the same position during the human trials. As a result, the collected data and experiment were inaccurate and excluded from the manuscript”. Their second paper then had no brain state recording and was completely outside the ear.

Finally, the device presented in our manuscript represented the first approach towards integrated brain state and health-related metabolite sensing in a highly wearable setting (integrating onto a generic earphone). As an instance of application, we have demonstrated the feasibility of continuous collection of lactate sensing data for 40 min during physical performance with concurrent collection of EEG data. To our best knowledge, there has not been any new utility that is provided from the form factor in this work, and can do integrated brain state and metabolite sensing.

Minor technical criticisms/questions:

1. Line 140. Did all users use the same earpiece?

No, they used different earpieces, we used 4 different earphones, each assembled with ear sensors. In case an earphone was used more than once, the printed sensors alongside the connection pad were removed, followed by a cleaning with isopropanol alcohol pads to remove any sweat residue.

2. Line 169. Please define CA, it's only defined in the figure caption. Please also be sure that all acronyms are defined throughout the text.

Chronoamperometry (CA) was defined in its first occurrence at manuscript line 127. All acronyms have been examined to be spelled out at the first occurrence.

3. In the sweat detection video – why is there such a steep drop off in the CA curve? Shouldn't there be a slightly more gradual response?

As introduced in the Methods section, we used 50-point sliding window averaging method to process lactate current for the results shown in the manuscript. While in the video recording, we showed raw and unprocessed data for real time demonstration. Additionally, the change of CA current is subject dependent with sweat formation. The video shows a subject with rapid response to the sweat formation in the PVA gel.

For clarity, we now have added to the Method section: *The recorded CA currents from the eChem sensors were temporally averaged over a 50-point sliding rectangular window to filter out high-frequency noise due to PVA hydrogel electrochemical fluctuations at the eChem sensor-skin interface* (manuscript lines 533–535). While in the supplementary video description, we added “*The video demonstrated real-time, raw and unprocessed lactate CA current*” (supplementary information lines 845–847).

Missing details regarding statistics:

1. In ear electrode impedance measurements lack important statistical information.

We have provided the statistical information in both the Method section and the caption.

2. Lactate sensing also seems to lack population size.

In the manuscript, we reported lactate sensing with 6 subjects in total, we have provided the statistical information in both the Method section and the figure caption.

Missing citations:

Optional suggestions for improvement:

1. Rewrite the introduction to motivate the work and explain more in depth applications of such an integrated device.

We have now reworked the second paragraph of the Introduction (manuscript lines 33–45). Here we highlighted the conducive role of combined brain state and health-related metabolite sensing for neurodegenerative diseases, and cognitive state monitoring, and referenced further clinical supporting research.

2. Perform simultaneous EEG and EDA measurement.

We have conducted further experiments to demonstrate simultaneous EEG and EDA measurement as described in Supplementary Note 3 (supplementary information lines 425–485).

3. Discuss the impact of motion artifacts in the experiment.

We have conducted further experiments to demonstrate EEG, motion artifacts, and a potential ASR-based signal processing approach to counter motion artifacts as described in Supplementary Note 3 (supplementary information lines 486–538).

4. Provide quantitative metrics for lactate sensing if possible.

We appreciate the suggestion, in this work, blood lactate was used as validation however not as quantitative metrics. Based on the control assessment (Fig. 3h), there is a strong correlation of the blood lactate and the sweat lactate (n=3).

5. Add a conclusion to tie up the findings and reconnect it to the motivation of your work.

We have now added a dedicated conclusion paragraph (manuscript lines 286–296).

Stylistic issues:

1. Figure 1 is a bit hard to follow visually (which I realize may be due to the figure count limitation). Would it be possible to place boxes to at least visually separate disparate parts of the figure? For example, a box could be placed around a-d, a separate box around f, and a third box around g-j.

We appreciate the reviewer's comment. However, the other reviewer suggested removing such boxes for clarity. To address the reviewer's concern, we have made the separation of sub-figures in Fig. 1 larger to promote readability.

Just a thought. 2. A more traditional 2D fabrication process guide may be more intelligible than the 3D one currently in Figure 1. It would also be significantly more space efficient.

We have now added pointers to the Extended Data Figure 2 when describing fabrication methods to show detailed fabrication procedures (manuscript line 102).

References

- 1 Sikander, G. & Anwar, S. Driver fatigue detection systems: A review. *IEEE Transactions on Intelligent Transportation Systems* **20**, 2339-2352 (2018).
- 2 Mishra, S., Kim, Y.-S., Intarasirisawat, J., Kwon, Y.-T., Lee, Y., Mahmood, M., Lim, H.-R., Herbert, R., Yu, K. J., Ang, C. S. & Yeo, W.-H. Soft, wireless periocular wearable electronics for real-time detection of eye vergence in a virtual reality toward mobile eye therapies. *Science Advances* **6**, eaay1729 (2020).
- 3 Ma, J., Zhang, Y., Cichocki, A. & Matsuno, F. A novel EOG/EEG hybrid human-machine interface adopting eye movements and ERPs: Application to robot control. *IEEE Transactions on Biomedical Engineering* **62**, 876-889 (2014).
- 4 Li, Y., He, S., Huang, Q., Gu, Z. & Yu, Z. L. A EOG-based switch and its application for “start/stop” control of a wheelchair. *Neurocomputing* **275**, 1350-1357 (2018).
- 5 Kaveh, R., Doong, J., Zhou, A., Schwendeman, C., Gopalan, K., Burghardt, F. L., Arias, A. C., Maharbiz, M. M. & Muller, R. Wireless User-Generic Ear EEG. *IEEE Transactions on Biomedical Circuits and Systems* **14**, 727-737 (2020).
- 6 Kappel, S. L., Rank, M. L., Toft, H. O., Andersen, M. & Kidmose, P. Dry-Contact Electrode Ear-EEG. *IEEE Transactions on Biomedical Engineering* **66**, 150-158, doi:10.1109/tbme.2018.2835778 (2019).
- 7 Nakamura, T., Goverdovsky, V., Morrell, M. J. & Mandic, D. P. Automatic Sleep Monitoring Using Ear-EEG. *IEEE Journal of Translational Engineering in Health and Medicine-Jtehm* **5**, doi:10.1109/jtehm.2017.2702558 (2017).
- 8 Athavipach, C., Pan-Ngum, S. & Israsena, P. A wearable in-ear EEG device for emotion monitoring. *Sensors* **19**, 4014 (2019).
- 9 Mikkelsen, K. B., Tabar, Y. R., Kappel, S. L., Christensen, C. B., Toft, H. O., Hemmsen, M. C., Rank, M. L., Otto, M. & Kidmose, P. Accurate whole-night sleep monitoring with dry-contact ear-EEG. *Scientific reports* **9**, 16824 (2019).
- 10 Bleichner, M. G., Mirkovic, B. & Debener, S. Identifying auditory attention with ear-EEG: cEEGrid versus high-density cap-EEG comparison. *Journal of neural engineering* **13**, 066004 (2016).
- 11 Sempionatto, J. R., Lin, M., Yin, L., De la Paz, E., Pei, K., Sonsa-Ard, T., de Loyola Silva, A. N., Khorshed, A. A., Zhang, F. & Tostado, N. An epidermal patch for the simultaneous monitoring of haemodynamic and metabolic biomarkers. *Nature Biomedical Engineering* **5**, 737-748 (2021).
- 12 Smith, C. J. & Havenith, G. Body mapping of sweating patterns in male athletes in mild exercise-induced hyperthermia. *European journal of applied physiology* **111**, 1391-1404 (2011).
- 13 Rosa, B. G., Anastasova-Ivanova, S. & Yang, G. Z. A Low-powered and Wearable Device for Monitoring Sleep through Electrical, Chemical and Motion signals recorded over the head in 2019 *IEEE Biomedical Circuits and Systems Conference (BioCAS)*. 1-4 (IEEE).
- 14 Gil, B., Anastasova, S. & Yang, G. Z. A smart wireless Ear-Worn device for cardiovascular and sweat parameter monitoring during physical exercise: design and performance results. *Sensors* **19**, 1616 (2019).
- 15 Elger, C. E. & Hoppe, C. Diagnostic challenges in epilepsy: seizure under-reporting and seizure detection. *The Lancet Neurology* **17**, 279-288 (2018).
- 16 Jardanhazy, A., Jardanhazy, T. & Kalman, J. Sodium lactate differently alters relative EEG power and functional connectivity in Alzheimer’s disease patients’ brain regions. *European Journal of Neurology* **15**, 150-155 (2008).
- 17 Fechir, M., Schlereth, T., Kritzmann, S., Balon, S., Pfeifer, N., Geber, C., Breimhorst, M., Eberle, T., Gamer, M. & Birklein, F. Stress and thermoregulation: Different sympathetic responses and different effects on experimental pain. *European Journal of Pain* **13**, 935-941 (2009).
- 18 Azgomi, H. F., Cajigas, I. & Faghieh, R. T. Closed-Loop Cognitive Stress Regulation Using Fuzzy Control in Wearable-Machine Interface Architectures. *IEEE Access* **9**, 106202-

- 106219 (2021).
- 19 Gao, K.-P., Shen, G.-C., Zhao, N., Jiang, C.-P., Yang, B. & Liu, J.-Q. Wearable multifunction sensor for the detection of forehead EEG signal and sweat rate on skin simultaneously. *IEEE Sensors Journal* **20**, 10393-10404 (2020).
- 20 de Aguiar Neto, F. S. & Rosa, J. L. G. Depression biomarkers using non-invasive EEG: A review. *Neuroscience & Biobehavioral Reviews* **105**, 83-93 (2019).
- 21 Acharya, U. R., Fujita, H., Sudarshan, V. K., Bhat, S. & Koh, J. E. Application of entropies for automated diagnosis of epilepsy using EEG signals: A review. *Knowledge-based systems* **88**, 85-96 (2015).
- 22 Tsolaki, A., Kazis, D., Kompatsiaris, I., Kosmidou, V. & Tsolaki, M. Electroencephalogram and Alzheimer's disease: clinical and research approaches. *International journal of Alzheimer's disease* **2014** (2014).
- 23 Valenzuela, P. L., Castillo-Garcia, A., Morales, J. S., de la Villa, P., Hampel, H., Emanuele, E., Lista, S. & Lucia, A. Exercise benefits on Alzheimer's disease: State-of-the-science. *Ageing research reviews* **62**, 101108 (2020).
- 24 Lee, S., Choi, Y., Jeong, E., Park, J., Kim, J., Tanaka, M. & Choi, J. Physiological significance of elevated levels of lactate by exercise training in the brain and body. *Journal of Bioscience and Bioengineering* (2023).
- 25 Chen, X., Zhang, Y., Wang, H., Liu, L., Li, W. & Xie, P. The regulatory effects of lactic acid on neuropsychiatric disorders. *Discover Mental Health* **2**, doi:10.1007/s44192-022-00011-4 (2022).
- 26 Jia, W., Bandodkar, A. J., Valdés-Ramírez, G., Windmiller, J. R., Yang, Z., Ramírez, J., Chan, G. & Wang, J. Electrochemical tattoo biosensors for real-time noninvasive lactate monitoring in human perspiration. *Analytical chemistry* **85**, 6553-6560 (2013).
- 27 Keogh, A., Argent, R., Anderson, A., Caulfield, B. & Johnston, W. Assessing the usability of wearable devices to measure gait and physical activity in chronic conditions: a systematic review. *Journal of NeuroEngineering and Rehabilitation* **18**, 1-17 (2021).
- 28 Knight, J. F. & Baber, C. A tool to assess the comfort of wearable computers. *Human factors* **47**, 77-91 (2005).
- 29 Levander, M. S. & Grodzinsky, E. Variation in normal ear temperature. *The American journal of the medical sciences* **354**, 370-378 (2017).
- 30 Gray, R., Sharma, A. & Vowler, S. Relative humidity of the external auditory canal in normal and abnormal ears, and its pathogenic effect. *Clinical otolaryngology* **30**, 105-111 (2005).
- 31 Gellett, W., Kesmez, M., Schumacher, J., Akers, N. & Minteer, S. D. Biofuel Cells for Portable Power. *Electroanalysis* **22**, 727-731, doi:10.1002/elan.200980013 (2010).
- 32 Bandodkar, A. J., Jeerapan, I. & Wang, J. Wearable Chemical Sensors: Present Challenges and Future Prospects. *ACS Sensors* **1**, 464-482, doi:10.1021/acssensors.6b00250 (2016).
- 33 Chaleckis, R., Murakami, I., Takada, J., Kondoh, H. & Yanagida, M. Individual variability in human blood metabolites identifies age-related differences. *Proceedings of the National Academy of Sciences* **113**, 4252-4259, doi:10.1073/pnas.1603023113 (2016).
- 34 Hermann, R., Lay, D., Wahl, P., Roth, W. T. & Petrowski, K. Effects of psychosocial and physical stress on lactate and anxiety levels. *Stress* **22**, 664-669, doi:10.1080/10253890.2019.1610743 (2019).
- 35 Kubera, B., Hubold, C., Otte, S., Lindenberg, A.-S., Zeiß, I., Krause, R., Steinkamp, M., Klement, J., Entringer, S., Pellerin, L. & Peters, A. Rise in Plasma Lactate Concentrations with Psychosocial Stress: A Possible Sign of Cerebral Energy Demand. *Obesity Facts* **5**, 384-392, doi:10.1159/000339958 (2012).
- 36 Chi, Y. M., Jung, T.-P. & Cauwenberghs, G. Dry-contact and noncontact biopotential electrodes: Methodological review. *IEEE Reviews in Biomedical Engineering* **3**, 106-119 (2010).
- 37 Crabbe, J. B. & Dishman, R. K. Brain electrocortical activity during and after exercise: a quantitative synthesis. *Psychophysiology* **41**, 563-574 (2004).
- 38 Bailey, S. P., Hall, E. E., Folger, S. E. & Miller, P. C. Changes in EEG during graded exercise on a recumbent cycle ergometer. *Journal of Sports Science & Medicine* **7**, 505 (2008).
- 39 Paul, A., Hota, G., Khaleghi, B., Xu, Y., Rosing, T. & Cauwenberghs, G. Attention State

- Classification with In-Ear EEG in *2021 IEEE Biomedical Circuits and Systems Conference (BioCAS)*. 1-5 (IEEE).
- 40 Huang, J. & Mercier, P. P. A 178.9-dB FoM 128-dB SFDR VCO-Based AFE for ExG Readouts With a Calibration-Free Differential Pulse Code Modulation Technique. *IEEE Journal of Solid-State Circuits* **56**, 3236-3246, doi:10.1109/jssc.2021.3112635 (2021).
- 41 Kim, C., Joshi, S., Courellis, H., Wang, J., Miller, C. & Cauwenberghs, G. Sub- μ Vrms-Noise Sub- μ W/Channel ADC-Direct Neural Recording With 200-mV/ms Transient Recovery Through Predictive Digital Autoranging. *IEEE Journal of Solid-State Circuits* **53**, 3101-3110, doi:10.1109/jssc.2018.2870555 (2018).

Newly added authors in revised manuscript

The following authors have made significant contributions to the manuscript revision. All authors agreed on adding them to the author list. Specific contributions by the new authors are listed below:

Kuldeep Mahato

- Performed sensor material characterization.
- Executed Interfacing PVA gel characterization.
- Executed the eChem sensor development and modification.
- Performed in vitro testing of the electrodes including:
 - Multiple scan stability
 - Long run stability
 - Stability at varying humidity
 - Stability at varying temperatures.
- Addressed eChem comments for the revision.
- Participated in on body testing for the eChem sensor.
- Interfacing SEBS characterization and integrated exercise sensing experiments.

Abhinav Uppal

- Participated in EEG data collection with ePhys sensors with motion.
- Led headset-based EEG data collection in conjunction with accelerometer inputs, exploring the feasibility to combine with in-ear EEG data.
- Implemented ICA based motion artifact reduction algorithm on the collected EEG data.
- Implemented ASR based motion artifact reduction algorithm on the collected EEG data.
- Composed the motion artifact analysis section and reviewed the other parts of the manuscript, response letter.

William Chen

- Supported with the SEBS based inks (SEBS, carbon, silver) preparation for electrodes printing.
- Screen printing of the electrodes.
- Inspected the electrode after development.

Rebuttal 2

Response to referees

Editor Comments

Thank you for your revised manuscript, "Unobtrusive In-Ear Integrated Physiological and Metabolic Sensors for Continuous Brain-Body Activity Monitoring", which has been seen by the original reviewers. In their reports, which you will find at the end of this message, you will see that the reviewers acknowledge the improvements to the work and raise a few additional technical questions and suggestions that should help you improve the discussion and reporting quality of the work.

We sincerely appreciate the editorial team and reviewers for their constructive comments that have been very helpful to further improve the manuscript. Here, we provide point-by-point responses to address each of the comments with explicit reference to the changes made accordingly in the revised manuscript. We first address two similar comments raised by both reviewers and then the remaining comments by the two reviewers individually.

Response to all reviewers

Reviewer #1: The discussion on the clinical relevance of combined brain state and metabolite monitoring was insightful. However, can the authors provide any direct, experimental evidence from the work that supports the synergistic effect of these two modalities in diagnosing disorders?

and

Reviewer #2: The introduction much more clearly describes the different modalities and benefits of integrating multiple physiological modalities into a single device. The authors mention that while both EEG and chemical sensing can both monitor cognitive changes. EEG is better suited for acute neuromodulation, rehabilitation, and brain-machine interfacing while metabolite monitoring can provide insight into longer term changes. It stands to reason that having both measures can help account for day-to-day variation in user-specific EEGs (a problem for many epilepsy monitoring applications) and provide vital training data for different machine learning algorithms. I would suggest the authors add a sentence around lines 33-45 to specify a precise example for how these measures can be used to solve a current problem as opposed to 'generally providing better monitoring solutions'.

The synergistic effects of brain state and metabolite monitoring can lead to important applications towards neurodegenerative disease detection. As the reviewers suggested, we have expanded our discussion at lines 42–45 of the revised manuscript

to highlight application of combined EEG brain state and lactate metabolic sensing modalities to one specific disorder diagnosis and monitoring scenario: epileptic seizures. EEG can support, yet rarely prove, the diagnosis of an epileptic seizure. EEG helps the diagnosis in the rare event of recording being active during an epileptic seizure, but frequently is unspecific between seizures^{13,68}. Previous work^{13,14} has demonstrated the synergy with separate EEG and lactate sensors mounted on the body for more accurate epileptic seizure detection. Specifically, serum lactate in patients with generalized tonic-clonic seizures was significantly increased in comparison to other forms of seizure incidences⁶⁸. Serum lactate levels in patients with generalized tonic-clonic seizures were significantly higher than in the patients with psychogenic non-epileptic seizures and syncope¹³. Serum lactate level, in turn, has been proven to be highly correlated with sweat lactate as demonstrated in previous work⁶⁹ and confirmed in our work (Fig. 3h, j). The synergy between EEG and lactate sensing modalities for accurate epileptic seizure detection provides an important use case for the in-ear sensors demonstrated here that will be the subject of clinical validation in future work. The specific interactions between these two modalities in differentiating generalized epileptic seizures from psychogenic non-epileptic and syncopal events are very important clinically, but beyond the scope of the present study focusing on reliable and unobtrusive sensing, which in turn enables greater accessibility for future clinical use. However, we believe we have demonstrated essential capability of our sensing platform to serve this application by validating the equivalent of the following clinical protocols in these previous studies to improve seizure detection accuracy:

- Measure serum lactate after an EEG-confirmed seizure or syncope¹⁴.
- Serum lactate level when measured in the first 2 hr after the index event has a high clinical utility in the differential diagnosis of generalized tonic-clonic seizures, psychogenic non-epileptic seizures, and syncope. With concomitant clinical signs and physical examination findings besides EEG, elevated levels of lactate should be taken into account¹³.

Reviewer #1

I highly appreciate the authors' comprehensive response that provides detailed insights into highlight the unique strengths, breakthroughs, and creativity of the work. The detailed comparative analysis of the reported methodology with previous approaches is appreciated by the reviewer. It is a great pleasure to contribute to the refinement and improvement of such an interesting study. However, to further improve the clarity and depth of the current manuscript, there are a few additional questions and concerns that needs to be addressed:

Thank you very much for the feedback! We have made changes accordingly:

1.The discussion on the clinical relevance of combined brain state and metabolite monitoring was insightful. However, can the authors provide any direct, experimental evidence from the work that supports the synergistic effect of these two modalities in diagnosing disorders?

Please refer to our response and reference to changed manuscript text in the above "Response to all reviewers" section.

2.The authors have thoroughly described the advantages of the ear as a site for integrated sensing, from its anatomical stability and proximity to the brain to its high density of blood vessels and sweat glands. Regarding the combination of brain state and metabolic monitoring, I found your argument compelling. Can the authors elaborate on the trade-offs, if any, of combining electrophysiological and electrochemical sensing in your in-ear sensors, given the ear's complex and enclosed geometry? For example, does the addition of one sensing modality negatively impact the performance or accuracy of the other in any way.

The reviewer points out a very important trade-off that may affect the integrated sensors' performance. The ePhys and eChem sensors were located next to each other to effectively utilize the limited space available in the ear. We had two main design considerations to minimize such trade-off. First, the 3D ePhys electrodes and the PVA hydrogel-covered eChem electrodes were physically separated by a SEBS-covered substrate in between. Second, the PVA hydrogel was chosen for its hydrophilicity to collect sweat. Compared with the dome-shaped ePhys electrode, it had a much larger and flatter contact surface with the skin. The hydrophilicity and flat surface made it more likely to collect sweat. To verify these and other considerations in the sensor design, we validated the crosstalk between both sensing modalities in detail, observing minor transient crosstalk only at the initial onset of the lactate sensing session. Apart from the transient initial crosstalk, we did not observe any further negative trade-off

effects of the integration of the two sensing modalities in the ear, as we briefly discussed in lines 231–243 and extensively covered in Supplementary Note 4.

3. Although the authors have mentioned the sizes (small, medium, large) of the earphone silicone tip, could the authors specify the range of sizes these categories cover? This will help to understand if there are any limitations in the user base due to size restrictions. In the experiments, did the authors find any differences in sensor readings based on the size of the earphone silicone tip used? If so, how could the author account for these differences in your analysis?

The diameter dimensions of the earphone tips are: 11.6 mm (small), 12 mm (medium), and 13.5 mm (large). When assembling the sensors, we made sure to orient the eChem sensors towards the tragus (described in the "Sweat gland mapping" section). Consequently, the contact locations of the ePhys sensors in the ear differed slightly depending on the earphone tip size, but they were all oriented toward the temporal lobe. In terms of signal quality and the analyzed features (such as EEG alpha modulation and sweat lactate), we did not notice any differences in sensor readings. Before conducting experiments, for all earphone tip sizes, we conducted an impedance check to assure that the sensors were in firm contact with the ear (described in "Electrophysiological measurement system integration and on-body setup"). We added specifications of earphone tip size to the "Subject-specific tight fitting inside the ear" section (lines 357–358).

4. The design integrates a sensor into an earbud, presumably to be used in conjunction with a personal audio device. It's good to see that the PVA-hydrogel interfacing to the eChem sensor provides a spongy microporous rough surface to help stability. But are there any effects? Have the authors done any testing to ensure that the added sensor does not interfere with the audio quality of the earbuds?

We chose PVA hydrogel because, as the reviewer indicated: 1) its microporous, sponge-like structure helps to stabilize the device; and 2) its hydrophilicity helps to collect sweat. Compared to the dome-shaped ePhys electrode, it had a significantly larger and flattened skin contact area. We found no negative effects from the addition of PVA-hydrogel. Regarding the audio quality of the earbuds, the hydrogel is not in the way of the sound pathway. This was evidenced in the co-sensing crosstalk characterizations, showing accurate recording of 40 Hz auditory steady-state response to audio stimuli presented to the subject (Fig. 3m, n, Extended Data Fig. 9). This suggests the audio quality from the earphones is not adversely affected by the PVA hydrogel. We have added mention of the sensors' design preserving the integrity of the earbud's audio functionalities (lines 241–242).

5.Regarding the supplementary video, is it possible to add a quantitative measure, such as a stability index, to show the firmness of the sensor in place during intense physical motion? This would lend more credibility to the demonstration.

To add credibility to mechanical stability of the sensor mounted in the ear, we have extended our recording setup and updated supplementary video S3. In the new video, an additional over-the-head camera was head-mounted serving two purposes: 1) To obtain a close-up view of the sensors; and 2) To provide a steady frame of reference fixated to the head, in order to directly observe any relative displacements between the in-ear sensors and the ear. In the video, we displayed both the view from the primary camera as well as the view from the over-the-head camera. Throughout the entire physical activity session, we did not observe detachment of the sensors from the ear, nor did we register any discernable movement between the sensors and the ear.

For a concrete stability index, we inspected the sensors-ear contact from the first and last frames of video S3 at the same view angle. Panels c and d in the newly added Fig. S14 show that the contours of the sensor-ear contact match precisely before and after a series of physical activities, with no discernible difference. In addition, we would like to point out that the recordings of multiple sensing modalities throughout exercise, such as lactate shown in Fig. 4, provides further evidence that the sensors remained in firm contact with the ear. Indeed, lactate recording necessitates a stable sensor-ear contact for sweating collection.

Along with the updated video S3 and newly added Fig. S14, we have modified the description for Video S3 in the Supplementary Videos section (Supplementary Information lines 859–870).

Fig. S14 | Validation of mechanical stability of in-ear sensors in Video S3. **a.** Side view showing the over-the-head camera for a close-up head-referenced view of the in-ear sensors and the earphone. **b.** Rear view showing the DAQ and its connection to the in-ear sensors via fPCB. The in-ear sensors' stability was evaluated by comparing the **c.** first and **d.** last frame of video S3 showing the earphone and cable extensions of the sensors at the same view angle with no discernable change in the contours of their contact to the ear.

6. The authors have mentioned that the DAQ was attached to the subject's collar. Was this the most effective position for data collection? Was comfort or interference ever an issue with this setup?

The DAQ configuration was chosen for the flexibility of the experiments even though it would not be the most optimal for use in a product where ease and comfort of the user are paramount. Specifically, the collar position was chosen for convenience in the EEG characterization without the need for more fixation of wiring. Using the current mounting setup, either on the collar or on the headband, we did not experience any discomfort or interference with the recording.

We note that miniaturization and integration efforts of the in-ear DAQ, including our own work⁷⁰, enable greater flexibility for its mounting position. In our updated new video S3, we showed mounting the DAQ to a headband (Fig. S14). We briefly added mention of future developments for fully integrated earphones with miniature integrated DAQ (lines 301–302).

7.The authors mentioned using a silicone hook for attaching the sensors and flex PCB tightly to the earphone, but also noted that it could be eliminated in future designs. Could you elaborate on the reasons why you consider this improvement, and what alternatives are you contemplating?

Thank you for pointing this out. We did use the silicone hook as a sensor anchor. According to our experiments, it improves stability but is not necessary. The appropriately sized earphone tip, 3D ePhys electrodes, and PVA gel cushioning structure have already achieved a tight fit between the sensors and the ear. We also utilized the hook because we constructed the ePhys REF and DRL electrodes on it (Fig. 1h, i). The anchoring mechanism and the ePhys REF, DRL electrodes can be integrated into a custom-designed earphone case, which our work did not address but would be a viable step towards a higher degree of integration, while eliminating the silicone hook. We briefly addressed these points at lines 366-371.

Also towards further integration, the earphone tip will be directly integrated with the sensors. The separate flex PCB will be eliminated along with the silicone hook. Minimizing the signal path and interconnection can also reduce the recorded motion artifacts, as mentioned at line 542 of Supplementary Information.

8.It's good to see that the authors are considering industrial consortiums like NEXTFLEX for scaling up solutions. While the large-scale production feasibility is addressed in theory, have any practical tests been performed yet to confirm the success of this automated process? The authors mention using 3D-printed custom molds for future production. Could the authors provide more detail on how this would work and any potential benefits or challenges you foresee? How will the automation process maintain the customization aspect of the sensor, particularly in relation to the variations in users' ear sizes and shapes?

As depicted in Extended Data Fig. 2, a number of manufacturing procedures have been automated using screen-printing techniques. Several manual material deposition techniques (Extended Data Fig. 2d, e, i) and bonding procedures (Extended Data Fig. 2i, k) prevent the current fabrication from being entirely automated due to restrictions in automation tools. Industrial consortia such as NEXTFLEX have been extensively used to transform research prototypes into commercial devices. NEXTFLEX's direct

printing and encapsulation services should address our current limitations without too much difficulty⁷¹.

Regarding 3D-printed custom molds, we have already used such molds to fabricate the 3D ePhys electrodes (detailed in the “Three-dimensional ePhys electrode simulation and fabrication” section), and we believe similar approaches can improve the silver epoxy and ePhys electrode backing structure deposition processes.

Regarding the large variance in ear shapes across the user population, the sensor dimensions in this work accommodate three sizes of earphone tips. Consequently, the fabrication parameters require no customization as of now. Moreover, given that industrial consortia such as NEXTFLEX specialize in flexible and stretchable form factors, conforming to various ear sizes and shapes should not pose a problem.

We briefly addressed these points at lines 368-371.

9.The authors mentioned that the impedance-checking procedure is typical for EEG instruments. While this makes sense for professionals used to handling EEG equipment, how would a layperson, an everyday user, navigate this procedure? Do the authors anticipate developing a user-friendly interface or guide for this?

In previous work we used a threshold of 1 MΩ contact impedance to assess functional operation of the electrodes. For everyday use of impedance checking to ensure signal integrity without the need for any user action or measures, this procedure could be hardcoded in the control program of the integrated system. For instance, colored LED indicators or alerts over a mobile device can provide direct visual feedback. The development of a user-friendly interface is of paramount importance to the adoption of this technology, however it is beyond the scope of the current work reported here.

10.It's great to see the questionnaire results showing high ratings for usability and wearability. Can the authors provide more details about the six categories used for assessment? Were there any outliers or common issues identified in the questionnaire results that could lead to improvements in future iterations of the sensor design?

Descriptions of the 6 assessment categories of the questionnaire are provided in Supplementary Note 5. The construction of the 6 categories follows a widely-cited research⁶⁴ that develops this assessment tool for comfort assessment of various wearable technologies.

The in-ear integrated sensors in this work received an average score greater than 4 out of 5 across all six usability and wearability categories of the survey, indicating that participants were overall satisfied (>80% rating percentile) with the usability and wearability of the sensors. Consistent with feedback received from this peer review,

future iterations of the sensor design will improve aspects of fabrication automation, low-power IC designs, and electronic integration.

Reviewer #2

Thank you for your updated manuscript, it is substantially improved with updated experimental results to showcase a significantly more compelling piece of work. Some minor comments are written below.

Thank you so much for the feedback!

The introduction much more clearly describes the different modalities and benefits of integrating multiple physiological modalities into a single device. The authors mention that while both EEG and chemical sensing can both monitor cognitive changes. EEG is better suited for acute neuromodulation, rehabilitation, and brain-machine interfacing while metabolite monitoring can provide insight into longer term changes. It stands to reason that having both measures can help account for day-to-day variation in user-specific EEGs (a problem for many epilepsy monitoring applications) and provide vital training data for different machine learning algorithms. I would suggest the authors add a sentence around lines 33-45 to specify a precise example for how these measures can be used to solve a current problem as opposed to 'generally providing better monitoring solutions'.

We have added the sentence to specify a precise example: epileptic seizures (lines 42–45). For more information, please refer to the response to the first comment in the “Response to all reviewer” section.

I appreciate the additional writing that highlights specific issues with commercial metabolic health monitoring solutions (invasiveness or bulky optical equipment requirements) and state of the art wearable platforms (cross talk/bridging + location requirements that negate possibilities for better sensor fusion). I would consider adding a point around lines 73 - 90 that specifies that minimized crosstalk is a benefit of this work over the state of the art. I realize it comes in the subsequent paragraph, but is phrased as just a design parameter as opposed to a system-level improvement over existing solutions.

Great suggestion! We have amended a sentence to underscore the benefit of minimized crosstalk (line 91).

The adjustments to the pre/post exercise experimental procedure make for a more compelling demonstration.

I appreciate the difficulty in training subjects/redone analysis for the full EEG & EEG + Lactate trials, but electrode-ear impedance characterization is a straight forward

experiment. Having six trials from two subjects isn't much of an average. At the very least the plots should be labeled with ($n = 6$) so that readers are aware of the small number of trials. The DC offset characterization plots should also be labelled with an ($n = 96$) - which is a much more useful amount of samples. Future works should prioritise more in-ear characterization trials across more than 2 users.

Thank you and we totally agree! We have updated Fig. 2, Fig. 3 and Fig. 4 along with their captions to clearly specify the number of trials conducted for each experiment. The number of trials for all experiments have been mentioned in each experiment's Method section as well as Table S4. As the reviewer suggested, in future work, characterization and clinical validation across a large population pool will be a priority. We have amended the statements in the limitation section accordingly (line 302).

I'm a little confused by Fig 3 H & J. After staring at it for a while (and reading lines 207-225), I recognize that you are plotting the change in sweat lactate measurements in each trial relative to the 'ground truth' blood-lactate meter. What the plot makes it seem like is that there is no change in sweat-lactate after enzyme modification? I think it's a scaling issue (there is currently some change across the 40 min trial - but it's less than 0.25 uA after you've improved the sensors...). S11 is significantly clearer than fig 3j. Is there a reason that isn't a main figure? I would suggest double checking the plotting code and addressing any possible confusion about the lack of change in sweat lactate around lines 207-225.

Thanks for pointing out the complexity of the data representation. We have updated Fig. 3 to improve clarity.

In Fig. 3 g–j, we intended to show how electrode modifications achieved the eChem sensors' lactate sensing capability. Specifically, Fig. 3 g–j are controlled experiments with and without enzyme (lactate oxidase) modifications to achieve such sensing capability. Fig. 3g shows the response of the eChem sensor with the surface modification (Prussian blue mediated Carbon ink / Lactate Oxidase / Chitosan). Since it has an enzyme (lactate oxidase) selective to the lactate, it responds to sweat lactate concentration. The signal obtained by the developed sensors was compared with the blood lactate concentrations at various points (i.e. *before* exercising, *during* the exercise, and *after* exercise or resting period). This comparison is based on the empirical relation of the blood and sweat lactate correlation, where the increase in the blood lactate is directly correlated to the sweat lactate concentrations. This has been shown in Fig. 3h, where the increase in the lactate concentration at these points (i.e. *during* and *after*) is attributed to two important factors: 1) the presence of sweat; and 2) the presence of active enzymes in the sensor modification. The absence of either factor leads to “no signal” as in the points “*before* the exercise” (no sweat). As a comparison, the control study shown in Fig. 3i, the sensor modification does not

contain the lactate oxidase (Prussian blue mediated Carbon ink / Chitosan); there is no signal even if there is sweat, which is reflected Fig. 3j's "during" and "after" points (red bars). However, during this period, the participant was exercising. Thus, there was a buildup of lactate in his blood, which was observed. Thus, the blood lactate is observed elevated "during" and "after" points (blue bars). The overall study confirms that the presence of the enzyme and the sweat is required for signal generation.

We apologize for the confusion of potential scaling issue. The current change for the recorded sweating session in Fig. 3g was $-0.4 \mu\text{A}$ instead of $-0.25 \mu\text{A}$, which is reported at line 216. $-0.4 \mu\text{A}$ of lactate current change was in line with what we measured in the co-sensing experiment ($-0.47 (\pm 0.10) \mu\text{A}$, line 284).

Regarding Fig. S11, Fig. 3 g–j and Fig. S11 are for different studies. Figure S11 is not the repetition of controlled experiments Fig. 3g or i. Instead, Fig. S11 is an extended study of the co-sensing experiment, which has already been reported in Fig. 4e and Extended Data Fig. 10o as subject 5. We do see why the reviewer recommends Fig. S11 since it's clear and indicated different phases of the experiments. Thus we kept the current Fig. 3g–j to still demonstrate the controlled experiment results, but implemented the following changes to Fig. 3 g–j to improve clarity:

- Added legends to Fig. 3 g and i for controlled conditions.
- Added markers to Fig. 3g and i to indicate the timing of blood lactate recordings corresponding to values shown in Fig. 3 h and j.
- Added note " $\Delta i = -0.4 \mu\text{A}$ " to clearly show the current change.

Additional References

- 68 Matz, O., Zdebik, C., Zechbauer, S., Bündgens, L., Litmathe, J., Willmes, K., Schulz, J. B. & Dafotakis, M. Lactate as a diagnostic marker in transient loss of consciousness. *Seizure* **40**, 71-75, doi:10.1016/j.seizure.2016.06.014 (2016).
- 69 Sakharov, D. A., Shkurnikov, M. U., Vagin, M. Y., Yashina, E. I., Karyakin, A. A. & Tonevitsky, A. G. Relationship between Lactate Concentrations in Active Muscle Sweat and Whole Blood. *Bulletin of Experimental Biology and Medicine* **150**, 83-85, doi:10.1007/s10517-010-1075-0 (2010).
- 70 Paul, A., Lee, M., Xu, Y., Deiss, S. & Cauwenberghs, G. A Versatile In-Ear Biosensing System for Continuous Brain and Health Monitoring. in *2022 IEEE International Symposium on Circuits and Systems (ISCAS)*. (IEEE).
- 71 Bergman, J., Cook, A., Nachnani, S., Mueller, A. & McManus, R. Maturing additively printed flexible electronics for wearable sensing and security. in *Nano-, Bio-, Info-Tech Sensors and Wearable Systems*. 1159008 (SPIE).

Rebuttal 3

Response letter

Editor Comments

Thank you for your revised manuscript, "Unobtrusive In-Ear Integrated Physiological and Metabolic Sensors for Continuous Brain-Body Activity Monitoring". Having consulted with Reviewers #1 and #2 (whose comments you will find at the end of this message), I am pleased to write that we shall be happy to publish the manuscript in Nature Biomedical Engineering.

We will be performing detailed checks on your manuscript, and in due course will send you a checklist detailing our editorial and formatting requirements. You will need to follow these instructions before you upload the final manuscript files.

Thank you for your patience in waiting for the guidelines for the final submission of your manuscript, "Unobtrusive In-Ear Integrated Physiological and Metabolic Sensors for Continuous Brain-Body Activity Monitoring" to Nature Biomedical Engineering. Please carefully follow the instructions provided in the attached file.

For primary research originally submitted after December 1, 2019, we encourage authors to take up transparent peer review. If you are eligible and opt in to transparent peer review, we will publish, as a single supplementary file, all the reviewer comments for all the versions of the manuscript, your rebuttal letters, and the editorial decision letters. When submitting the final version of your manuscript please indicate whether you opt in to transparent peer review. In the interest of confidentiality, we allow redactions to the rebuttal letters and to the reviewer comments. If you are concerned about the release of confidential data, please indicate in the cover letter what specific information you would like to have removed; we cannot incorporate redactions for any other reasons. More information on transparent peer review is available.

We appreciate the efforts made by the reviewers and the editorial team to improve this work. We have prepared the manuscript and materials in accordance with the received instructions and improved clarity based on the received feedback. In addition, we have submitted separate documents detailing all modifications made to the manuscript and supplementary information.

We would like to opt into transparent peer review as we have indicated in the submission system.

Reviewer #1

Based on the authors' response to my previous comments, it is clear that they have adequately addressed all of my concerns. They have provided additional information and evidence to support their claims and have demonstrated that the integration of multiple sensing modalities in the ear does not negatively impact the performance or accuracy of each other. They have also addressed the issue of earphone tip sizes and ensured that the sensor readings remain consistent across different sizes. The authors

have taken measures to preserve the audio quality of the earbuds and have shown that the addition of the sensor does not interfere with the sound pathway. Furthermore, the authors have provided a video demonstrating the mechanical stability of the sensor during intense physical motion and have included a stability index to enhance the credibility of their demonstration. They have explained the reasons for using the current DAQ position and have indicated that future developments will allow for greater flexibility in mounting positions. The authors have successfully addressed all my concerns and have presented a well-designed and promising integrated sensing system for in-ear monitoring. Based on the authors' comprehensive response and the evidence provided, I only have several very minor suggestions regarding the figures in the current manuscript:

1. Ensure that the legends are placed within the respective areas of the labels. This will help readers easily identify which legends correspond to each label. In Figure 2, the legends "ijkl" extend beyond the label area, causing confusion regarding their association with the corresponding figure. Similarly, in Figure 3, the cartoon patterns should be placed within the corresponding label areas.

We have reorganized Fig. 2 and Fig. 3 accordingly to improve clarity.

2. Removal of Meaningless Elements: In Figure 3, it is noted that the dashed lines within subfigure "k" lack meaningful representation. It is suggested to remove these lines unless they serve a specific purpose to avoid any confusion or misleading interpretation.

To avoid confusion or misleading interpretation, we changed the color of the dashed line and also the boundary of the zoom-in inset. We do believe the dashed lines help identifying the corresponding data portion that the inset shows.

3. Improving Data Representation in Figure 4: The representation of data in Figure 4, specifically subfigure "f," requires optimization for enhanced readability.

Following both the reviewer's and editor's suggestions, we have modified the legends to clearly indicate "Thin lines show 5 individual participants". We have also removed the transparency setting of the thin lines and overlay them onto the averaged data to improve clarity.

Reviewer #2

My comments have been adequately addressed. No further comments.

Thank you for all the suggestions!